# Assessment of the Greenland ice sheet - atmosphere feedbacks for the next century with a regional atmospheric model coupled to an ice sheet model

Sébastien Le clec'h[1,2], Sylvie Charbit[1], Aurélien Quiquet[1], Xavier Fettweis[3], Christophe Dumas[1], Masa Kageyama[1], Coraline Wyard[3], and Catherine Ritz[4]

[1]Laboratoire des sciences du climat et de l'environnement, Gif-sur-Yvette, FR
[2]Earth System Science and Department Geografie, Vrije Universiteit Brussel, Brussels, Belgium
[3]Laboratory of Climatology, Department of Geography, University of Liège, Liège, Belgium
[4]Institut des Géosciences de l'Environnement, Université Grenoble-Alpes, CNRS, 38000 Grenoble, France

*Correspondence to:* Sébastien Le clec'h (sebastien.le.clech@vub.be)

**Abstract.** In the context of global warming, a growing attention is paid to the evolution of the Greenland ice sheet (GrIS) and its contribution to sea-level rise at the centennial time scale. Atmosphere-GrIS interactions, such as the temperature-elevation and the albedo feedbacks have the potential to modify the surface energy balance and thus to impact the GrIS surface mass balance (SMB). In turn, changes in the geometrical features of the ice sheet may alter both the climate and the ice dynamics governing the ice sheet evolution. However, changes in ice sheet geometry are generally not explicitly accounted for when simulating atmospheric changes over the Greenland ice sheet in the future. To account for ice sheet-climate interactions, we developed the first two-way synchronously coupled model between a regional atmospheric model (MAR) and a 3D ice sheet model (GRISLI). Using this novel model, we simulate the ice sheet evolution from 2000 to 2150 under a prolonged RCP8.5 scenario. Changes in surface elevation and ice sheet extent simulated by GRISLI have a direct impact on the climate simulated by MAR. They are fed to MAR from 2020 onwards, i.e. when changes in SMB produce significant topography changes in GRISLI. We further assess the importance of the atmosphere-ice sheet feedbacks through the comparison of the two-way coupled experiment with two other simulations based on simpler coupling strategies: i) a one-way coupling with no consideration of any change in ice sheet geometry; ii) an alternative one-way coupling in which the elevation changes feedbacks are parameterised in the ice sheet model (from 2020 onwards) without taking into account the changes in ice sheet topography in the atmospheric model. The two-way coupled experiment simulates an important increase in surface melt below 2000 m of elevation resulting in an important SMB reduction by 2150 and a shift of the equilibrium line towards elevations as high as 2500 m despite a slight increase in SMB over the central plateau due to enhanced snowfall. In relation with these SMB changes, modifications of ice sheet geometry favour ice flux convergence towards the margins, with an increase in ice velocities in the GrIS interior due to increased surface slopes and a decrease in ice velocities at the margins due to decreasing ice thickness. This convergence counteracts the SMB signal in these areas. In the two-way coupling, the SMB is also influenced by changes in fine scale atmospheric dynamical processes, such as the increase in katabatic winds from central to marginal regions induced by increased surface slopes. Altogether, the GrIS contribution to sea-level rise, inferred from variations in ice volume above

floatation, is equal to 20.4 cm in 2150. The comparison between the coupled and the two uncoupled experiments suggests that the effect of the different feedbacks is amplified over time with the most important feedbacks being the SMB-elevation feedbacks. As a result, the experiment with parameterised SMB-elevation feedback provides a sea- level contribution from GrIS in 2150 only 2.5 % lower than the two-way coupled experiment, while the experiment with no feedback is 9.3 % lower. The change in the ablation area in the two-way coupled experiment is much larger than those provided by the two simplest methods, with an underestimation of 11.7 % (resp. 14 %) with parameterised feedbacks (resp. no feedback). In addition, we quantify that computing the GrIS contribution to sea level rise from SMB changes only over a fixed ice sheet mask leads to an overestimation of ice loss of at least 6 % compared to the use of a time variable ice sheet mask. Finally, our results suggest that ice loss estimations are diverging when using the different coupling strategies, with differences from the two-way method becoming significant at the end of the $21^{st}$ century. In particular, even if, averaged over the whole GrIS, the climatic and ice sheet fields are relatively similar, at the local and regional scale there are important differences, highlighting the importance of correctly representing the interactions when interested in basin scale changes.

## 1  Introduction

The Arctic is the region of the Earth experiencing the largest increase in temperature since the pre-industrial era (Serreze and Barry, 2011), with consequences already perceptible on the mass evolution of the Arctic ice caps and the Greenland ice sheet (Rignot et al., 2011). The evolution of the Greenland ice sheet (GrIS) is governed by variations of ice dynamics and surface mass balance (SMB), the latter being defined as the difference between snow accumulation, further transformed into ice, and ablation processes (i.e. surface melting and sublimation). While surface melting strongly depends on the surface energy balance, snowfall is primarily controlled by atmospheric conditions (wind, humidity content, cloudiness. . . ). However, various feedbacks between the atmosphere and the GrIS impact the surface characteristics such as ice extent and thickness. This has potential consequences on ice dynamics (e.g., due to changes in surface slopes) and may lead to SMB variations that can therefore affect the total ice mass. These changes may in turn alter both local and global climate. As an example, changes in near-surface temperature and surface energy balance may occur in response to changes in orography (temperature-elevation feedback) or changes in ice-covered area (planetary albedo feedback; see Lunt et al. 2004 and Vizcaíno et al. (2008, 2015)). On the other hand, topography changes may alter the atmospheric circulation patterns (Doyle and Shapiro, 1999; Petersen et al., 2003; Moore and Renfrew, 2005) causing changes in heat and humidity transports.

Quantifying the balance between these different processes and feedbacks that regulate transient ice sheet change is required to understand and project more confidently the evolution of the GrIS under current and future global warming. Although numerous studies highlighted the importance of correctly representing the interactions between the GrIS topography changes and the atmosphere (Huybrechts et al., 2002; Alley and Joughin, 2012; Edwards et al., 2014a, b; Vizcaíno et al., 2015), only few global or regional models have taken the GrIS topography changes into account to compute the future evolution of the SMB and energy budget over the GrIS. For example, the CMIP5 climate models (Taylor et al., 2012) unanimously represent the ice sheet component with a fixed and constant topography, even under a warm transient climate forcing.

To explore the importance of SMB-elevation feedbacks for the future GrIS evolution, Vizcaíno et al. (2015) used a coarse resolution atmosphere-ocean general circulation model (AOGCM, ECHAM5.2) coupled to an ice sheet model (ISM, SICOPO-LIS3.0) forced under different RCP scenarios (up to 2100) and their extensions (from 2100 to 2300). Compared to a control experiment in which the ISM is forced off-line by the atmospheric model run with the fixed present-day GrIS topography, they found an amplification of ice mass loss of 8–11 % and 24–31 % in 2100 and 2300 respectively, when the elevation feedbacks are taken into account (i.e. in the coupled experiment). This results from the combination of the positive elevation-SMB feedback in low lying areas, the negative feedback related to the elevation-desertification effect in accumulation areas, and the changes of surface slopes resulting from high mass loss in ablation areas and slight snowfall increase in the accumulation zone, enhancing the ice transport from the central regions to the ice margins. Their study is focused on the added value of incorporating the coupled processes. However, as specified in Vizcaíno et al. (2015), their model is not able to accurately reproduce the observed GrIS because 1/ the ice sheet model, based on the shallow-ice approximation (Hutter, 1983), is not designed to properly represent fast ice flows in outlet glaciers and 2/ the resolution of the AGCM (∼3.75°) and the ice sheet model (10 km) are too coarse to correctly capture the steep slopes at the ice margins and the atmospheric processes acting on the SMB calculation.

Using the AGCM NCAR-CAM3 run at different spatial resolutions (T21 to T85) and coupled to the SICOPOLIS ice sheet model, Lofverstrom and Liakka (2018) investigated how the atmospheric model resolution influences the simulated ice sheets at the Last Glacial Maximum. They found that the North American and the Eurasian ice sheets were properly reproduced with the only T85 run. According to the authors, this is likely due to the inability of the atmospheric model to properly capture the temperature and precipitation fields (used to compute the SMB) at lower horizontal resolutions, as a consequence of the poorly resolved planetary waves and smooth topography. However, running high resolution atmospheric models at the global scale requires large computing resources. To circumvent the low resolution, some authors have used the method of elevation classes and are therefore able to offer high resolution in the direction of the slope gradient (e.g., Vizcaíno et al., 2013).

An alternative solution consists in using regional climate models (RCM) to produce high resolution atmospheric fields and much more robust energy balance and SMB calculations. A number of RCMs have been developed for the polar regions such as MAR (Fettweis et al., 2017), RACMO2 (Noël et al., 2015), Polar MM5 (Box, 2013) or HIRHAM5 (Langen et al., 2015). However, the highest resolution of the RCMs is limited by the use of the hydrostatic approximation and often remains below the resolution of Greenland ice sheet models which are generally running at a 5-10 km scale (Bueler and Brown, 2009; Greve et al., 2011; Price et al., 2011) or even below (e.g., Gagliardini et al., 2013). This means that SMB fields must be corrected for resolution (and thus for elevation) differences between the RCM and the ISM. With the aim of investigating the influence of the MAR resolution on the computed SMB fields, Franco et al. (2012) developed a method to downscale each SMB MAR components (snowfall, rainfall, runoff, sublimation and evaporation) onto a finer grid as a function of elevation changes. An alternative approach to correct the SMB field from surface elevation changes is based on statistical relationships between altitude and SMB (Edwards et al., 2014b). Also been derived with MAR, Edwards et al. (2014b) approach compute a SMB-elevation feedback gradient for regions below and above the equilibrium line altitude in the northern and southern parts of GrIS, with limited additional computing resources. However, in both parameterisations by Franco et al. (2012) and Edwards

et al. (2014b), the authors only consider a strict linear relationship between topography and SMB changes. Although changes in temperature can be derived from a linear vertical lapse rate, other processes governing the SMB such as those related to energy balance, precipitation or atmospheric circulation do not follow a linear relationship with the altitude. While this approach may be valid at the local scale for small elevation changes, it may lead to a misrepresentation of the SMB-elevation feedbacks for

substantial changes in altitude, especially at the ice sheet margins.

One of the first requirements to improve the representation of atmosphere-GrIS feedbacks is to use a high resolution atmospheric model (Box, 2013; Langen et al., 2015; Noël et al., 2015; Fettweis et al., 2017) to better represent the elevation gradients and therefore the steep topography near the ice margins in the ablation zone. Additionally, the use of a detailed snow model such as those implemented in MAR (Fettweis et al., 2017) or RACMO2 (Noël et al., 2015) allows a more accurate

description of the surface properties (e.g., snow cover, albedo, surface melting) and therefore a better representation of the surface energy balance and hence of surface mass balance. RCMs developed for polar regions are also able to represent more atmospheric and land surface processes occurring in these regions such as bare ice albedo (Box et al., 2012) and katabatic winds (Ettema et al., 2010; Noël et al., 2014), being also strongly dependent on topography and thus on resolution.

The second fundamental requirement to describe the interactions between atmosphere and GrIS is to represent the ice sheet

topography changes in the atmospheric model by using an ISM (instead of the fixed geometry typically used) to take into account the effects of ice dynamics on the ice sheet topography changes. This can be achieved through a numerical coupling between the RCM and the ISM. More than twenty ice sheet models exist (e.g., Ritz et al., 2001; Bueler and Brown, 2009; Larour et al., 2012; Fürst et al., 2013; de Boer et al., 2014; Pattyn, 2017), and are currently compared in the Ice Sheet Model Intercomparison Project (Nowicki et al., 2016; Goelzer et al., 2018). They represent thermo-dynamical and physical processes

of the GrIS with different levels of complexities (Gagliardini et al., 2013; Saito et al., 2016). They all compute the dynamical response of the GrIS to a given climate forcing such as the SMB and the near surface temperature (ST) fields computed by RCMs (or global models). However, as SMB and ST from the climate models do not take into account the GrIS evolution, the climate forcing used by the ISM could be flawed.

In order to explicitly represent the feedbacks between the GrIS and the atmosphere and to evaluate their impacts on the

ice sheet evolution, we coupled the polar regional climate model MAR (Fettweis et al., 2017) to the GRISLI ice sheet model (Ritz et al., 2001; Quiquet et al., 2012). To assess the importance of an explicit representation of the surface elevation and albedo feedbacks, this coupled experiment is then compared to a one-way coupling experiment, in which the GrIS-atmosphere interactions are not taken into account, and to a third experiment where the effects of topography changes on the simulated SMB are parameterized.

A description of the atmospheric and the ice sheet models is given in Sect. 2. Sect. 3 describes the experimental setup of the three coupling methods considered in this study. The results are presented in Sect. 4. We first describe the coupled experiment in detail before comparing it to the other uncoupled experiments. These sections are followed by a discussion related to the different coupling approaches (Sect. 5) and the conclusions of this study (Sect. 6).

## 2 Models

### 2.1 The MAR atmospheric model

MAR is a regional atmospheric model fully coupled with the land surface model SISVAT (Soil Ice Snow Vegetation Atmosphere Transfer model, see Gallée and Duynkerke 1997) which includes the detailed one-dimensional snow model Crocus (Brun et al., 1992) which simulates fluxes of mass and energy between snow layers and reproduces snow grain properties and their effect on surface albedo. MAR has been developed to simulate the GrIS SMB and has been extensively validated against in situ observations (Fettweis et al., 2017). In MAR the SMB is computed as follows:

$$SMB = SF + RF - SU - RU \tag{1}$$

Where SF, RF, SU and RU represent snowfall, rainfall, sublimation and runoff respectively. Note that RF contributes to the SMB since liquid precipitation may percolate and refreeze at depth either in the snowpack or in the ice column.

The MAR has a horizontal resolution of 25 km x 25 km and 24 vertical levels to describe the atmospheric column in sigma-pressure coordinates (Gallée and Schayes, 1994). The MAR domain covers the Greenland region (6600 grid points), from 60 °W to 20 °W and from 58 °N to 81 °N. SISVAT has 30 levels to represent the snowpack (with a depth of at least 20 m over the permanent ice area) and 7 levels for the soil in the tundra area.

MAR uses the solar radiation scheme of Morcrette et al. (2008). The representation of the atmospheric hydrological cycle (including a cloud microphysical model) is based on Lin et al. (1983) and Kessler (1969). To facilitate the coupling with an ice sheet model that has a higher resolution than MAR, each grid cell is assumed to be covered by at least 0.001 % of tundra and at least 0.001 % of permanent ice. This makes possible the explicit computation of ice sheet SMB outside the original ice-sheet mask. At each time step SISVAT computes the albedo of each surface type as well as the characteristics of the snowpack which are weighted and averaged as a function of the snow and vegetation coverage in each grid point before being exchanged with MAR. The present work uses MAR version 3.6. The differences with previous MAR versions (e.g., Fettweis et al., 2013) are only related to adjustments of some parameters in the representation of cloudiness and bare ice albedo. These new parameterisations allow to better account for the positive feedback that cloud cover exerts on surface melting (Van Tricht et al., 2016) and to represent the impact of melt ponds that strongly reduce surface albedo (Alexander et al., 2014)

At its lateral boundaries, MAR is forced with 6-hourly atmospheric fields (temperature, humidity, wind and surface pressure) and surface oceanic conditions (sea surface temperature and sea ice extent) provided either by reanalysis dataset (such as ERA-interim or NCEP) to reconstruct the recent GrIS climate (1900-2015, Fettweis et al., 2017) or by general circulation models (GCMs) to perform future projections such as those used for the last IPCC report (e.g., Fettweis et al., 2013). As a result, the atmospheric circulation simulated by MAR over the Greenland ice sheet is strongly dependent on the quality of the climatic fields computed by GCMs or reanalyses as an input to the model. Fettweis et al. (2013) have shown that GCMs which satisfactorily simulate the present-day free-atmosphere mean summer temperature at 700 hPa and the large-scale circulation over Greenland at 500 hPa are best suited to force MAR. For the present study we therefore choose to force MAR with the MIROC5 model outputs (Watanabe et al., 2010) because it has been shown to be the best GCM choice from the CMIP5

database to reproduce the present-day climate compared to the results of MAR forced by reanalyses (Fettweis et al., 2013). The greenhouse gas forcing used in MAR (scenario RCP8.5) is the same as that used in the MIROC5 simulation (Watanabe et al., 2010). Except for the experiment presented later in this study in which MAR is coupled to an ice sheet model, the topography of the GrIS as well as the surface types (ocean, tundra and permanent ice) are taken from the Bamber et al. (2013) dataset aggregated on the 25 km grid.

Because the snowpack in the land model requires generally longer time scale than MAR to reach an equilibrium with the atmospheric forcing, here MAR is spun-up for 6 years forced at its lateral boundaries by outputs from MIROC5 from 1970 until 1975 and by an initialised snowpack coming from a previous MAR simulation carried out under present-day conditions (1960-1999). However, in this paper, the MAR results will be analysed for the period spanning from years 2000 to 2150.

## 2.2 The GRISLI Ice sheet model

### 2.2.1 Model description

The GRISLI (GRenoble Ice Shelf and Land Ice) ice sheet model was first developed to compute the dynamical evolution of the Antarctic ice sheet (Ritz et al., 2001; Philippon et al., 2006; Alvarez-Solas et al., 2011a; Quiquet et al., 2018). It has then been successfully applied to the northern hemisphere ice sheets (Peyaud et al., 2007; Alvarez-Solas et al., 2011b; Charbit et al., 2013) and the Greenland ice sheet (Quiquet et al., 2012, 2013). In the present work, we use a 5 km resolution grid covering Greenland (301 x 561 grid points) and 21 evenly spaced vertical levels in the ice and 4 levels in the bedrock. The regular grid is projected on a polar stereographic grid with a standard parallel at 71 °N and a central meridian at 39 °W (same as in Bamber et al. (2013)). GRISLI is a three-dimensional thermo-mechanically coupled ISM computing the temporal evolution of the ice sheet, which is a function of surface mass balance, ice flow and basal melting:

$$\frac{\partial H}{\partial t} = -\nabla(U^G H) + SMB - b_{melt} \tag{2}$$

where t is time, H the ice thickness, $U^G$ the vertically-averaged velocity, $\nabla(U^G H)$ the ice flux divergence, SMB the surface mass balance and $b_{melt}$ the basal melting. Basal melting occurs when the basal temperature is at the pressure melting point. The ice temperature plays a crucial role in the dynamics of the ice sheet because it affects the viscosity, and thus the ice flow in the entire ice column (Ritz et al., 1996, 2001). In turn, heat released by internal ice deformation and basal dragging over the bedrock modifies the temperature. The temperature field is computed by solving a time-dependent heat equation both in the ice and in the bedrock accounting for advection and vertical diffusion processes. At the surface, the boundary condition is provided by the prescribed surface temperature. At the base of the ice sheet, the boundary condition is given either by the geothermal heat flux or by the temperature melting point at the ice-bed interface.

The ice flow is computed using both the shallow ice (Hutter, 1983) and shallow shelf (MacAyeal, 1989) approximations to solve the Stokes equations (Ritz et al., 2001). The shallow ice approximation (SIA) assumes that ice flow is caused only by vertical shear stress, neglecting the longitudinal stresses. This assumption is only valid for slow flowing ice. For fast flowing regions, vertical shearing becomes smaller than longitudinal shearing and the shallow-shelf approximation (SSA), which neglects the vertical stresses, is used. The ice thickness and the ice sheet surface slopes control the SIA and the SSA velocity

components, but the SSA is also governed by basal dragging. Using a hybrid model (i.e. based on both SIA and SSA approximations) allows to better represent the different deformation regimes found in an ice sheet. In GRISLI, the SSA velocity is used as a sliding velocity (Bueler and Brown, 2009) when the basal temperature is at the pressure melting point. In this case, we assume here a power-law basal friction (Weertman, 1957) and the presence of sediments allowing for viscous deformation. The relationship between the basal shear stress ($\tau_b$) and the basal velocity ($u_b$) is expressed as:

$$\tau_b = -\beta u_b \tag{3}$$

where $\beta$ is a time constant but spatially variable basal drag coefficient. For cold base conditions, the sliding velocity is set to zero.

The resulting velocity for every model grid point is the addition of the SIA and SSA components. For floating ice points (ice shelves), we assume no basal drag. In addition, if the ice thickness of the floating ice shelves is below 250 m and if no neighbouring points are grounded, the point is removed and the corresponding ice mass loss is considered as a calving flux. Determination of the grounding line position is based on a floatation criterion.

The isostatic adjustment in response to ice loading changes is governed by the relaxation of the asthenosphere with a characteristic time constant of 3000 years and by the deformation of an elastic lithosphere (Le Meur and Huybrechts, 1996).

The climatic forcing is given by the mean annual SMB and the mean annual ST. Because seasonal variations of surface temperature are rapidly dampened, ST is considered as a good approximation of the bottom snowpack temperature. The initial GrIS surface and bedrock topographies come from Bamber et al. (2013) and the geothermal heat flux is taken from Fox Maule et al. (2005).

### 2.2.2  Initialisation procedure

Due to the long time scale response of the ice sheet to a given climate forcing, a proper initialisation of the model is required before performing forward experiments. For future sea-level projections, the aim of the initialisation is to start the simulations from a present-day equilibrated ice sheet geometry as close as possible to the observed one while ensuring consistency between internal properties of the ice sheet (e.g. basal sliding velocities and vertical profile of temperature) with the climate forcing. Here we use an inverse method of the basal sliding velocities in order to reduce the mismatch between the simulated GRISLI ice thickness and the observed one (Bamber et al., 2013). The method is fully described in Le Clec'h et al. (2018). Below we only remind the basic principles of our approach.

The procedure starts from initial conditions (i.e. vertical temperature and velocity profiles, first guess of the $\beta$ coefficient) coming previous GRISLI simulations carried out within the framework of the Ice2Sea project (Edwards et al., 2014a) and from the present-day observed topography (Bamber et al., 2013). The SMB and ST climatological means computed by MAR-MIROC5 (Fettweis et al., 2013) are used as climate forcing.

Our initialization procedure is based on an iterative process divided in two main steps:

1/ The first step consists in the iterative adjustment of the spatially-varying basal drag coefficient related to sliding velocities through Eq. 3. At each model time step, the ratio of modelled to observed ice thickness (i.e. the data-model mismatch) is used

to adjust the sliding velocities via the basal drag coefficient, while the shearing velocities (SIA velocities) remain unchanged. This adjustment is typically performed for short time periods (a few decades).

2/ After this adjustment phase, the model is let to freely evolve (second step) for a few decades to a few centuries with the last inferred basal drag coefficient.

At the end of this second step, we obtain a new GrIS topography and, thus, a new data-model mismatch, which is used to start a new cycle in which the first and the second steps are repeated. This new cycle starts with exactly the same initial and boundary conditions as those used for the first cycle, except for the basal drag coefficient and the data-model ice thickness mismatch which are inferred from the values computed at the previous cycle. The overall process is stopped when the ice thickness root mean square error is not significantly improved from one cycle to the other. This ensures a good compromise between the reduction of the mismatch between observed and simulated ice thickness and the rapidity of the convergence of the initialisation procedure. In the present paper, the best fit with observations (RMSE = 63 m) is obtained for a first step duration of 20 years, a duration of 200 years for the relaxation GRISLI simulation (second step) and 8 iterative cycles.

This method is based on the same basic principles as that of Pollard and DeConto (2012) except that their basal drag coefficient is adjusted as a function of the difference between modelled and observed ice surface elevation while we use the ice thickness ratio instead. Moreover, while the method suggested by Pollard and DeConto (2012) requires long (multi-millennial) integrations for the method to converge, we use an iterative method of short (decadal to centennial) integrations starting from the observed ice thickness allowing a more rapid convergence.

To further reduce the model drift in terms of ice volume and before starting the forward GRISLI experiments, a 2000-yr GRISLI relaxation run is performed as a continuation of the second step of the last iteration (i.e. after the end of the $8^{th}$ cycle). As such, the value of the basal drag coefficient is that obtained at the end of the first step of the $8^{th}$ cycle. Over the last 150 years of this free-evolving simulation, the model drift is only $\pm\,10^{-5}$ mm yr$^{-1}$ sea-level equivalent. At the end of this 2000-yr simulation, the simulated GrIS topography is slightly different from the observations (RMSE = 132 m, see Fig. S1). It will be referred hereafter to as $S_{ctrl}$ and will be used as initial topography for the transient GRISLI simulations described in the following.

For all the simulations presented in this study, including those carried out within the iterative initialisation framework, we apply a strong negative SMB value outside the observed ice sheet extent (Bamber et al., 2013). This avoids ice growth where there is none in reality and allows to correct for both the potential atmospheric model biases (e.g., positive SMB values over tundra areas) and the initialisation procedure biases (i.e. too strong ice export towards the margins). However, for GrIS projections run the RCP8.5 forcing scenario, this condition has only a limited impact on GrIS contribution to sea-level rise since the ice extent will likely to keep on retreating over the next centuries.

## 3  Coupling methods

The aim of this study is to assess to what extent accounting for the atmosphere-GrIS interactions influences the GrIS evolution in terms of changes in SMB, ST, ice thickness and SLR. To achieve this goal, we designed three experiments based on coupling

methods of different complexities to account for the interactions between MAR and GRISLI. For all the experiments described below, the climatic forcing is designed as follows: MAR is forced at its lateral boundaries by transient MIROC5 atmospheric fields from the CMIP5 historical run (1970—2005) and RCP8.5 scenario (2006—2100). In order to extend the MAR simulation until 2150 and in the absence of a MIROC5 simulation performed under a prolonged RCP8.5 scenario (i.e. after 2100), MAR is forced from 2101 to 2150 with the 2095-yr MIROC5 climate. We chose the year 2095 because, averaged over the entire GrIS, the 2095 mean climate is one of the closest to the decadal 2090—2100 one. This implies that both climate changes and large-scale inter-annual variability are neglected beyond 2100.

## 3.1   The No Feedback experiment

The first one (referred hereafter to as NF) is based on a one-way coupling approach in which GRISLI is forced by the climatic outputs (SMB and ST) obtained from the MAR simulation spanning from 2000 to 2150. The aim of this experiment is to examine the ice sheet response to the climatic forcing without accounting for the feedbacks related to GrIS changes. This means that, while the ice sheet mask and topography evolve freely in the ISM, MAR is run throughout the simulation on a fixed observed ice sheet geometry and does not see any changes in the ice sheet mask. Using an inverse distance weighting method, the SMB and the ST are first interpolated on the GRISLI grid to account for the difference of resolution between both models (25 km vs 5 km). To account for the differences in surface elevations between the 25 km and 5 km Bamber et al. (2013) topographies, we also apply a vertical correction following Franco et al. (2012) who derived a local vertical gradient for the SMB as a function of altitude. Thus, this method allows to generate a 5 km resolution SMB entirely adapted to the finer scale features of the 5 km Bamber et al. (2013) topography. While this procedure can be followed at the daily time scale (Noël et al., 2016), in the present study, the vertical gradients are averaged at the annual time scale and used as corrective factors to downscale the SMB and ST fields onto the 5 km grid at the end of each model year. Note that while the topography and the ice sheet mask evolve freely in the ISM, MAR uses a fixed ice mask deduced from Bamber et al. (2013) in the NF experiment.

## 3.2   The Parameterised Feedbacks experiment

In the second experiment (referred to as PF in the following), the SMB and ST fields simulated by MAR are corrected each year following the method of Franco et al. (2012) to account for the evolution of the simulated GRISLI topography. This correction is made from 2020 onwards, as changes in SMB through 2006—2020 do not produce any significant topography changes in GRISLI. The new corrected SMB and ST values are computed at the altitude S(t) defined on the 5 km grid as:

$$S(t) = S_{Bamber} + \triangle S_{GRISLI}(t) \tag{4}$$

where $S_{Bamber}$ is the present-day observed topography defined on the 5 km GRISLI grid and $\triangle S_{GRISLI}(t)$ the difference between the altitude simulated by GRISLI at time t and $S_{ctrl}$. Due to the topography differences between MAR and GRISLI, this approach has been chosen to avoid large inconsistencies between the SMB and ST fields computed by MAR and the ones corrected to account for the GRISLI topography.

This method offers the possibility to account artificially for the elevation feedbacks when using existing RCM simulations in which the topography and the ice sheet mask are kept constant. As such, it is also transferable to any ice sheet model. However, as in the NF experiment, the changes in GrIS geometry have no consequence on the climate as simulated by the atmospheric model.

## 3.3 The two-way coupling experiment

The third method (2W in the following) is based on a two-way coupling strategy between MAR and GRISLI. Both models used the same boundary and initial conditions as those of the NF and PF experiments. At the end of a MAR model year, MAR is paused and GRISLI is forced by the downscaled SMB and ST fields with the method of Franco et al. (2012) as in PF (Eq. 4). Then, GRISLI computes a new 5 km GrIS topography and a new ice extent at a 5 km resolution. This new GrIS topography is then aggregated (i.e. geographically averaged) at the yearly time scale onto the 25 km MAR grid. The number of ice covered GRISLI grid points within a MAR grid cell relative to the number of ice-free GRISLI grid points is used to compute the new ice extent in MAR and to update the fraction of tundra relative to ice/snow covered surface type for the subsequent MAR run. To account for the differences between MAR and GRISLI topographies, the surface elevation which is aggregated onto MAR is computed from GRISLI surface elevation anomalies added to the present-day observed topography (Eq. 4). It is then used as the updated surface elevation. As previously mentioned, topography changes are negligible before 2020. Hence, changes in ice sheet geometry are fed to MAR only after this date. Compared to the NF and PF approaches, this two-way coupled method is the most accurate to represent the GrIS-atmosphere feedbacks.

## 4 Results

### 4.1 The Greenland ice sheet evolution in the 2W experiment

#### 4.1.1 Changes in the forcing climate

The evolution of the SMB and of its different components simulated in the 2W experiment (Eq. 1) and integrated over the entire GrIS is displayed in Fig. 1. During the 2000—2040 period, the averaged SMB remains positive with a mean value equal to $280 \pm 95$ Gt yr$^{-1}$ (where the notation $\pm$ represents the standard deviation computed from yearly values) but slightly decreases by 4 Gt yr$^{-1}$. This decrease becomes substantially stronger from 2040 to 2100 (-17 Gt yr$^{-1}$ on average), and the mean SMB reaches strong negative values (-638 $\pm$ 271 Gt yr$^{-1}$ over the 2090—2100 period). As the same MIROC5 year 2095 is repeatedly used to force MAR after 2100, there is no longer inter-annual variability and the integrated SMB remains quite stable between 2100 and 2150 (-812 $\pm$ 13 Gt yr$^{-1}$). Indeed, MAR generates its own surface boundary layer fields which are not impacted by the MIROC5 forcing, explaining a slight SMB increase of $\sim$1 Gt yr$^{-1}$ over the last 50 years. Throughout the simulation, the evolution of the SMB signal is dominated by surface runoff whose increase rate (in absolute value) ranges from 5 Gt yr$^{-1}$ (2000—2040) to 19 Gt yr$^{-1}$ (2040—2100). After 2100, it slightly decreases ($\sim$2 Gt yr$^{-1}$), explaining the slight SMB increase.

The SMB anomaly between the beginning and the end of the 2W experiment is displayed in Fig. 2a. 65 % of the grid points having surface elevations higher than 2000 m are characterized by a positive SMB anomaly, ranging from 0.07 m yr$^{-1}$ ($5^{th}$ percentile) to 0.2 m yr$^{-1}$ ($95^{th}$ percentile) at the end of the simulation. This SMB increase is particularly pronounced in the eastern part of the GrIS between 67 and 70 °N and in the north central part. It is due to a strong increase in snowfall (> 0.5 m yr$^{-1}$, Fig. 2b) which occurs mainly during the winter season in the east and during autumn in the north (Fig. S2). On the other hand, 87 % of the GrIS grid points with surface elevation lower than 2000 m are dominated by an increase in surface runoff (Figs. 2c, S3) and by an increase in the fraction of rainfall over snowfall in summer and in autumn (Fig. S4). As a result, strong negative SMB anomalies are found in these regions ranging from -3.3 m yr$^{-1}$ ($5^{th}$ percentile) to -0.1 m yr$^{-1}$ ($95^{th}$ percentile) and reaching more than -6 m yr$^{-1}$ along the western and the southeastern margins (Fig. 2a).

The equilibrium line altitude (ELA, i.e. altitude for which SMB = 0) increases significantly between the beginning and the end of the 2W experiment, as a consequence of increased runoff for areas below 2000 m. As an example, at around 73.5 °N, on the eastern side of the ice sheet, the ELA moves from ∼1000 m to ∼2500 m (Fig. 3). In other regions, at the end of the 2W experiment, the ELA is generally situated between 1500 and 2000 m high, except in the northern part where it is between 1000 and 1500 m. This shift of ELA towards higher altitudes represents an increase of 24 % of the ablation area between the beginning and the end of the experiment.

The ST anomaly (Fig. 2d) ranges from 2.2 °C ($5^{th}$ percentile) to 6.5 °C ($95^{th}$ percentile) and is characterized by a south-north gradient with the highest values found in the northern part. Beyond 78 °N the ST anomaly reaches locally values greater than 11 °C. This temperature increase from 2000 to 2150 contributes to the amplification of the ablation processes below the ELA. However, while the stronger temperature anomaly is found in the northeastern part of the ice sheet, this region is aslo marked by the increasing snowfall in 2150 compared to 2000 (Fig. 2b) which counteracts the ablation processes.

### 4.1.2 Changes in Greenland ice sheet geometry

Figure 4 displays the ice thickness anomaly between the beginning and the end of the 2W experiment. For surface elevations higher than 2000 m in the northern part, and higher than 2500 m in the central and southern parts of the ice sheet, the ice thickness increases by 5 m on average, with the increase ranging from 1.5 m ($5^{th}$ percentile) to 17 m ($95^{th}$ percentile). On the other hand, in regions whose surface elevation is lower than 2000 m, the ice thickness decreases from -248 m ($5^{th}$ percentile) to -3 m ($95^{th}$ percentile) with a mean value equal to -100 m. As a result of these GrIS ice thickness changes, the surface slope between the central part of the ice sheet and the margins increases. On top of that the ice sheet mask (defined as the fraction of a MAR grid cell with permanent ice cover, e.g. Fig. S5a) decreases by 2.8 ± 0.1 % (mean ± standard deviation computed from yearly values) over the 2140—2150 mean period compared to the 2000—2010 mean period, and some GrIS margin regions become ice free.

### 4.1.3 Impact of ice flow changes

The ice thickness anomaly is due to the complex combination of changes in surface atmospheric conditions (SMB, Fig. 5a), ice flow (ice flux divergence, Fig. 5b) and basal melting (not shown), following the continuity equation (Eq. 2). To quantify the

role of ice flow on the GrIS geometry (Fig. 4), we plotted the ice flux divergence integrated over 150 years (2000—2150, see Fig. 5b). In particular, over the central plateau, the cumulated SMB (Fig. 5a) reaches about 50 m, 40 m of which are transported away by the ice flow (Fig. 5b). As a result, the ice thickness anomaly is reduced to only ∼10 m in this region (Fig. 4). An opposite behaviour is found near the western coast, where the runoff is partly compensated by ice convergence, resulting in a

less negative ice thickness anomaly than that related to the SMB forcing. This shows that ice flow act to counteract ice loss from surface melting, as previously noticed by several authors (Huybrechts and de Wolde, 1999; Goelzer et al., 2013; Edwards et al., 2014a). As a consequence, it appears to be essential to account for ice dynamics to estimate accurately the mass balance of the whole ice sheet.

In turn ice flow is impacted by changes in ice sheet geometry as illustrated by the mean surface velocity anomaly (Fig. 6a).

For regions with surface altitudes between 2000 and 2500 m, the anomaly of the ice flow increases from the inner GrIS areas towards the edges of the ice sheet. The increase in the mean ice flow for the 2140—2150 period compared to 2000—2010 period, ranges from 0.08 m yr$^{-1}$ (5$^{th}$ percentile) to 17 m yr$^{-1}$ (95$^{th}$ percentile). These faster ice velocities at the end of the 2W experiment are mainly explained by a larger surface slope between the central and the margin regions of the ice sheet. This is consistent with information inferred from ice flux divergence as shown in Fig. 5b.

On the contrary, for the margin regions, with altitudes lower than 1500 m, the anomalies of surface ice velocities strongly decrease (Fig. 6a). Compared to the 2000—2010 period, this decrease ranges from -213 m yr$^{-1}$ (5$^{th}$ percentile) to -0.2 m yr$^{-1}$ (95$^{th}$ percentile), and agrees with the decrease in ice thickness (Fig. 4).

The changes in local ice flow between the first and the last 10 years of the 2W experiment are also related to changes in surface slope and ice thickness, particularly at the margins. To investigate the ice flow changes at the local scale, we used the

examples of the Jakobshavn (western coast) and the Kangerlussuaq (eastern coast) glaciers for which the fine scale structures of the ice velocity, obtained after the GRISLI initialisation procedure, are relatively well reproduced compared to the observations (Figs. 7a-b).

For the Jakobshavn glacier, the ice sheet areas located above 1500 m, are mainly characterised by an increase of more than 15 m yr$^{-1}$ (i.e. 10 %) of the vertically-averaged velocity as a result of increasing surface slopes (Fig. 7c). Conversely, areas below

1000 m are dominated by a slow down of the ice flow of more than 200 m yr$^{-1}$ (i.e. 29 %) due to the decreasing ice thickness (Fig. 7e). For altitudes above 500 m, the vertically-averaged velocity is mainly driven by the SIA velocity (Figs. 7e-g). On the contrary, below 500 m, basal sliding velocities are large due to low basal drag coefficient (see Fig. 3 in Le Clec'h et al. 2018) and the SSA velocity component dominates the ice flow (Figs. 7e-i). However, while basal drag is lower in locations below 500 m, the ice flow is limited by the strongly reduced ice thickness (Fig. 4).

The Kangerlussuaq glacier is located in regions where the bedrock is characterised by a succession of valleys surrounded by mountains merging in a canyon where the deepest part is located 100 km away from the coast (Morlighem et al., 2017). The ice flow of the Kangerlussuaq is therefore divided in different branches with increasing ice velocities towards the ice sheet margin and becoming even larger when merging in the canyon (Fig. 7d). As for the Jakobshavn glacier, the ice flow accelerates at the end of the 2W experiment as a consequence of the increase in surface slope for high altitudes (∼2000—2500 m, see

Fig. 4). Conversely, a strong decrease of the ice flow is found in most of margin regions (Fig. 7f) directly related to the ice

thinning (Fig. 4). Contrary to the case of the Jakobshavn glacier that presents large basal sliding velocities only below 500 m, the Kangerlussuaq shows low basal drag coefficients in the entire glacier (see Fig. 3 in Le Clec'h et al. 2018) and thus the ice flow is mainly governed by the SSA component (Fig. 7j).

These results are in line with Peano et al. (2017) who also found a decreasing ice flux at the end of the $21^{st}$ century (w.r.t

1970) in downstream regions of the Jakobshavn and Kangerlussuaq glaciers as a consequence of ice sheet thinning at the margins.

## 4.2   Differences between the NF and the 2W experiments.

### 4.2.1   Impact on SMB and ST

To assess the importance of the atmosphere-GrIS feedbacks, we now compare the NF and the 2W experiments. The main SMB

differences between both experiments, averaged over the 2140—2150 period, highlight higher SMB values in NF compared to 2W for altitudes below 2000 m, with the exception of some margin locations in the eastern part (Fig. 8a). This SMB difference is driven by a snowfall increase in low altitude areas (Fig. S6) and by the runoff decrease in NF with respect to 2W (Fig. S7). This decreased runoff results from colder temperatures over the whole GrIS (up to -1 °C in the western and northern parts, Fig. 8b), except in the regions at the very edge of the GrIS, which sees a significant warming (up to 8 °C, Fig. 8b) despite an

increase in ice thickness (w.r.t 2W). The cooling can be explained by the absence of the temperature-altitude feedback in the NF experiment. Indeed, taking this feedback into account in 2W, results in lower altitudes as the ice thickness decreases (Sect. 4.1.2 and Fig. 4) and therefore in warmer 2W temperatures compared to NF. The warming simulated in NF (w.r.t 2W) over the very edge of the ice sheet can be explained by changes in atmospheric circulation.

Indeed, unlike 2W, NF allows for an explicit computation of changes in ice sheet surface slopes due to increased melting at

the margin. This has important consequences on the atmospheric circulation and in particular on the katabatic winds (Fig. 9). Over the ice sheet, the surface slopes simulated in NF in 2150 are less steep compared to those simulated in 2W (discussed in Sect. 4.1.2) lead to a slight increase in katabatic winds (Fig. 9). However, at the ice sheet margin, i.e. where the ice mask in MAR is below 100 %, there is a substantial decrease in surface winds. This stems from the fact that the changes in surface elevation seen by the atmospheric model are computed from the aggregated changes in GRISLI at 5 km. As such, a non-

zero fraction of tundra, which presents no change in surface elevation, results in smaller elevation changes compared to grid cells with permanent ice cover only. This induces in 2W lower surface slopes at the margin with respect to the interior and thus weakened surface winds in these regions. Altogether, the slightly stronger katabatic winds over the ice sheet and their weakening at the margin lead to a cold air convergence towards the ice sheet edge, absent in NF (Figs. 8b, 9 and Figs. S8-S9). Another consequence of the stronger katabatic winds in 2W (w.r.t NF) due to increased surface slopes in the GrIS interior, is

to enhance the atmospheric exchanges along the slope of the ice sheet. The area with lower atmospheric pressure generated by the stronger katabatic winds is filled in by the warmer air coming from higher atmospheric levels in the boundary layer. The warming of the upper part of the boundary layer in 2W combined with the lower surface elevation, explains the ST increases in the interior of the GrIS.

### 4.2.2 Impact on ice thickness and ice dynamics

The most important ice thickness difference between the last ten years of the NF and the 2W experiments is a higher ice thickness in NF compared to 2W. As mentioned in the previous section, this is mainly explained by the positive SMB-elevation feedback in 2W that results in increased surface temperatures compared to NF, and thus increased runoff, when surface eleva-
tion decreases. Areas with this type of behaviour cover most of the Greenland ice sheet slopes and reach the interior of the ice sheet from the western or the northeastern margins. The largest changes occur over the western edge of the GrIS, where the thickening between NF and 2W reaches more than 50 m (Fig. 8c). The ice thickness difference pattern is essentially mimicking the SMB differences between NF and 2W (Fig. 8a), suggesting that the two-way coupling induces only a relatively limited change in ice flow, as shown by the ice flux divergence anomaly (Fig. S10), although the surface velocities (Fig. 6b) are slightly
higher in NF due to higher ice thickness (Fig. 8c).

### 4.3 The PF experiment

As previously described (see Sect. 3.2), the PF experiment is based on a parameterisation of the surface elevation feedbacks. In this section, we present the differences PF-2W in SMB, ST and ice thickness averaged over the 2140-2150 period (Fig. 10) so as to examine the efficiency of this parameterisation. The first key feature is that the PF-2W SMB difference (Fig. 10a) is
less positive than the NF-2W one (Fig. 8a). This results from the fact that the decreasing altitude is taken into account in PF through the altitude feedback parametersisation, leading to smaller differences with the 2W experiment. In most margin areas, the SMB simulated in PF has even become lower compared to 2W due to a much complex representation of the ice sheet climate interactions. Indeed, as mentioned in see Sect. 4.2.1, in the 2W experiment, the GRISLI topography feedbacks onto the MAR simulated climate leads to a cold air convergence at the ice sheet margins and thus to a higher simulated SMB. Cumulated over
the entire GrIS, the PF-2W SMB difference is 28 Gt yr$^{-1}$ (149 Gt yr$^{-1}$ for NF-2W). In the same way, the PF-2W differences in ST and ice thickness (Figs. 10b-c) are also less pronounced than the NF-2W differences (Figs. 8b-c), highlighting the importance of the elevation feedbacks. These results show that over ∼150 years, the topography correction used in PF, makes possible to obtain from an uncoupled experiment simulated fields close to those of the 2W coupled experiment.

To illustrate the spatial variability of the ice thickness response to the different coupling methods, we plotted the ice thickness
differences between NF and 2W (red dots, Fig. 11a), NF and PF (green dots, Fig. 11a) and PF and 2W (Fig. 11b) as a function of the ice sheet altitude. The NF-PF ice thickness differences are mostly positive, while the NF-2W (Fig. 11a) and PF-2W (Fig. 11b) differences yield both positive and negative values, while the NF-PF differences are positive, illustrating the stronger variability in the 2W experiment. For both the 2W and PF experiments, the regions at low to medium elevations are the most sensitive to the coupling approach with the stronger spatial variability of the ice thickness found for altitudes below 1000 m.
For example, the NF-2W ice thickness difference ranges between 31.9 m (5$^{th}$ percentile) and -6.5 m (95$^{th}$ percentile), and between 27.8 m (5$^{th}$ percentile) and 0 m (95$^{th}$ percentile) for the NF-PF case. Overall, the ice thickness differences decrease with increasing altitudes (Fig. 11a) and increase with time (Fig. 11b).

## 4.4 Impact on GrIS contribution to sea-level rise and ice sheet mask

At the end of the simulation (i.e. after 150 model years), the GrIS contribution to sea-level rise (computed from the change in GRISLI ice volume), simulated in the 2W experiment, reaches 20.4 cm, against 18.5 cm and 19.9 cm in the NF and PF experiments respectively (Table 1 and Fig. 12). Owing to the negligible model drift ($\sim 10^{-5}$ mm yr$^{-1}$, see Sect. 2.2.2), these differences only result from the better representation of the GrIS-atmosphere feedbacks in 2W leading to increased runoff due to warmer temperatures (see Sect. 4.2). In 2100 (Table 1), the differences between the three experiments are smaller, with the NF and PF contributions being respectively 4.4 % and 0.4 % lower than the 2W contribution, against 9.3 % and 2.5 % in 2150. These results reflect several key aspects. First they show that the GrIS mass loss substantially accelerates from the second half of the $21^{st}$ century onwards and that the effect of the different feedbacks, as simulated in 2W, is enhanced over time. Figure 12b displays the sea-level anomalies between 2000 and 2100 to better illustrate the divergence of the three experiments as soon as 2025—2030. Secondly they illustrate the effect of the feedbacks thenselves. As an example, accounting for the parameterised feedbacks (PF) leads to an additional SLR contribution (w.r.t NF) of 4.2 % in 2100 (7.6 % in 2150). This is smaller than that reported in Calov et al. (2018) who also used the MAR model to force the hybrid SICOPOLIS3.3 including a representation of subglacial hydrology. However, our estimate is comparable with the 4.3 % additional contribution found by Edwards et al. (2014a) in 2100 who used ECHAM5 and HadCM3 to force MAR simulations under the SRES A1B scenario and five ISM projections, and within the range of uncertainties of the $8 \pm 5$ % additional surface mass loss reported in Fettweis et al. (2013). As for Vizcaíno et al. (2015), they also conclude that the melt-elevation feedbacks, simulated with the ECHAM5.2-SICOPOLIS3.0 coupled model under the RCP8.5 scenario, contribute to 11 % to SMB changes and to 8 % to SLR. While the importance of the SMB-elevation feedback may be dependent on the model itself, the larger contribution found in Vizcaíno et al. (2015) compared to our own study, could be explained by the coarser resolutions of ECHAM5.2 ($\sim 3.75°$) and SICOPOLIS3.0 (for the GrIS, 10 km) with respect to MAR and GRISLI resolutions, implying for example that ablation areas or processes such as katabatic winds are less well represented. Our results also suggest that, at the centennial time scale, the SMB-elevation feedback is the most important since its parametrization in PF allows to reduce the mismatch between the 2W and NF GrIS SLR contributions by 73.7 % (resp. 91.4 %) in 2150 (resp. 2100), the remaining contributions being attributed to atmospheric feedbacks. However, to assess more accurately the relative importance of the elevation feedbacks, a more appropriate procedure would be to cut off the elevation feedbacks in the 2W experiment.

Compared to the NF and the PF experiments for which the ice sheet mask is fixed to observations from 2000 to 2150, the 2W ice sheet extent is reduced by $\sim 2.8$ % in 2150 as a result of increased ablation. As MAR sees the ice sheet retreating over time in 2W concomitantly with the increase in bare ground or tundra fractions (Fig. S5b), the albedo feedback takes place favouring further the ice melting , though counteracted by the katabatic wind anomalies (see Sect. 4.2). Although the ice sheet retreats, the extent of the ablation zone increases with time. This process is faster in 2W than in NF and PF. In 2150, the ablation zone is 14 % (resp. 11.7 %) larger in 2W than in NF (resp. PF) causing 112 Gt yr$^{-1}$ of extra ice ablation in 2W (w.r.t NF). As a consequence, the ELA is located further inland in 2W compared to NF with a maximum inland retreat of 120 km located in northeastern Greenland (Fig. 3).

A widely used method to estimate the GrIS contribution to global sea-level rise is to compute the GrIS mass loss as the time-integral of the SMB computed by an atmospheric model over a fixed ice sheet mask (Fettweis et al., 2013; Church et al., 2013; Meyssignac et al., 2017). In the present study, we use a more complex method since the ice mass variations related to SMB changes are computed by MAR over a changing ice sheet mask and topography as simulated by GRISLI. However, in both the NF and the PF experiments, the atmospheric model does not account for the variations in the ice sheet extent simulated in GRISLI and the ice sheet mask, taken from the observations (Bamber et al., 2013), is kept constant throughout the simulation. Taking the changes in ice sheet mask into account may have strong impacts on the computed GrIS contribution to sea-level rise. To illustrate the influence of the ice sheet mask, we used the SMB outputs from the NF experiment at the MAR resolution and applied the integrated SMB method over the fixed observed ice sheet mask ($SMB_{MSK_{NF}}$) and over the updated 2W mask ($SMB_{MSK_{2W}}$). Differences in SMB values exceed 23 % in 2150 (-842 Gt yr$^{-1}$ for $SMB_{MSK_{NF}}$ against -647 Gt yr$^{-1}$ for $SMB_{MSK_{2W}}$). In the same way, compared to a time variable ice sheet mask, the use of a fixed ice sheet mask overestimates the sea-level rise by ∼6 % in 2150. Though a bit lower, this error has a similar magnitude compared to errors made when the SMB-elevation feedbacks are not taken into account (i.e. 7.6 %) and when all the feedbacks are ignored (i.e. 9.3 %). This strongly suggests that realistic SLR projections cannot neglect the evolution of the ice sheet extent, only accounted for through the use of an ice sheet model.

## 5  Discussion

The evolution of the GrIS and its contribution to sea-level rise presented in this study are the first ones inferred from a regional atmospheric model synchronously coupled to an ice sheet model, thus accounting for the GrIS-atmosphere feedbacks. To evaluate the added value of a coupled RCM-ISM model, we explored the importance of the GrIS-atmosphere feedbacks by comparing the results of the coupled experiment to those coming from two uncoupled experiments, PF (Parameterised elevation feedbacks) and NF (No feedback). We showed that the impact of taking the feedbacks into account increases over time. This study is therefore a necessary step toward a more accurate assessment of the contribution of Greenland to future sea-level rise and of its impact on the climate system. However, future refinements could be envisaged.

One of the main uncertainty in assessing the GrIS contribution to future sea-level rise comes from the climate projections themselves. For example, using five different global climate models to force MAR at its lateral boundaries under RCP8.5 conditions, Fettweis et al. (2013) provide SMB-inferred estimates of this contribution ranging from 4.6 to 13.1 cm in 2100. This range is fully comparable to that reported by Calov et al. (2018) who used MAR simulations (forced by three GCMs chosen from the Fettweis et al. (2013) sample) to force the SICOPOLIS ice sheet model. Whatever the experimental design, the large spread in SLR projections highlights the great uncertainty associated with the choice of the global climate model used to force MAR at its lateral boundaries. It raises the question to what extent the differences between 2W and PF or NF experiments would be amplified (resp. mitigated) with a stronger (resp. weaker) climate forcing than that simulated by MIROC5. Another question concerns the impact of a constant MIROC5 climate used to force MAR beyond 2100. As outlined in Sect. 3, this results in discarding the continued change that the climate will likely undergo beyond 2100 suggesting that our

SLR projections are underestimated. Another consequence is that inter-annual variability is neglected after 2100. This can lead to conservative estimates of the Greenland contribution to sea level rise in the future due to non-linearities of the SMB. On the other hand, the imprint of the 2095 MIROC5 climate may amplify regional changes of the GrIS response. There is therefore a strong need for iterating the present study with different global climate simulations run under an extended RCP8.5 scenario, but also with different regional climate models that may have different sensitivities, to assess more accurately the impact of the different GrIS-atmosphere feedbacks and to better evaluate the uncertainty associated with the projected sea-level rise contribution from the GrIS.

Another limitation is related to the 2000-yr relaxation GRISLI experiment, run at the end of the initialisation procedure to reduce the model drift in terms of ice volume. This produces residual differences with the observed topography (Bamber et al., 2013) used in the MAR simulations. This has important consequences on the MAR simulated climate. In particular, the steeper slopes existing in the GRISLI topography (i.e. $S_{ctrl}$) tend to produce unrealistic katabatic winds. Therefore, we have chosen in this study to use an anomaly method of the surface elevation onto which the SMB and ST fields are downscaled at the 5 km resolution grid (Eq. 4). The objective of this approach was first to maintain the realism of the simulated present-day climate computed on the observed topography (Bamber et al., 2013) and, secondly, to avoid inconsistencies between the climate simulated by MAR and that used to force GRISLI. However, this implies that the forcing climate is not fully consistent with the GRISLI topography. This should be taken into consideration in a future work to improve the quality of our results. As an example, a reasonable compromise to avoid the use of anomaly method would be to use the topography obtained at the end of the iterative initialisation process (rather than $S_{ctrl}$) as initial MAR topography to keep the mismatch with the observed topography as low as possible, and to initialise and perform MAR simulations with this new topography. Moreover, the use of an anomaly method to account for the change in topography is incompatible with a conservative coupling between the ice sheet model and the climate model. This is further amplified by the fact that we use a flux correction outside the present-day ice margin to force ice removal. This methodology has been followed to limit the impact of biases from the atmospheric model and from the initialization procedure, but the imposed ice removal outside the present-day ice mask may bias locally the model response towards increased ice thinning. Since our simulations are run under the RCP8.5 forcing scenario, this has probably a negligible impact because the GrIS is likely to experience a retreat from the present-day ice margin. However further studies with alternative scenarios and/or GCM forcing, and even more paleoclimate studies, should ideally avoid using this kind of flux correction.

In addition, difference of resolution between MAR (25 km) and GRISLI (5 km) can cause artefacts in the results, especially at the edges of the ice sheet. Indeed, in the corresponding MAR grid cells, a fraction of permanent ice cover may coexist with a non-zero fraction of tundra. Since the surface elevation changes computed in MAR from the aggregated GRISLI topography are weighted as a function of the fraction of the different surface areas, they may be underestimated as tundra soil type is not subject to any change in altitude. This artefact has been illustrated in Sect. 4.2.1 with the example of the behaviour of katabatic winds that are artificially reduced in our simulation at the ice sheet margin. Moreover, since the margin regions are those experiencing the strongest changes in altitude, they are also the most sensitive to climate change. As a consequence, an improper estimation of the topography changes may induce improper SMB changes. This underlines the need for increasing

the atmospheric model resolution as far as possible to avoid such artefacts and to better represent the fine scale atmospheric-topography feedbacks impacting the SMB. Indeed, higher spatial resolution could resolve finer scale ice sheet dynamics to better represent the ice flow in outlet glacier or better represent fine scale atmospheric-topography feedbacks impacting the SMB in these regions. However, a compromise must be reached between the additional computing resources and the required degree of accuracy of sea-level projections.

Regarding the ice sheet model, a 5 km horizontal resolution does not permit to capture the complex ice flow patterns of smallest outlet glaciers, whose characteristic length scale can be less than 1 km (Aschwanden et al., 2016) and to quantify accurately the ice discharge at the marine front. This may have large implications in the sea-level rise estimates. Using a 3D ice sheet model with prescribed outlet glacier retreat, Goelzer et al. (2013) found an additional SLR contribution from outlet glaciers of 0.8 to 1.8 cm in 2100 and 1.3 to 3.8 cm in 2200, with the influence of their dynamics on SLR projections decreasing with time and with the increasing importance of the atmospheric forcing. This is in line with the fact that ice flow act to counteract ice loss from surface melting (see Sect. 4.2), as previously outlined by several authors (Huybrechts and de Wolde, 1999; Goelzer et al., 2013; Edwards et al., 2014a). However, despite the possible decreasing influence of marine terminating glaciers, at the centennial time scale, it is essential to evaluate more accurately the impact of ice dynamics and to better capture the complex geometry of fjords surrounding the marine-terminating glaciers.

There is a growing number of evidence for attributing the acceleration of outlet glaciers to the intrusion of warm waters from adjacent oceans in the fjord systems or in the cavity of floating ice tongues (e.g., Johnson et al., 2011; Straneo et al., 2012; Rignot et al., 2015) that can destabilise the glacier front and/or favour the ice-shelf breakup (Gagliardini et al., 2010), decreasing thereby the buttressing effect and increasing the ice calving. In turn, the released freshwater flux in ocean may impact sea-surface temperatures, oceanic circulation and sea-ice cover. Moreover, atmosphere-ocean feedbacks also have an impact on the GrIS. As an example, Fettweis et al. (2013) showed that the disappearance of Arctic sea ice in summer induced by ocean warming enhances surface melting in northern Greenland through a decrease of surface albedo and the subsequent atmospheric warming. Thus, the absence of the oceanic component in our modelling setup appears as a limiting factor, although, the direct impact of ocean via sub-shelf melt at the ice sheet margin will likely be limited in the future as a result of inland retreat of GrIS.

Our initialisation method adjusts the basal drag coefficient in such a way that the departure between the observed and the initial GRISLI topographies is reduced. The resulting $\beta$ coefficient is spatially varying but is taken constant in time. This assumption may likely be valid for short-term forward simulations but is probably overly simplistic. On the one hand, the basal drag tends to be smaller towards the margins with respect to the interior. As the ice sheet retreats inland, it can be expected a reduction in basal drag for a specific location, due for example to a decreasing effective pressure. On the other hand, changing basal hydrological conditions can also alter the basal drag. This can occur as a result of rainfall or surface meltwater infiltration that can refreeze at depth or propagate all the way to the bottom of the ice sheet and increase basal lubrication (Kulessa et al., 2017). Therefore, a time constant basal drag coefficient inferred under present-day conditions may underestimate the ice flow acceleration. A few models describing the vertical inflow exist (e.g., Clason et al., 2015; Banwell et al., 2016; Koziol et al., 2017) but are generally run at the regional scale and at very high spatial resolution (a few tens to a few hundreds of meters at

most). Implementing such models in large-scale ice sheet models is currently outside the realm of possibilities. However, as there is a growing interest in performing ice sheet projections over multi-centennial time scale, large-scale ice sheet models would undoubtedly benefit from the implementation of simplified infiltration schemes (e.g., Goelzer et al., 2013) so as to account for the impact of ongoing changes in surface meltwater on ice flow.

An additional limitation related to the choice of our spin-up procedure is that the glacial-interglacial signature of past climatic changes is ignored. Neglecting the climate history of the Greenland ice sheet implies too warm ice temperatures. This may have an impact on the future GrIS evolution and on its contribution to sea-level rise. Indeed, the basal drag coefficient inferred from the inverse method may be too high so as to compensate the errors induced by the artificial warming. However, using a higher-order ice flow model, Seroussi et al. (2013) showed that at the centennial time scale the basal conditions and the
GrIS projections are only poorly sensitive to the initial vertical temperature profile but are critically dependent on atmospheric conditions.

Despite these limitations, the sea-level projections performed with GRISLI compare well with those conducted with more sophisticated ice sheet models (Edwards et al., 2014b), and the simulated surface ice velocities present a good agreement with the observed ones (Fig. S11). It appears thus as a good numerical tool to be coupled with a regional climate model with a
reasonably good representation of the ice dynamics and limited computational resources.

## 6    Conclusions

This study is based on the first regional atmospheric – ice sheet coupled model allowing the GrIS-atmosphere feedbacks to be accounted for. Using this new model, we investigated the GrIS evolution and its contribution to sea-level rise from 2000 to 2150 under a prolonged RCP8.5 scenario (2W experiment). The importance of the GrIS-atmosphere feedbacks has been
assessed through the comparison of the two-way coupled experiment with two other simulations based on simpler coupling strategies: the NF experiment in which the MAR outputs are directly used as GRISLI forcing and the PF experiment in which the elevation feedbacks are parameterized. In both NF and PF experiments, changes in topography simulated by GRISLI are not updated in the atmospheric model. The main conclusions drawn from this study are the following:

– Accounting for the GrIS-atmosphere feedbacks amplifies the ice mass loss and changes in ice sheet geometry with
25        increased surface slopes from the central regions to margin areas and consequences on the Greenland ice velocities.

– The effect of accounting for the feedbacks between GrIS and the atmosphere increases with time and becomes significant at the end of the $21^{st}$ century, as illustrated by the 2W-NF difference in GrIS contribution to sea-level rise in 2150, i.e. 1.9 cm, against 0.3 cm in 2100.

– Accounting for the parameterized elevation feedbacks in the PF experiment leads to an additional SLR contribution of
30        $\sim$7.6 % in 2150 compared to NF. On the other hand, the parametrization used in PF allows to reduce the mismatch (in terms of SLR projections) between the one-way and the two-way coupled approaches by 73.7 % in 2150, showing that at this time scale, changes in ice sheet geometry appear to be dominated by the SMB-elevation feedback.

– Finally, with our modelling setup, we showed that estimating the GrIS contribution over a fixed ice sheet mask (as in PF and NF experiments) overestimates the SLR contribution by $\sim$6 %, suggesting that most of RCM-based studies have probably overestimated the ice loss computed from changes in SMB.

## 7 Data availability

The model output from the simulations described in this paper are freely available from the authors without conditions. The source code of MAR version 3.7 is available on the MAR website: http://mar.cnrs.fr. The GRISLI source code are hosted at https://forge.ipsl.jussieu.fr/grisli, but are not publicly available due to copyright restrictions. Access can be granted on demand by request to Christophe Dumas (christophe.dumas@lsce.ipsl.fr).

*Author contributions.* The implementation of the three coupling methods as well as the simulations were done by X. Fettweis and C. Wyard. S Le clec'h, S. Charbit and A. Quiquet analysed the results and wrote the manuscript with contributions from M. Kageyama, C. Dumas and X. Fettweis. The GRISLI model was developed by C. Ritz.

*Competing interests.* The authors declare that they have no conflict of interest.

*Acknowledgements.* The authors are very grateful to J. Fyke and two anonymous reviewers for their numerous and fruitful comments that helped to improve the manuscript. S. Le clec'h, M. Kageyama, S. Charbit and C. Dumas acknowledge the financial support from the French-Swedish GIWA project and the ANR AC-AHC2, as well as the CEA for the S. Le clec'h PhD funding. A Quiquet is funded by the European Research Council grant ACCLIMATE no 339108. Computational resources (MAR and GRISLI) have been provided by the Consortium des Équipements de Calcul Intensif (CÉCI), funded by the Fonds de la Recherche Scientifique de Belgique (F.R.S.–FNRS) under grant no. 2.5020.11 and the Tier-1 supercomputer (Zenobe) of the Fédération Wallonie Bruxelles infrastructure funded by the Walloon Region under the grant agreement no. 1117545.

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

**Table 1.** Greenland ice sheet contribution (in cm) computed from ice thickness variations simulated with the 2W, NF and PF experiments.in 2050, 2100 and 2150 relative to 2000(first three lines) and fromintegrated NF SMB values cumulated over the entire GrIS defined by a fixed ice sheet mask (SMB$_{MSK_{NF}}$) and by the time-evolving ice sheet mask (SMB$_{MSK_{2W}}$) simulated in the 2W experiment (last two lines).

|  |  | GrIS contribution to SLR (in cm) | | |
| --- | --- | --- | --- | --- |
|  | Name of experiment | 2050 | 2100 | 2150 |
| From ice thickness | 2W | 0.7 | 7.9 | 20.4 |
|  | NF | 0.7 | 7.6 | 18.5 |
|  | PF | 0.7 | 7.9 | 19.9 |
| From SMB | SMB$_{MSK_{NF}}$ | - | 9.1 | 22.1 |
|  | SMB$_{MSK_{2W}}$ | - | 9.0 | 20.8 |

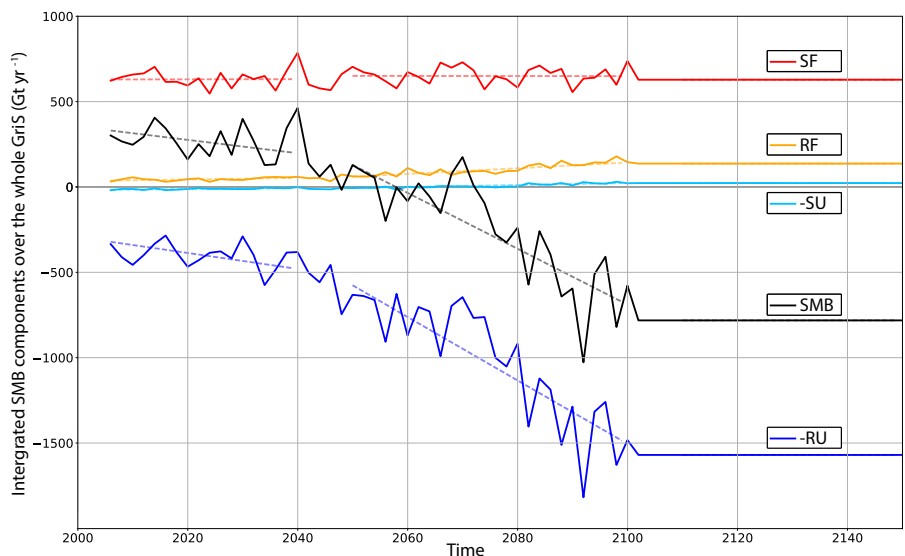

**Figure 1.** Evolution of the SMB (black line) and its components from 2005 to 2150 (in Gt yr$^{-1}$) simulated by MAR in the 2W experiment and integrated over the ice sheet mask taken from Bamber et al. (2013). The SMB components are: snowfall (SF, red line), rainfall (RF, orange line), sublimation (-SU, light blue line), runoff (-RU, dark blue line). Dashed lines correspond to regression lines ranging from 2000 to 2039 and from 2041 to 2099. The light dark solid line corresponds to the zero line.

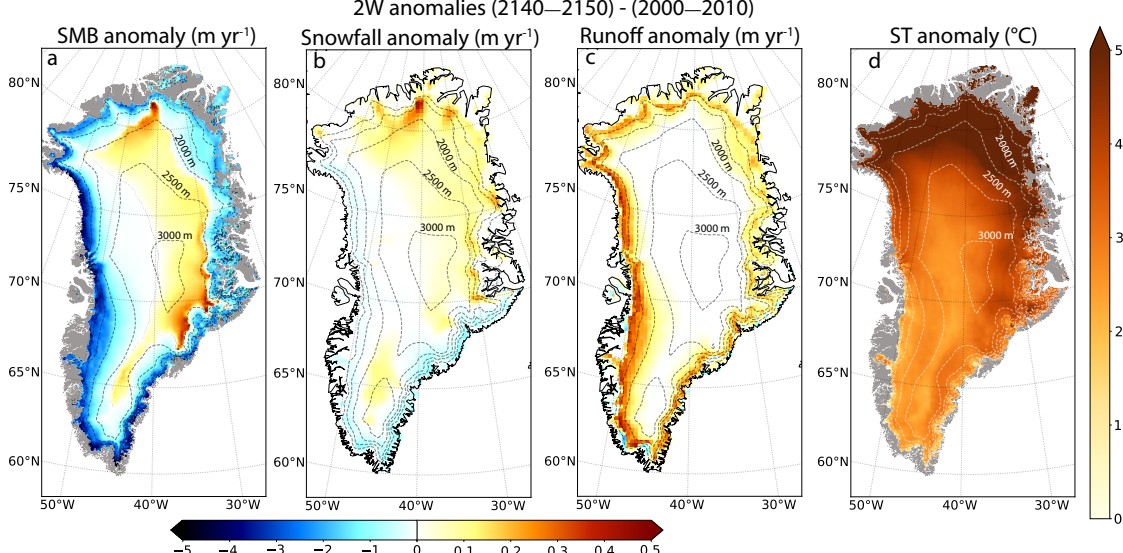

**Figure 2.** Anomalies of (a) mean annual surface mass balance in m yr$^{-1}$ (b) annual snowfall in m yr$^{-1}$ (c) annual runoff in m yr$^{-1}$ and (d) mean annual surface temperature (in °C). These anomalies are given between the last 2140—2150 and the first 2000—2010 ten years of the 2W experiment. (a) and (d) are computed on the GRISLI grid and (b) and (c) are given on the MAR grid. The dashed lines correspond to the 500 m surface elevation iso-contours for the present-day observed topography. The grey shade represents the non ice-covered areas. Note for (a), (b) and (c) that the colour scale is not symmetric for positive and negative values.

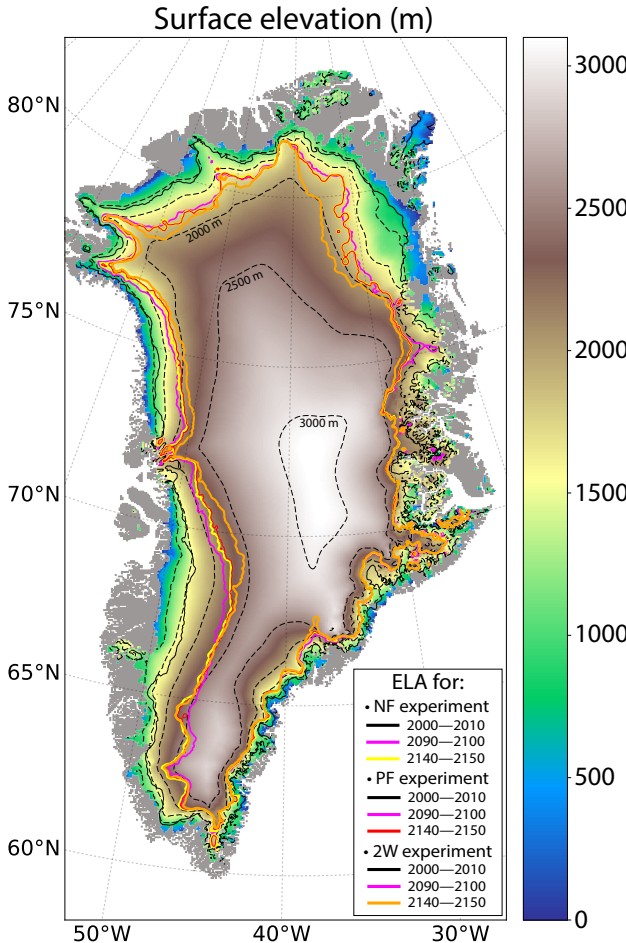

**Figure 3.** GrIS surface elevation in 2150 simulated in the 2W experiment (in m). The solid black and purple lines represent the equilibrium line altitudes (ELA, limit between the accumulation and the ablation zones) in the 2000—2010 and 2090—2100 mean periods respectively for the NF, PF and 2W experiments. The yellow, red and orange solid lines indicate the ELA position over 2140—2150 for the NF, PF and 2W experiments respectively. The dashed lines correspond to the 500 m surface elevation iso-contours for the present-day observed topography. The grey shade represents the non ice-covered areas.

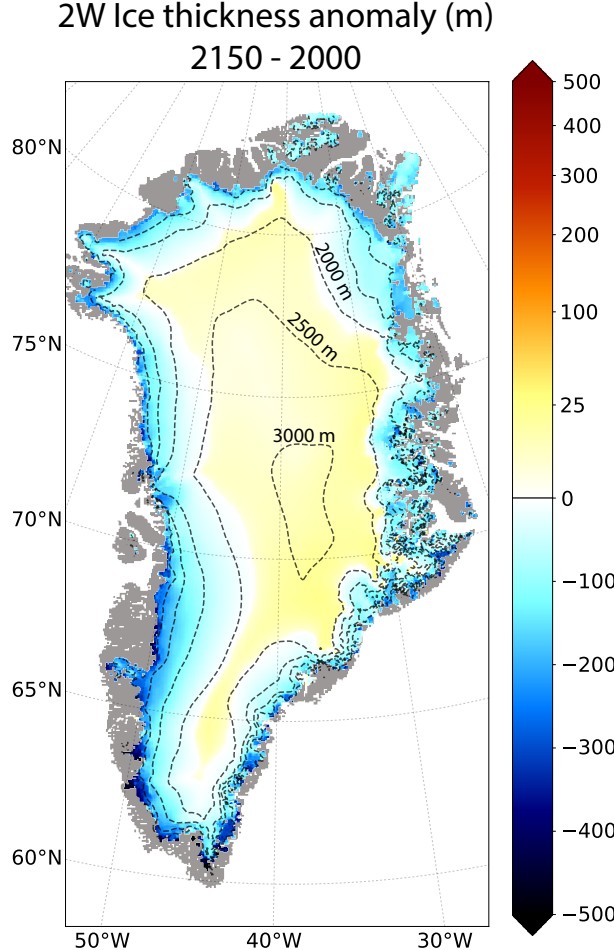

**Figure 4.** Ice thickness anomaly (2150-2000) simulated in the 2W experiment (in m). The dashed lines correspond to the 500 m surface elevation iso-contours for the present-day observed topography. The grey shade represents the non ice-covered areas. A non-linear color scale is used for positive values.

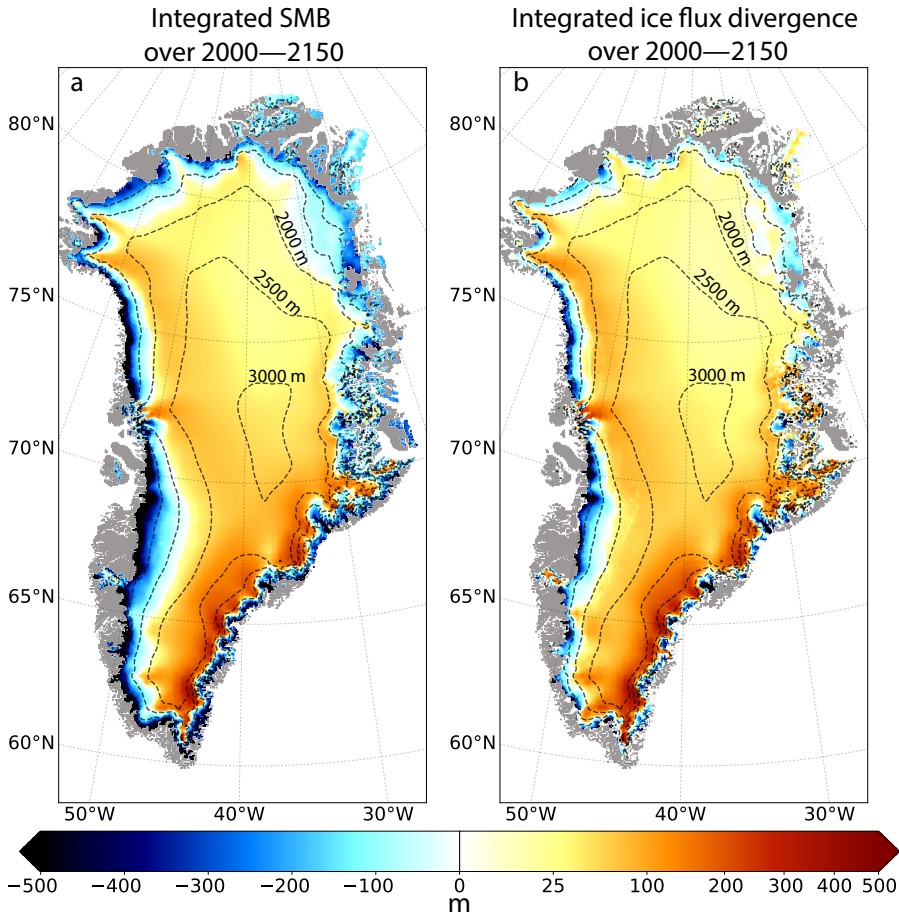

**Figure 5.** Cumulated SMB (a) and ice flux divergence (b) throughout the entire 2W experiment (from 2000 to 2150) given in meters and computed on the GRISLI grid. The dashed lines correspond to the 500 m surface elevation iso-contours for the present-day observed topography. The grey shade represents the non ice-covered areas. A non-linear color scale is used for positive values.

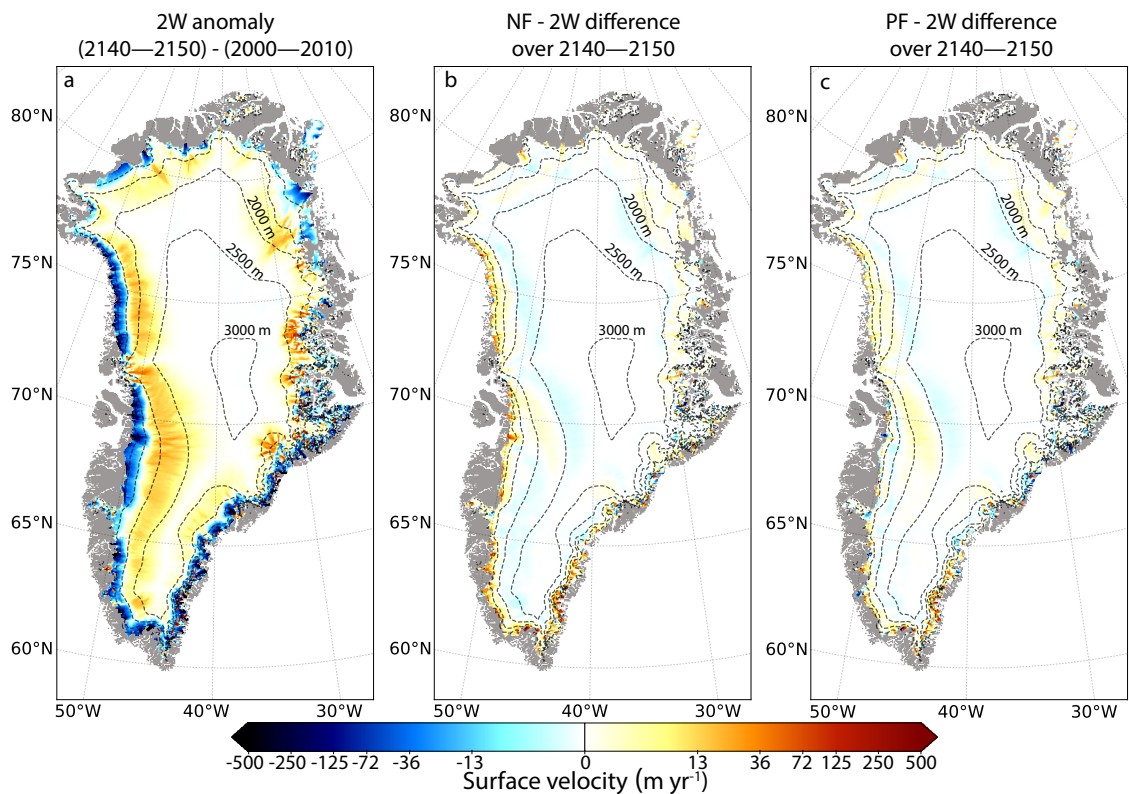

**Figure 6.** (a) Mean surface velocity anomaly (in m yr$^{-1}$) between the 2140—2150 and the 2000—2010 mean periods for the 2W experiment. (b) Mean surface velocity difference between the NF and the 2W experiments for the 2140—2150 mean period.(c) Same as (b) for the PF and 2W experiments. These differences are computed on the GRISLI grid. The dashed lines correspond to the 500 m surface elevation iso-contours for the present-day observed topography. The grey shade represents the non ice-covered areas. A non-linear color scale is used for both positive and negative values.

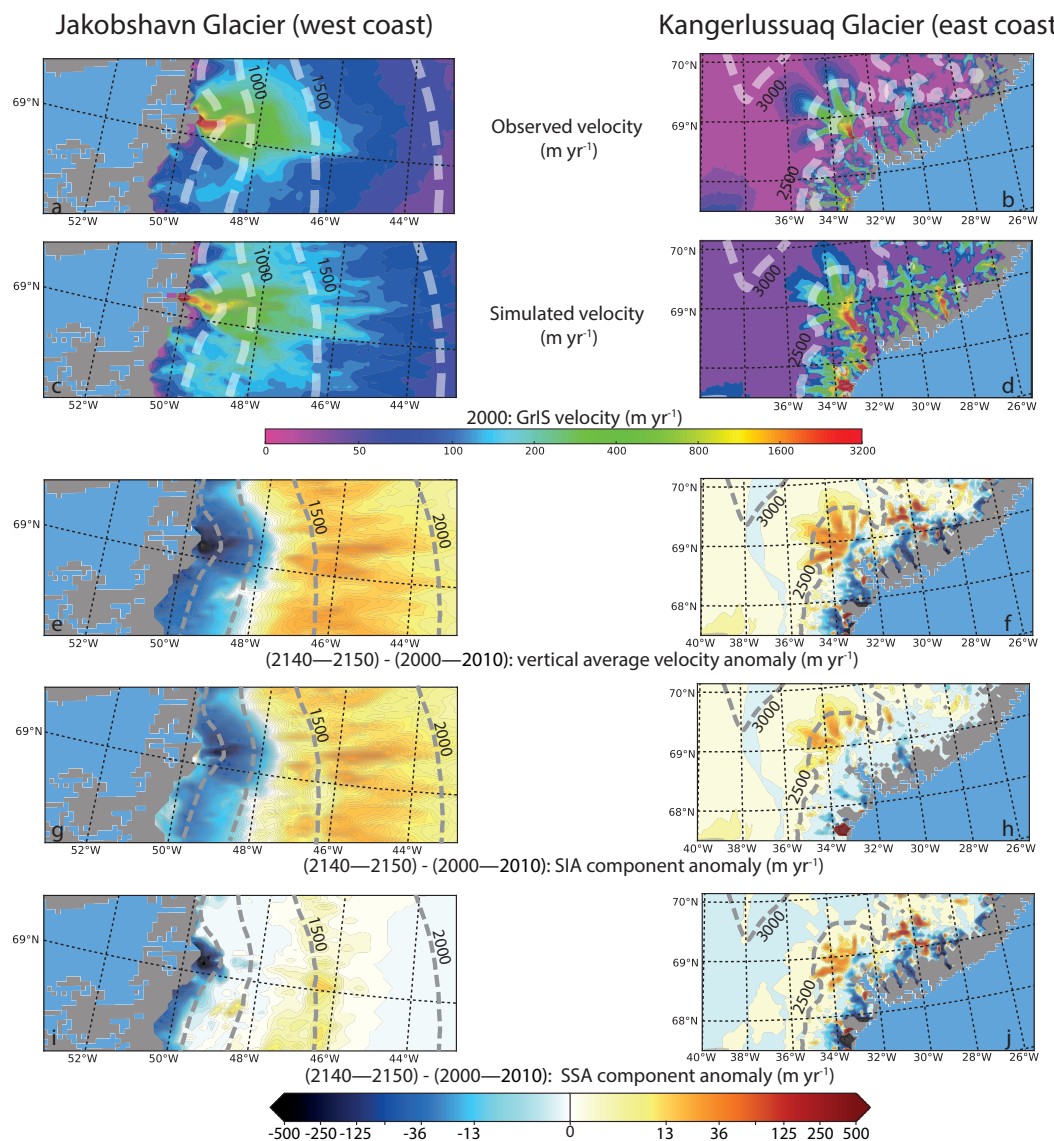

**Figure 7.** Regional zoom over the Jakobshavn (left panels) and the Kangerlussuaq (right panels) glaciers for the 2W experiment. (a) and (b) are the observed velocities (in m yr$^{-1}$) from Joughin et al. (2018). (c) and (d) are the simulated velocity (in m yr$^{-1}$) after the initialization procedure. Panels from (e) to (j) represent the velocity anomalies (in m yr$^{-1}$) between the 2140—2150 and the 2000—2010 mean periods of the 2W experiment for the vertically averaged surface velocity (e and f), the SIA velocity component (g and h) and the SSA velocity component (i and j). Note that a logarithmic scale is used for the velocity anomalies. The dashed lines correspond to the 500 m surface elevation iso-contours for the present-day observed topography. The grey shade represents the non ice-covered areas and the blue shade is the ocean mask.

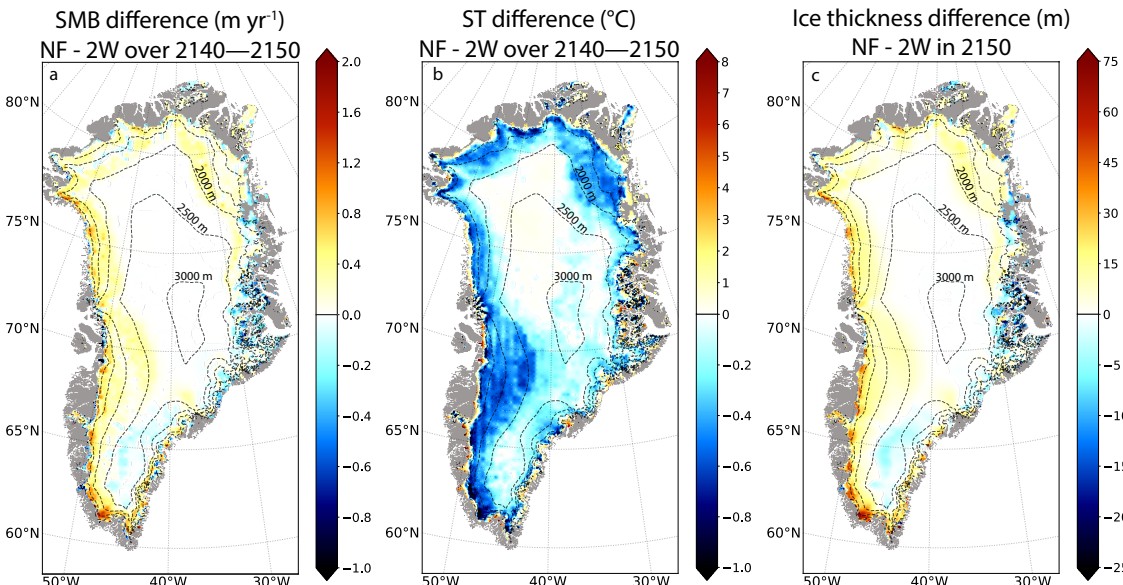

**Figure 8.** Mean differences (NF-2W) for the 2140—2150 mean period (a) Annual surface mass balance (in m yr$^{-1}$), (b) Annual surface temperature (in °C), (c) Ice thickness (in m) in 2150 between NF and 2W. Note that these differences are computed on the GRISLI grid. The dashed lines correspond to the 500 m surface elevation iso-contours of the present-day observed topography. The grey shade represents the non ice-covered areas.

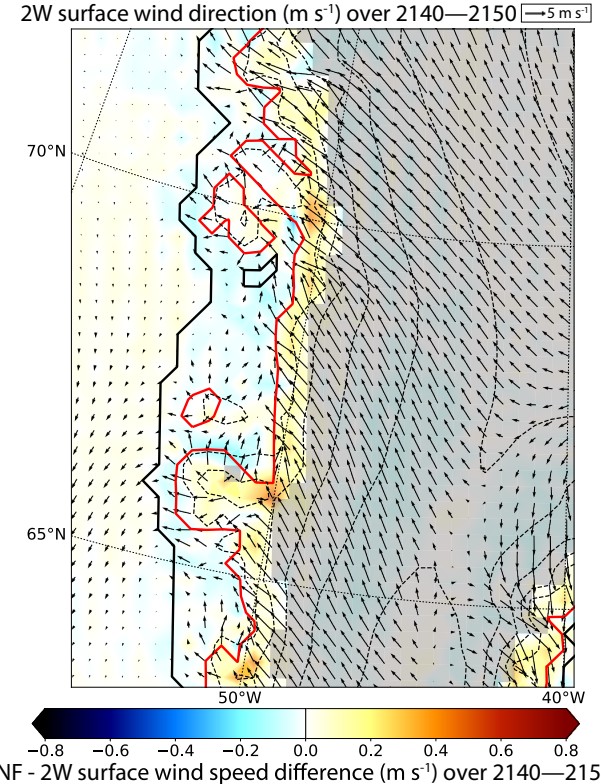

**Figure 9.** Surface wind speed difference (shaded) between the NF and the 2W experiments for the 2140—2150 mean period. Black arrows represent the wind direction in the 2140—2150 mean period of the 2W experiment. The length of the arrows indicate the magnitude of wind speed. The grey shaded area stands for the extent of the region for which the permanent ice fraction is 100 % (no tundra). Red solid line indicate the ice sheet extent The dashed black lines correspond to the 500 m surface elevation iso-contours.

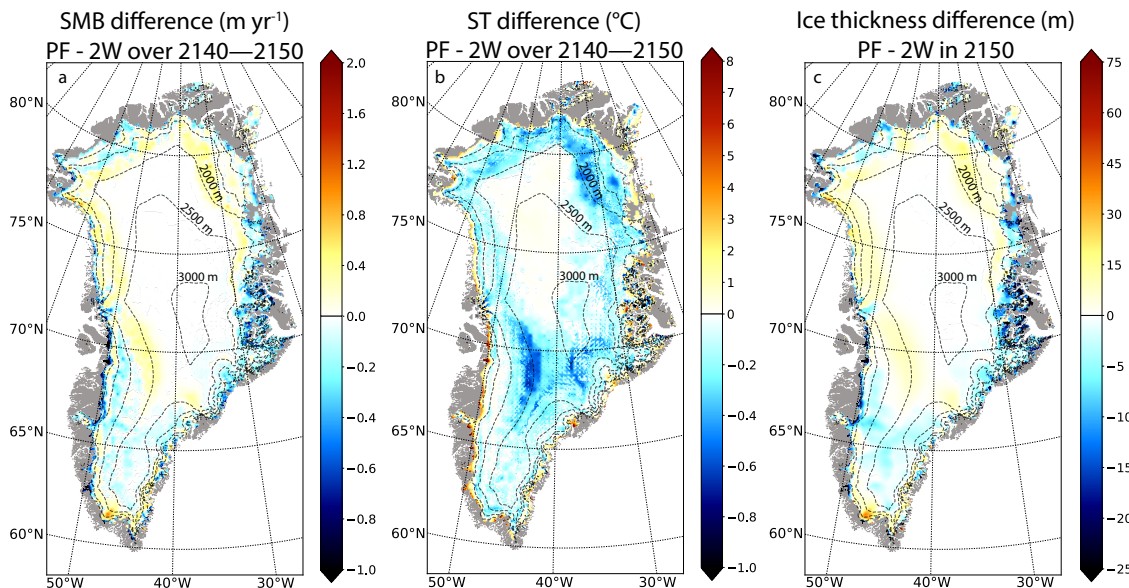

**Figure 10.** Mean differences (PF-2W) for the 2140—2150 mean period (a) Annual surface mass balance (in m yr$^{-1}$) and (b) Annual surface temperature (in °C) (c) ice thickness (in m) in 2150 between PF and 2W. Note that these differences are computed on the GRISLI grid. The dashed lines correspond to the 500 m surface elevation iso-contours of the present-day observed topography. The grey shade represents the non ice-covered areas.

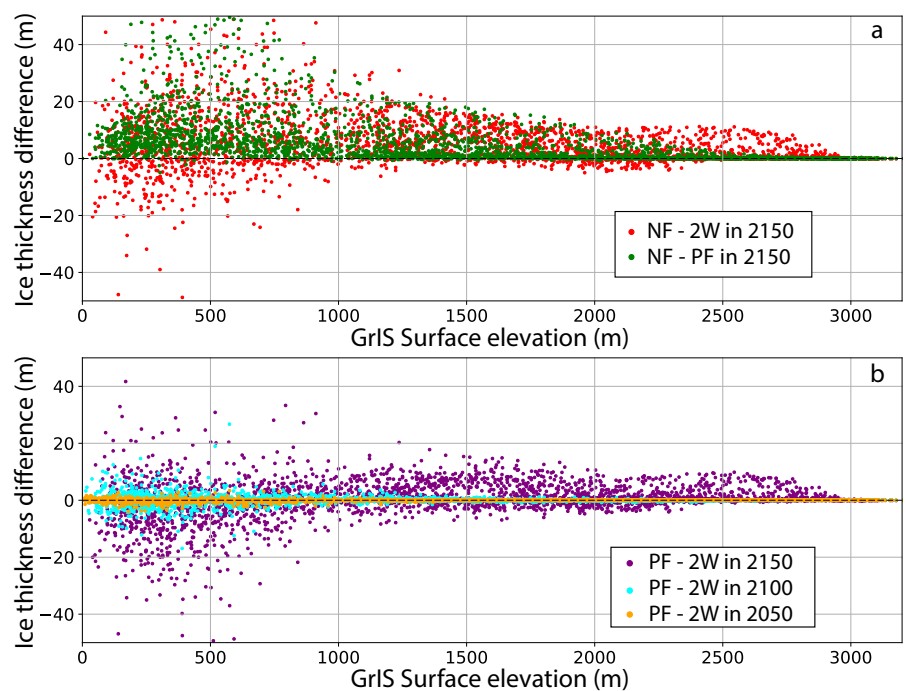

**Figure 11.** Ice thickness difference (m) as a function of the GrIS surface elevation (m) for the three coupling experiments. In (a) the red and green dots represent the NF-2W and the NF-2W differences in 2150; (b) Ice thickness difference (PF-2W) in 2050 (purple dots), 2100 (light blue dots) and 2150 (orange dots) mean periods.

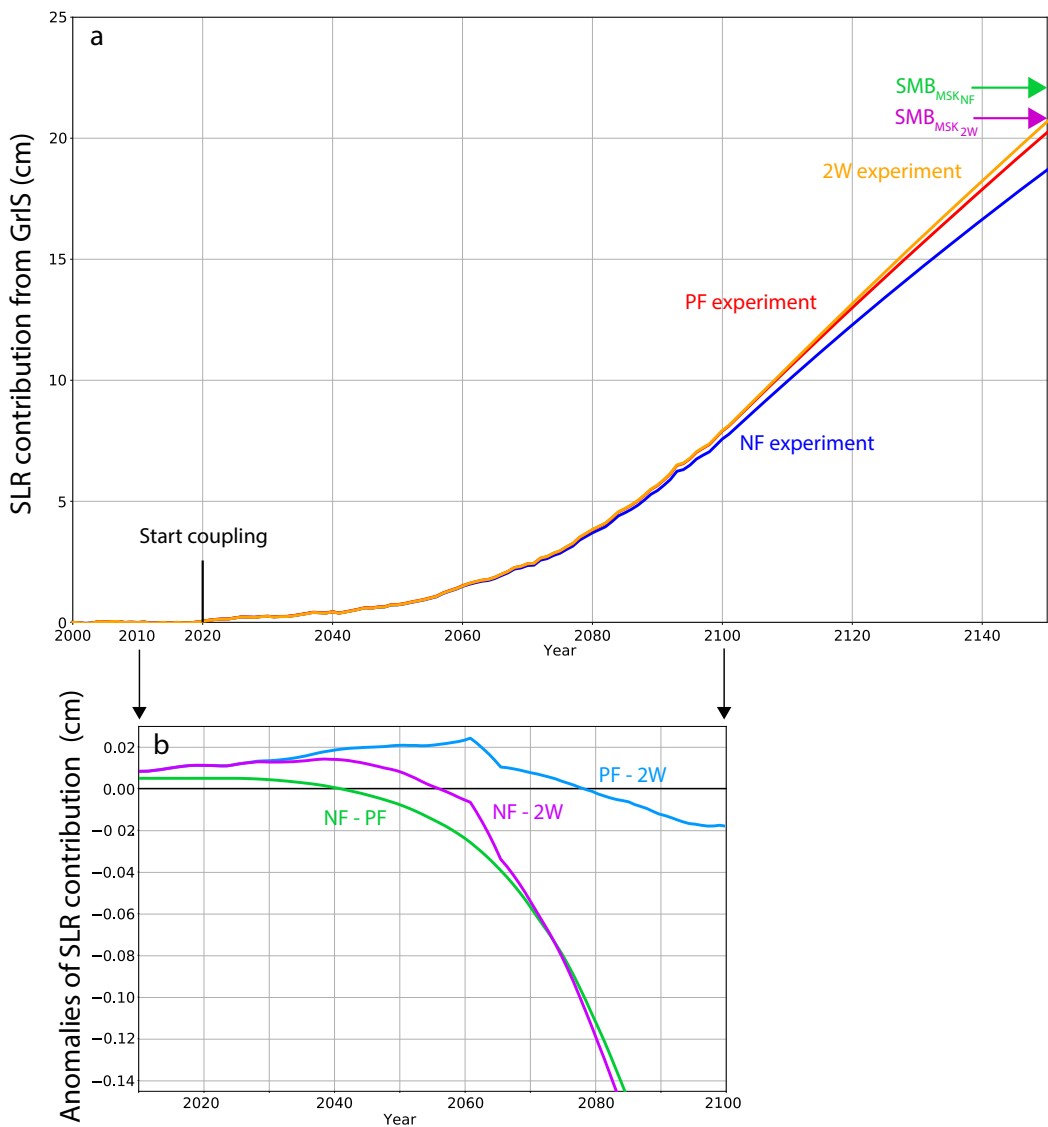

**Figure 12.** (a) Contribution of the GrIS to sea-level rise (in cm) as simulated in the NF (blue line), PF (red line) and 2W (yellow line) experiments and inferred from the ice thickness changes between the 2000 and 2150. Green and purple arrows indicate the projected contribution from GrIS inferred from SMB changes integrated over the same period over a fixed ice sheet mask ($SMB_{MSK_{NF}}$) and a time variable ice sheet mask ($SMB_{MSK_{2W}}$). (b) Zoom of the differences of GrIS contributions to sea-level rise between the PF and the NF experiments (green line), between the 2W and the NF experiments (purple line) and between the 2W and the PF experiments (light blue line).