# Peer review of "Assessment of the Greenland ice sheet - atmosphere feedbacks for the next century with a regional atmospheric model coupled to an ice sheet model"

_The Cryosphere, 2017_

## Referee Comment (RC1) · Dr. Fyke (Referee) · 22 Nov 2017

Review of Leclerc et al 2017:

Le clec'h et al present a study that assesses the strength of Greenland ice-sheet-atmosphere feedbacks over the 21st century using a regional model that is coupled to an ice sheet model. I think this is a novel experiment and valuable study and has the potential to be cited extensively as ice sheets are increasingly incorporated into various climate model architectures. My suggestions for improvement, listed in 'order

of appearance', are below. My primary general concerns, which I hope the authors can address adequately, involve some apparent inconsistencies in the coupling/spin-up (e.g. use of topography anomalies, and uncertainty on how land surface types change in response to ice retreat, and what happens if the ice sheet wants to expand beyond present-day margins). Finally, please feel free to counter my suggestions if you think I'm in error.

**Comments**

P1L1: "the projected Greenland sea level rise contribution is mainly controlled by the interactions between the Greenland ice sheet (GrIS) and the atmosphere": while I tend to agree, relevant models can't yet fully assess the ocean contribution, so I think this statement is overconfident. Please moderate.

P1L2: "in particular through the temperature and surface mass balance – elevation feedback": no, the atmospherically-driven GrIS SLR contribution is controlled by radiative excess/warming. Feedbacks reinforce this effect but is do not control it.

P1L2: "fine scale processes"->"fine scale dynamical processes" ?

P1L15: "Furthermore, in 2150, using a fix ice sheet mask, as in the no coupling method, overestimates by 24 % the SLR contribution from SMB compared to the use of the ice sheet mask as simulated in the two-way method" this seems counter to the previous statement that SLR from two-way coupling is 9.3% larger than the uncoupled case. Is the difference due to dynamic discharge term?

P2L 4: "The atmospheric conditions control the variability" -> "Atmospheric conditions control variability and change"

P2L7: "SMB directly affect the GrIS total ice mass by impacting its characteristics such as thickness, ice volume and ice extent" - this can occur both directly and via impacts on ice dynamics. Explicitly state the latter (dynamics) for clarity.

P2L9: there are more foundational references regarding the dynamical GrIS impact

on atmospheric flow. Suggest to use these in addition/instead. As just one arbitrary example: http://onlinelibrary.wiley.com/doi/10.1034/j.1600-0870.1996.00014.x/abstract

P2L11: "different processes and feedbacks"->"different processes and feedbacks that regulate transient ice sheet change"

P2L16: "The climate models usually represent" -> "For example, CMIP5 climate models unanimously represented"

P2L24: Suggest citing recent Lofverstrom et al. discussion study on resolution dependence of ice sheet conditions in GCMs: https://www.the-cryosphere-discuss.net/tc-2017-235/

P2L35: "the authors only consider a strict linear relationship between topography and SMB changes" - please note more clearly either here or in next paragraph why this is a handicap to these methods, leading to why your approach is better

P2L9: "The second fundamental requirement is to represent the ice sheet topography changes in the atmospheric model by using an ISM instead of the fixed geometry usually used" This sentence is tautological since by definition a fixed geometry will not capture topography changes. Reword sentence.

Throughout text: "developped" -> "developed"

P4L7: 16 km high, from surface? Sea level?

P4L12: "hydrological cycle" -> "atmospheric hydrological cycle" ?

How does Crocus differ/integrate with SISVAT? Please clarify.

In the case where the ice sheet expands or contracts, how is under-snow (or snow free) ice sheet surface exchanged for bare land surface (or vice versa)?

P4L20: "The topography of the GrIS as well as the surface types (ocean, tundra and permanent ice) are provided by Bamber et al. (2013)" -> clarify this is for the NC

experiment (presumably)

P5L10: "we have repeated the MIROC5 year 2095 (representative of the years 2090s) for 50 additional years" - this repetition is certainly not representative of this time period due to lack of continued change, and also lack of internal variability. While I don't think this is a fatal flaw of the study, the authors should clearly note this caveat here and later during discussion of results, so readers clearly realize the effects of this artificial 'extension' (probably, fairly strongly reduced overall change, making the results presented here conservative).

P6L11: Why is the annual mean bottom snowpack temperature not used as the boundary condition for the ISM instead?

P6L19: also just due to the long timescale of ice sheet responses?

P7L1: what is meant by 'vertical fields'? Please clarify.

Spin-up procedure: How does this procedure deal with ice growth outside the observed ice sheet extent? Figure 2 suggests this ice is simply removed? If so, how does this effective strong artificial sink of ice impact all subsequent sensitivity experiments? Please explain the impacts of this clearly in the text, if this is the case.

P8L21: why not simply start the coupling at 2005 (i.e. the end point of the 1976-2005 initialization/spin-up period)?

P9L13: The use of topography anomalies is concerning since it implies the SMB/ST field received by GRISLI is inconsistent with GRISLI's height (for example, the ELA on the GRISLI grid would exist at a different elevation than if the GRISLI elevation was directly used). Can the authors comment on why this approach does not introduce problems with their experimental design? As it stands, this is not justified adequately. An alternate approach that would have avoided this problem would have been to use the spun-up GRISLI topography as the 'fixed' topography instead of the Bamber topography.

[Figure]

Figure 2 and other figures: 5 years is likely not long enough to generate robust climatologies. Suggest using at least 10 years instead.

P11L10: the finding of very strong marginal cooling due to increased katabatics is very interesting and pertinent, and deserves a further explaining. It would be very useful if the authors plotted overlaid near-surface wind anomaly vectors plus ST changes in a 'zoomed-in' plot of a good illustrative portion of the margin.

Similar to above point: it would be excellent to see a quiver plot of wind anomalies over the entire ice sheet, given their importance. Also would it be possible to visualize the increased mixing in the boundary layer, leading to warming in the 2-W coupled case?

P11L23: do authors mean "Following the increase of the ST"?

P11L25: ", there is a decrease of 112 Gt yr$-$1 25 of ice " -> "112 Gt/yr extra ice ablates"

P11L30: "14 % larger in 2-W" - can an estimate be made of the uncertainty in this value (and others) due to interannual variability? Put another way, can the authors confirm that the changes they see are significant in the face of background noise in ablation area (for example)?

P12L5: "lower surface temperature over these regions" - suggest reinforcing to readers once more here that this is *relative* to the NC experiment.

P12L8/9: what does the +/- indicate here?

P12L13: "become ice or snow-free or snow free, exhibiting bare ice " this is confusing. What happens if the entire GRISLI ice column disappears? Does tundra emerge?

P12L25: Previous studies have highlighted a strong decrease in ice discharge across outlet glacier grounding lines as a consequence of increased surface melting. E.g. Gillet-Chaulet 2012, Goelzer 2013 and others. Is this same effect seen here?

P12L25: Is it completely correct to say the entire SLR contribution is caused by the 'melting contribution'?

P12L25: Can the authors quantify the reduction in marine margin extent in 2-W?

SP13L1: "This higher integrated SMB, obtained when using no updated ice sheet mask" - do the authors mean "lower"..? This sentence seems to directly contradict the previous sentence. If I'm mistaken here, a clearer description of the processes here is needed.

General: The authors should consider quantifying actual feedback factors associated with the inclusion of elevation feedbacks (see Roe 2009, Reviews of Geophysics). This would be a good benchmark number to produce, for other works to compare to.

P14L11: "As for the ISM, increasing the grid resolution of MAR" - do you mean "as for the regional climate model"..?

P14L35: "…underestimated by simulating." Unclear.

P15L1: "surface albedo and strength of katabatic winds." -> "surface albedo and strength of katabatic winds, with a demonstrably strong return influence on SMB"

P15L27: "optimal resolution of the ice sheet and the atmospheric model, for ISM-RCM coupling." While an interesting-sounding statement, I find it also a bit vague: by optimal, do the authors mean something like "of high enough respective resolutions to resolve both important atmospheric and important ice sheet dynamical processes"?

P15L30: "The next step of this study…" as described, this is extremely ambitious, with many challenges that outstrip the effort to implement atmospheric coupling. If it is truly a planned next step; great! But if not, I'd suggest not claiming to plan to do this.

General: while the writing is 100% understandable and clear, a final proof-read by a native English speaker would be useful as a final stage, if possible, to clear up remaining small grammar issues.

---

## Referee Comment (RC2) · Anonymous Referee #2 · 12 Dec 2017

**1   General Comments**

The paper claims to be focused on assessment of the future of the GrIS through 2150. But in fact, it seems more focused on assessment of a new technique for RCM-Ice Model coupling. Throughout the paper, focuse shifts back and forth between the two. The experiment is to run a future simulation of the GrIS using MAR coupled with GRISLI in three different ways, and then compare/analyze the results. from each other.

[Figure]

The coupling method is interesting, but the GrIS is more interesting. I believe the paper would be better if it would keep its focus firmly on the GrIS, while keeping the methods separate. I ultimately want to know, what do we learn about Greenland? Unfortunately, the figures do not really support that. Figures 1-4 do in a way; but the rest of the figures only really tell us about technical differences between coupling technique.

The experiments in the paper show that the different coupling techniques provide different answers. Unfortunately, it is hard to know which answers are closer to the truth, because there are no controls. I came into this believing that the most sophisticated copupler would produce the most melt and also be more accurate; but I had no proof on the accuracy part. This paper has reinforced my prior assumptions, without providing any additional evidence on accuracy. I am therefore hard pressed to say what it has added to my understanding of coupling technique.

I did learn some things about the future Greenland itself, in spite of the figures not really helping with this. I learned:

1. Expect a steeper slope and stronger katabaic winds, in addition to the expected smaller ice sheet. This will result in colder (not warmer) temperatures near the coast.

2. In parts of Greenland, the ELA could be as high as 3000m by the year 2150. I find that idea astounding, at 77 degrees North latitude. Some discussion of this result would be really interesting.

3. Expected sea level rise contribution of Greenland in 150 years is 20cm; and the rate of melting will be continuing to rise at that point.

4. Ice loss and SMB are highly correllated over the next 150 years; so much so that plots of the two look highly similar. Unfortunately, the paper does not try to quantify the correllation.

For the record, here's what I learned about coupling techniques:

1. Integrating SMB over a fixed ice mask over time is a poor way to calculate total SLR

contribution, due to the changing ice mask.

2. The 2w case melts more than the 1w or NC case in the RCP8.5 scenario.

3. Full Stokes solvers might yield better results.

Overall... I think this paper has done some interesting modelling runs, but so far has mostly failed to draw interesting conclusions from those runs, and to focus the reader's attention on those conclusions. I would suggest the authors think through the question "What have we learned about Greenland;" and then re-do the figures and commentary to support that learning, and focus the reader's attention on it. The paper will also need significant disucssion of these Greenland results, in comparison with other papers that have looked at the future of Greenland; for example, Vizcaino et al 2015. Especially interesting would be places where this paper predicts something DIFFERENT from those other papers, and why? In this way, the reader needs to be drawn to focus on the most interesting things — the surprises! — first, without having to dig for them.

Once the paper has focused primarily on Greenland, I would then think about how to add discussion of a new coupling technique, without taking away from the main scientific focus of the paper. But in the absence of any solid provable way to prove that one coupling technique is better than another, I would avoid making too many claims about the 2w coupling; just that you think it is better, and it certainly melts more ice. In the parts (bulk) of the paper focused on Greenland, I would use whatever coupling technique you think is most realistic.

A secondary issue: the paper reports many numbers, and only a few of them have error bars. Where did those error bars come from, and why are error bars not reported for other numbers? Would it be possible to get error bars for other numbers?

**2 Specific Comments**

p.2l.24: Studies by Vizcaino et al (and also at GISS; see Fischer Nowicki 2014) use elevation classes to develop an SMB. Elevation classes are mathematically equivalent to custom-designed gridcells that follow elevation contours. They are therefore able to offer high resolution in the direction of the slope gradient, while continuing with low resolution perpindicular to the gradient.

p.8 l.25: I have traditionally used different labels for the different coupling strategies described. Your "NC", I have traditionally called "1-way coupling." Your "2w", I would call "serial 1w coupling". Your "1w coupling," I would call "corrected 1w coupling." Given the differences in terminology, it's probably best to describe what each of your schemes is (which you do), but don't assume that others would use the same names. BTW, none of the coupling schemes here conserve energy, in the sense that two-way couplers (say) between the ocean and atmosphere typically do conserve energy. Therefore, I would be reluctant to call any of them true "two-way coupling."

p.9 l.7: Why is the 2w scheme more expensive? I see that you have to run the GCM and ice model together, rather than separately. But is any more expense actually involved?

p.9 l.21: Fig. 1 does not support the text. Now I see Fig. 1 is reporting anomalies; but I think it would be more interesting (and no less informative) if it would report actualy Temperature.

p. 10 l.2: Cause-and-effect is backwards. Actually, the lower SMB is the CAUSE of the ELA shift.

p. 10 section 4.1.2: This is the one section of the paper with error bars. How were those error bars computed, it didn't say? Unfortunately, some of the values reported are not statistically significant; and many others are barely. A more clear way to report the reports in this section would be something like "we saw no statistically significant change in the GRISLI ice sheet in the years 2000-2050." This conclusion is already

pretty apparent in the figure: the "interesting stuff" happens further out in time, especially with the more advanced coupling.

p.10 l.12-24: This looks like an explanation for the increased slope; but I'm not following it.

p. 10: In general, please report ice loss in dual units: both Gt, and mm of sea level rise. If this were done consistently, then section 4.2.3 would barely be needed.

Secion 4.2: Now, the paper stops telling us about Greenland, and analyzes minute differences between the coupling techniques Not so interesting.

p.11 l.20: The word "probably" is used. This indicates a hypothesis; how can that hypothesis be tested?

p.12 section 4.2.2: Ice thickness and SMB maps are highly correllated throughout this paper. For that reason, section 4.2.2 says pretty much the same thing as section 4.2.1. It would be better to (a) talk about the correlation explicitly, even quantify it, and then (b) keep ice thickness and SMB together in one section every time it is discussed in the results.

p.12 l.30: I appreciate that doing wrong calculations will give the wrong answer. I'm glad that you are not doing that. But is this worth half a section to explain? It seems you are going out of your way because someone else did something fishy.

p. 13 l.2: the last sentence of this paragraph is the most important. Don't "bury the leded"... put it up at the front.

p.14 l.15: I don't believe this argument on ice-ocean feedback. We know that tidewater glaciers retreat VERY quickly once they become imbalanced. How many tidewater glaciers will be left for us to simulate in the year 2050, 2100 or 2150? And what about going beyond that — when the REALLY interesting things start to happen? I just don't believe that ocean coupling is very important for GrIS.

** Any idea what happens beyond the year 2150? I know it's outside the scope of this paper. But this paper opens up more tantalizing questions by simulating a non-steady-state process just a little bit of the way — to a point where the changes are continuing to accelerate. What does this simulation look like in 500 years? 1000 years? 5000 years? How important are the feedbacks on that timescale?

Fig 6A: Why is there a vertical-stripe pattern in western Greenland? That makes me suspicious of the model. Please explain...

** Figures: Please make sure of the following in figures:

a) Avoid the rainbow color scale in most cases (Fig 4). There are better choices.

b) If you do use the rainbow, avoid splitting green at zero (Fig 4A). One figure has green fo both positive and negative numbers; not cool.

c) Avoid a color scale that's read on one end and violet on the other; because then the smallest and largest values look almost the same.

d) When using color scales with red on one end and blue on the other, make sure that red always corresponds to places that are melting / getting warmer / losing mass; and blue corresponds to the opposite. Reverse the color scale if needed, in order to keep this consistent.

e) The figures in this paper all use different color scales and conventions, for no apparent reason. It looks like they don't belong together. Please make them more uniform, unless there's a good reason for the difference.

f) Please put a title on top of every plot, in font large enough to read. Make sure that every plot has units on every axis (either the color scale, or the x-y axis. Most fonts on most figures need to be larger.

**3 Review Questions**

**Does the paper address relevant scientific questions within the scope of TC?**

Yes
**Does the paper present novel concepts, ideas, tools, or data?**

Yes

**Are substantial conclusions reached?**

The could be, but they are not currently explained well.

**Are the scientific methods and assumptions valid and clearly outlined?**

Partly. Problems: (a) no control on the novel methods, and (b) some hypotheses thrown out there without even a suggestion on how they would be tested.

**Are the results sufficient to support the interpretations and conclusions?**

Partly. I would start with writing more interesting and specific conclusions; and then working harder to support them based on the experiments, as well as comparisons to similar conclusions of other studies.

**Is the description of experiments and calculations sufficiently complete and precise to allow their reproduction by fellow scientists (traceability of results)?**

yes

**Do the authors give proper credit to related work and clearly indicate their own new/original contribution?**

yes

**Does the title clearly reflect the contents of the paper?**

no. The title claims to be about GrIS; but the paper is more titlted toward method comparison

**Does the abstract provide a concise and complete summary?**

The first half is clear and concise. The second half gives too much detail for an abstract; it would be easiesr to read if the main qualitative results were stated, without quantitative detail (which the reader can find by flipping into the results / conclusion sections).

**Is the overall presentation well structured and clear?**

no

**Is the language fluent and precise?**

yes

**Are mathematical formulae, symbols, abbreviations, and units correctly defined and used?**

mostly. Figures need better unit labelling. There are two formulas, neither of which is necessary. Formula 1 (p. 5) is just one formula of how a dynamic ice model works; why was this one just taken out of context and placed here in a generally non-formula paper? I don't think it adds much. Formula 2 (p. 8) would be more clear if it were just described in words.

**Should any parts of the paper (text, formulae, figures, tables) be clarified, reduced, combined, or eliminated?**

yes. Avoid repetitive sections analyzing first SMB and then ice sheet thickness (which is highly correllated to SMB). Re-do figures to tell us more about Greenland itself, rather than the difference in coupling methods. Talk more about the key interesting points, and less about minor details of differences between coupling methods.

**Are the number and quality of references appropriate?**

yes

**Is the amount and quality of supplementary material appropriate?**

yes

Please also note the supplement to this comment:
https://www.the-cryosphere-discuss.net/tc-2017-230/tc-2017-230-RC2-
supplement.pdf

**Supplement:**

[revised manuscript text omitted]

*[handwritten margin notes: "We call this one-way coupling", "I would call this 'corrected 1w'"]*

[Figure]

where $H_{Bamber}$ is the Bamber et al. (2013) topography at 5 km and $\Delta H_{GRISLI}$ is the topography anomaly simulated by GRISLI between the initial topography computed for year 2000 from the equilibrium state (t=0) and the ongoing time step (t). In doing so, this method artificially accounts for the elevation feedback because the SMB and ST are initially computed by MAR on a fixed ice sheet topography. With this method GRISLI is forced off-line by the MAR atmospheric conditions already computed in the NC run, therefore allowing sensitivity experiments in GRISLI with limited additional computer time. However, the changes in GRISLI topography are not taken into account by MAR.

*why? Not really...*

– The Fully Coupled method (hereafter 2-W). This coupling method is the most accurate way to represent the interactions between the GrIS and the atmosphere but it is also more computationally expensive. At the end of a MAR simulated year, MAR is paused and GRISLI is forced by the 5 km interpolated SMB and ST just computed by MAR. GRISLI then computes a new GrIS topography and extent which are aggregated on to the 25 km MAR grid for the simulation of the next year of the MAR experiment. GRISLI and MAR are never stopped, just alternatively paused and resumed until 2150. *↳ This coupling does not conserve energy,*

The differences between the GRISLI equilibrium state after the initialisation step and the observed topography (Bamber et al., 2013) (cf Sect. 2.2.2) could lead to inconsistencies between the results obtained by MAR under its usual setup, i.e. calibrated with the Bamber et al. (2013) topography, and the results that would be obtained by using directly the GRISLI topography. For this reason, in both the 2-W and the 1-W experiments, we use anomalies of GrIS topography applied on Bamber et al. (2013) topography rather than the absolute topography from GRISLI.

**4 Results**

**4.1 The uncoupled simulation: the NC experiment**

**4.1.1 MAR**

*for the GrIS*

*Error bars?*

*Looks like -1C on Fig. 1?*

The mean ST over the first two decades (2000-2020)  -18.7°C and the mean SMB *annual* is 434 $Gt\,yr^{-1}$ (Fig. 1). After 2020, the ST increases by 0.065°C $yr^{-1}$ until 2100 (Fig. 1). Over the same period (2020-2100), the averaged GrIS SMB decrease by 12.3 Gt $yr^{-1}$ (Fig. 1). The surface temperature anomaly in 2100 compared to 2000 shows a warming ranging from +1.5°C in the southern part of the GrIS to more than +8 ° C in the northern part (Fig. 2A). This warming in northern Greenland is a direct response to the MIROC5 forcing fields due to the polar temperature amplification. However, regional heterogeneities are observed in the annual mean GrIS SMB spatial distribution (Fig. 2B). Indeed, between 2000 and 2100 there is a positive SMB anomaly (i.e more ice accumulation) in a zone located along a South-North transect in the central part of the GrIS. This ice accumulation is mainly governed by the larger snowfall on the GrIS central part in winter and spring seasons (not shown). An opposite trend 10 times larger than ice accumulation, is simulated over the edges of the ice sheet, with a negative SMB (i.e ice ablation). In these regions, the summer season governs this negative SMB and is characterised by larger rainfall and melting ice (meltwater and runoff increase) than for other seasons (not shown). As a

consequence, the limit between the accumulation (SMB > 0) and ablation (SMB < 0) areas, also called the equilibrium line altitude (ELA, a line where SMB = 0), shifts inland through time (Fig. 3). This shift explains the mean decrease of SMB over the whole GrIS until 2100 seen in Fig. 1.

No- The shift is a consequence of lower SMB

**4.1.2  GRISLI *(uncoupled)* *[handwritten: not significant]**

*[handwritten: How?]*

[revised manuscript text omitted]

*Is this surface T or ΔT anomaly as compared w/ base?*

Put titles
on top

Color
Scale 3
backwards

[Figure]

**Figure 2.** Mean anomalies between the last five years (2095-2100) and the first five years (2000-2005) of the No coupling experiment: (a) Annual surface temperature anomalies °C; (b) Annual surface mass balance anomalies in Gt yr$^{-1}$; (c) Surface elevation anomalies in m. The positive scales for the b and c are 10 times lower than the negative scales.

Are any of these anomalies
greater than total ice in
that grid cell?

[Figure]

*Title?*

*What is the significance of an ELA at 3000m?*

*Units?*

**Figure 3.** Mean GrIS surface elevation for the last five years (2145-2150) of the 2-W experiment. The solid black line represents the equilibrium line altitude (limit between accumulation and ablation zone) in 2000 for the NC, 1-W and 2-W experiments. The straight and dashed colour lines represent respectively the ELA for the periods 2145-2150 and 2095-2100 for: in blue for the NC experiment ; in red for the 1-W experiment; in orange for the 2-W experiment. The ELA of the 2095-2100 (dashed lines) of NC, 1-W and generally 2-W are superimposed.

*Conclusion: No=1W, 2W melts more.*

*Do we get something like Fig. 2 for 1w & 2w cases?*

[Figure]

**Figure 4.** Mean surface velocity anomalies in m yr$^{-1}$: (a) between the last five years (2095-2100) and the first five year (2000-2005) of the NC experiment; (b) between the last five years (2145-2150) of the 2-W experiment and the last five years (2145-2150) of the NC experiment.

*(handwritten annotations: "(Title?)", "(Title?)", "Reverse color scale ↓", "temperature")*

**Figure 6.** Mean anomalies between the last five years (2145-2150) of the 2-W experiments and the last five years (2145-2150) of the NC experiments: (a) Annual surface temperature anomalies in ° C; (b) Annual surface mass balance anomalies in Gt yr$^{-1}$; (c) Surface elevation anomalies in m.

*(handwritten annotations:)*
Why the N-S striping artifacts?

Are those MAR or GRISLI plots?

Overall: what is correlation between SMB anomaly & elevation anomaly? It seems quite high.

[Figure]

[Figure]

[Figure]

**Figure 7.** Contribution of the Greenland ice sheet volume contributions to sea level rise (cm) compared to the year 2000 for the three experiments : - Blue line: anomaly between 2-W experiment and no coupling experiment; - red line: anomaly between 1-W experiment and no coupling experiment (NC); - Orange line: anomaly between 2-W coupling experiment and 1-W coupling experiment.

---

## Referee Comment (RC3) · Anonymous Referee #3 · 12 Dec 2017

Review of "Assessment of the Greenland ice sheet - atmosphere feedbacks for the next century with a regional atmospheric model fully coupled to an ice sheet model" by Sebastien Le clec'h and others.

Summary ——-

The two-way coupling between a regional climate model and an ice sheet model is an important development that marks a clear step forward to improve the projections of the future contribution of the Greenland ice sheet to sea-level change. The manuscript

compares results of the two-way coupling to former methods of representing the interactions between ice sheet and atmosphere and comes to important conclusions concerning the errors implicit to those simpler approaches. The manuscript is of clear interest to the readers of The Cryosphere but still needs to be substantially improved before being acceptable for publication. I recommend major revisions along the comments outlined below.

General comments ——————-

The language of the manuscript needs substantial improvement, because many formulations give rise to misinterpretations of the scientific content. While many mistakes could clearly be avoided with a better command of the English language, a large number of typographical errors and mistakes in the referencing suggest that the authors could have made a better effort to deliver a readable manuscript for the review process.

The text itself reveals that the models are not actually fully coupled (use of anomaly method) and also gives indications why a full coupling is so much more difficult to achieve. I suggest to adjust the title and modify other occurrences of "fully coupled" in the text to "two-way" coupled to take this into account. A discussion item on this point and next steps that need to follow to work towards truly full coupling between RCMs and ISMs should be included.

The ice sheet initialisation procedure is somewhat non-standard and therefore requires a much better explanation. As it is heavily based on former work that is in part not well documented, an additional effort is required to describe the method in a way reproducible for other modellers. Finally, the evaluation of the method appears to be based on an experiment that is not closely related to the model state actually used for the projections, which may be possible to resolve with an additional control experiment.

The thermodynamic aspect of the model is not well represented, arguably because it plays a minor role for the present work. Nevertheless, substantial computing time

is spent during initialisation to equilibrate the temperature and the role of bottom and surface boundary conditions is mentioned. Therefore, the model description in 2.2 requires at least a short description of this model component.

The experiment names are not specific enough and should be improved. For my understanding, what is presented as the method "no coupling" is in fact a one-way coupling, where the ice sheet is responding to changes computed by the RCM (with "no feedback"). 2-W is correctly described, but 1-W is somewhere between one-way and two-way coupling because it parameterises the feedback. Maybe you could use "no feedback", "parameterised feedback" and "two-way" instead.

The most important question in the comparison of results after 100 years and after 150 years is left open: why does the behaviour of 2-W suddenly change around 2010. For this it may be instructive to also look more detailed at around 2060, where a similar shift is possibly visible. Otherwise, I find the comparison redundant because the bottom line in most cases is 'like after 100 years, only stronger'.

The integration of SMB anomalies already discussed in the manuscript could be added as an additional experiment, possible even two, if masking would be additionally taken into account. This would facilitate the comparison and place the discussion of the effect of masking on firmer ground.

Please also see corresponding specific comments for where these issues appear in the text.

Specific comments —————

Title I would argue that the models are not "fully", but rather "two-way" coupled because an intermediate down-scaling step is necessary and, more importantly, an anomaly method is used.

P1.L1 Better "the projected sea-level contribution from the *Greenland ice sheet*". Also mention a typical time scale here to make clear this is about the centennial time-scale.

P1.L2 Be more precise about the mechanisms and feedback(s). The next sentence ("these feedback*s*") suggests that "temperature and surface mass balance – elevation feedback" refers to at least two feedbacks. What are these precisely? "surface mass balance – elevation feedback" is clear, but what is the role of temperature? Note also that melting is clearly related to temperature increase, but the SMB is ultimately controlled by the energy balance.

P1.L5 A bit confusing to mention start date as 2020. It is understood later that before 2020 elevation changes are considered too small to make a difference. But at this place it may be better to give the period of the entire simulation (2006 - 2150). Note also that the RCP is not defined beyond 2100, so it is better to mention "prolonged RCP 8.5 scenario".

P1.L5 It seems confusing to call this simple method "no coupling", since it represents a one-way coupling. See also general comment on naming the experiments.

P1.L6 Could mention that this one-way coupling methods attempts to incorporate or parametrise two-way interaction. It represents an intermediate method between one-way and two-way coupling. See also general comment on naming the experiments.

P1.L7 I suggest to omit "offline". The correction may be offline to MAR, but it is online to the ice sheet model, as the correction is updated every time step and dependent on the current ice sheet elevation. Could add what is happening with the extent, since it has been explicitly mentioned for the former method.

P1.L9 Clearer to replace "ice sheet elevation feedback" by "surface mass balance – elevation feedback".

P1.L9 Maybe ", the one-way and two-way coupling methods ..." since the amplification occurs in both cases.

P1.L11 Some ice sheet margins are not in the coastal region. Replace by "ice sheet margins" or similar. This should be followed throughout the document for other occurrences.

P1.L15 "52 400 kmˆ2 smaller"

P1.L16 "fixed ice sheet mask"

P1.L20 "always" is only true for the end of the simulation. In the first decades or so the volume loss difference cannot be significant. Maybe give an estimate for a time scale where this is true similar to the comparison one-way vs. two-way.

P2.L6 "the ablation" (singular) <–> "are processes" (plural). Reformulate

P2.L6 Some risk for confusion here. It is a bit simplistic for a paper discussing an RCM as an important component to reduce the interaction to changes in SMB and temperature. It is understood that these are the two variables used to force the ice sheet model, but a bit more detail is required. How does the change of ice extent change the albedo and therefore the energy balance? Does temperature enter the correction method and how? OK, accumulation and ablation are sensitive to ST, but why and how? Also, what is the role of ST other than its influence on SMB, as boundary condition to ice thermodynamics? Does it have an impact on the simulations at all (I don't expect it, but would be good to say something about why not and being able to exclude it).

P2.L6 Maybe already intended, but make really clear that the changes in ST have no direct effect on thickness volume and extent. Reformulate.

P2.L9 Replace "disrupt" by "modify"

P2.L19 More detail needed. Amplification of mass loss by what process under what forcing and compared to what other (control) experiment?

P2.L22 The beginning of this sentence suggests (and I agree) that increased resolution would help to improve the modelling compared to observations, while "more detailed physics" is at least for the ice sheet model typically associated with 'less approximation',

i.e higher order physics. Could you add some detail to distinguish these.

P2.L26 Should introduce RCMs and add references to MAR, RACMO, HIRHAM ... already here, as that is the obvious choice to increased resolution. Introducing the Franco and Edwards methods is already a step further as it is based on RCM output.

P3.L3 Sentence misses references for examples of RCMs.

P3.L9 Specify again for what it is a requirement.

P3.L10 Reformulate "usually used" to "typically used" or similar.

P3.L11 Add reference to Goelzer et al. 2017 here, since it is specifically on GrIS models.

P3.L18 Remove "high resolution" or specify explicitly at what resolution GRISLI is run.

P3.L21 "two-way"

P3.L25 I would consider the three methods part of the experimental setup and therefore name initialisation and experimental setup first.

P4.L4 "developed". Correct also throughout the manuscript.

P4.L4 "SISVAT" requires a reference and description of the acronym.

P4.L16 "ice albedo that has been improved by parametrising the impact of melt ponds on the albedo."

P4.L19 Replace "provided by" by "taken from"

P4.L20 "forced with 6-hourly atmospheric fields". See also P8.L6

P4.L24 Remove "forcing"

P4.L27 Suggest reformulation to "... because it has been shown by Fettweis et al. (2013), to be the best choice from the CMIP5 data-base to reproduce the present-day climate compared to results of MAR forced by reanalyses."

P5.L1 Heading "Climate model initialisation and experiment"

P5.L2 What is the difference between "spurious drifts" and "unwanted trends" or are they one and the same? Reformulate.

P5.L3 Replace "SISVAT requires more than 6 years", by "SISVAT requires less than 7 years" to make clear that the chosen 7-year period is long enough. Or otherwise explain why 7 years is considered OK.

P5.L5 Replace "provided by" by "taken from". Add explanation how the data was interpolated to the coarse MAR grid.

P5.L5 Be consistent in if SISVAT is written in italic or not.

P5.L6 Replace "following year 1976" by "from 1977 onward".

P5.L10 Need to explain in more detail why 2095 can be considered representative for the 2090s. Is it e.g. the year that is closest to the decadal mean? Are trends so linear that the middle of the decade are representative for the average? Typically one would use the decadal mean to represent the long-term average and not one individual year, unless it doesn't matter for some reason.

P5.L12 Better to omit "coupled" here, since it is not clear what is coupled to what and it is further detailed later.

P5.L15 "the northern hemisphere ice sheet*s* (NH references) and the Greenland ice sheet (GrIS references)". or "the northern hemisphere ice sheet*s* and the Greenland ice sheet (all references)".

P5.L16 "... covering Greenland with ...", since the coverage extends outside of the ice sheet mask. Add information about the vertical.

P5.L17 Need to specify what "hybrid" means.

P5.L19 Need to add explanation on the thermodynamic aspect of the model. See also
general comment.

P5.L27 SIA velocity is even stronger controlled by ice thickness.

P5.L28 "SSA component is mainly controlled by the ice flux" is confusing because ice flux is velocity x ice thickness. Clarify!

P5.L29 "rheologies" is the wrong term here. Maybe "deformation regimes".

P5.L30 Replace "ice melting point" by "pressure melting point"

P6.L3 Replace "floating criterion" by "floatation criterion"

P6.L4 What does "characteristics of the Greenland bedrock" mean? Explain

P6.L5 Does that mean the enhancement factor differs for different regions? Explain.

P6.L6 "ice loading changes"

P6.L7 Add a reference for the used isostatic model.

P6.L8 Add a reference describing the thermodynamic model.

P6.L10 This whole paragraph needs to be reworked. Be more specific. What is considered a boundary condition, what is input data and what is considered a forcing? What variables are concerned for ice flow, ice thermodynamics and isostasy?

P6.L11 Is there are a difference between "The annual mean near surface air temperature" and ST? If yes, explain, if not, use TS instead.

P6.L13 What data are these 'boundary conditions' and which variables are taken from which data set? Surface elevation, bedrock elevation and ice thickness are not boundary conditions to the equations that GRISLI solves in the proper sense. You could call this "input data" instead.

P6.L14 "The climatic forcings". Say what they are! TS and SMB?

P6.L15 If basal drag were a boundary condition, it could hardly be computed. Reformulate to make this clearer.

P6.L18 Heading "Ice sheet model initialisation and experiments"

P6.L19 The motivation is not quite correct. I would argue that to equilibrate the model to a steady state is not a necessity given the approximations, but rather a choice. One could envision a transient spinup as initialisation with the exact same model.

P6.L20 Again, more precision needed. What the ice sheet model equilibrates to is rather the climate forcing held constant for this particular initial steady state experiment.

P6.L20 Replace "sensitivity" by "forced" or "forward".

P6.L20 Reference Le clec'h et al. (in prep) is not in the reference list. If you are referring to the present manuscript, say that instead of using an external reference.

P6.L22 Replace "avoid" by "reduce", since the method is not perfect. Also I'd suggest the formulation ".. reduce an initial adjustment of the model during the first years of the simulation due to factors not related to the climate forcing alone." or similar.

P6.L25 Reformulate "just over the bedrock". Maybe "basal conditions".

P6.L26 If basal conditions are "likely to change in time" your method to define spatially variable but *temporally fixed* basal drag coefficient could never be successful. Should add here that your method assumes them to be constant over the 150 years of your experiment.

P6.L27 Suggest to remove sentence "As a result any error in the basal velocity computation can spread vertically in the ice and generate slowdown or acceleration of ice sheet motion." In its present form this sentence is generally true in any case and doesn't support your chose of assimilation method.

P6.L29 It is not clear to me at what point in the procedure observed velocities are actually used. Which observational data set is used? Reference needed.

[Figure]

P6.L30 "three main steps:" Make a numbered list (possible with lists of sub-steps) to facilitate navigation of the different steps.

P6.L31 Replace "not necessary consistent between them" by "not necessarily mutually consistent".

P6.L32 It looks to me like the first guess of basal drag mentioned here is a very good first guess and further adjustment of the basal drag coefficient is very much based on it. At any rate, a full description of the procedure used to arrive at that stage should be included, otherwise the method is not reproducible with another model (and not even with GRISLI itself). See also general comment on initialisation.

P6.L32 Edwards et al is a multi-model intercomparison and does not give specific details on the assimilation technique for GRISLI. The model reference there is given as Quiquet et al., 2012), which does not provide information on spatially variables tuning. Again, the method to produce the first guess basal drag needs to be made transparent for other modellers to be able to reproduce the results.

P6.L32 "surface and bottom". I think you mean surface elevation and bedrock topography. Be more specific!

P7.L1 "vertical fields" Be more specific!

P7.L2 If I understand correctly, you calculate something here in the first step to be used in the second step. Maybe you should say that. Confusing to mention already here "to have an ice flux as close as possible to observation" when diagnostically calculating something here will not have any influence on the match of the ice flux with observations in this step. This could be mentioned in the second step or as a general motivation for your method before.

P7.L3 Not clear to me how to derive a factor (a/b) from a difference (H1-H0). Please provide an equation or better explanation what the underlying idea is, what is done here, and how it is calculated?

P7.L4 You are mixing topography differences and ice thickness differences. Possibly similar or identical in absence of bedrock adjustment, but is it necessary to distinguish them?

P7.L4 Again "the factor allows to decrease (resp. increase) the surface ice velocity" is confusing, because this is not happening in this first step. Also "If *locally* the topography difference *is* positive ..."

P7.L5 How does deltaH translate into deltaV? How does the new velocity compare to observed surface velocities?

P7.L11 How is the new coefficient calculated? Explain in detail. This is reminiscent of the method of Pollard and DeConto 2012, could you describe the similarities and differences to their approach? Surprisingly your adjustment goes very fast (in total less than 2000 years). This makes me believe that the original basal drag was already a good guess and you only need minor adjustments. Is that correct? How different is the final basal drag field from the initial one? Can we see a figure for this comparison?

P7.L15 Replace "minimum gap" by "error".

P7.L15 You additionally need to convince the reader here that this method is optimal in the parameter choices (adjustment time 20 y, relaxation time 200 years) and to make clear in how far the results are (not) dependent on these choices.

P7.L20 After each step you have "a new set of initial conditions" for the next step. Maybe better to only name the final result of your initialisation your initial state as input for the forward experiments.

P7.L21 After 30 kyr, T is in equilibrium with the climate *and with the fixed geometry*, but not the other way around. In the next step of retuning basal drag, you further evolve the geometry and the ice temperature? Could you quantify, give an estimate how far from equilibrium you are now? Why could you not run (part of the initialisation) with freely evolving temperature? What is your stopping criterium at this point and the

reason for not iterating further?

P7.L27 It is not clear why evaluation of the initial state should be based on an experiment which includes further relaxation steps. The control experiment that offers itself naturally and should be used for that purpose is just running the model after step 3 forward with constant forcing. This would give a good indication of the match with observations (at t=0 or t=25) and the remaining model drift (after 150 years), since this is the model state actually used as initial state for the forward experiments. It anyhow seems strange to impose the observed geometry, when the model has been relaxed to a different geometry in step 3.

P7.L32 It is a bit unusual to specify errors in ice thickness as median values, given that errors locally could be positive or negative. Why not specify the absolute error or root mean squared error augmented with the quantiles given already. A map of the mismatch with observations should be given (possibly in the appendix), but then for the model state after step 3, which is assumed as the initial state for the projections.

P8.L1 The model state that has been compared to other models in the initMIP exercise appears to be different from the state used in the forward experiments, because it includes re-imposing the observed geometry and additional relaxation for 2000 years. This should be made very clear, especially in light of the claim that the model is one of the best in the model comparison. This statement in particular requires further qualification and needs to specify what criteria to consider, since the Goelzer et al paper does not provide any explicit ranking of the models and goes into length about how different criteria for evaluating models are not independent. Please use such community efforts to improve your model, but don't misuse them to gain credibility for your model.

P8.L5 SLR contribution as the most abstract change could be named last.

P8.L16 Is the elevation difference used for the correction calculated between Bamber (at 5 km) and Bamber (at 25 km) bi-linearly interpolated to 5 km? Please describe.

P8.L21 This seems to imply that at least until 2020, NC is an appropriate approximation to the full problem. This should enter the discussion and the abstract, following an earlier comment. Is there any reason why the modification starts at 2020 and not at 2000? It would seem like a cleaner comparison to start the interaction from the moment it is possible (i.e. 2000).

P8.L23 Another "coupling method" that is already discussed in the text and could be formally listed here as well is the one where MAR SMB anomalies alone are used to generate a changing ice sheet geometry (in the absence of an ice sheet model). This experiment can be performed with or without taking into account the surface elevation - SMB feedback and with or without fixed ice sheet extent.

P9.L13 This section reveals that the models are not actually fully coupled and also gives indications why a full coupling is so much more difficult to achieve. See general comment.

P10.L2 The mean decrease in SMB explains the shift in the ELA not the other way around. The ELA is an abstract concept, the SMB change is 'real'.

P10.L5 I am not sure reporting the changes in ice thickness changes as mean and standard deviation makes much sense, given the bipolar nature of thickening in the centre and thinning at the margins. More useful would be for me to describe the changes for specific regions.

P10.L12 What exactly is the impact of ice temperature on ice dynamics? Are you implying that changes of the surface boundary conditions modify the temperature structure of the ice and its deformation?

P10.L13 Are you talking about velocity or velocity anomalies here? Figure 4A shows anomalies! Please clarify.

P10.L15 This statement calls for a figure comparing modelled and observed velocities! Add a panel to substantiate this point.

P10.L18 Add "in this area" after "ice velocities" and remove it in the sentence after.

P10.L30 "amplification of all the changes" is a bit too general here. Better "amplification of the changes".

P10.L31 A figure showing the absolute sea-level changes for the different experiments would be in place, possible as additional panel in figure 7.

P11.L3 ST is already defined

P11.L4 Replace "is strongly colder" by "sees a strong cooling" or similar.

P11.L10 "Thus, the *stronger* ST decrease in 2-W compared to NC ...", assuming there is decrease in both cases. To check also in other places that you discuss differences in changes, not changes itself.

P11.L10 Not sure where "the middle of the slope is". Clarify!

P11.L14 Costal regions don't exist inland from the ice edge.

P11.L28 Replace "SMB anomalies increases by a factor of 10" by "SMB anomalies decreases by 10 cm yr-1"

P12.L1 Again mixing discussion of surface elevation and ice thickness here. Revise.

P12.L1 Add "difference" after "surface elevation change" and reformulate to "follow the patterns of SMB anomaly differences (Fig. 6B)".

P12.L5 Do you mean lower surface temperature in 2-W is the cause for higher SMB and therefore increasing ice thickness, or is the lower surface temperature directly impacting ice thickness (i.e. not through its effect on SMB)? In the first case, lower surface temperature and its effect on SMB should be mentioned first and higher SMB as a consequence. More precision needed here.

P12.L6 "in areas of lower ST"? In my eyes, ST and ST differences (Fig. 6A) are both high and positive in regions of negative thickness anomaly. Clarify that statement.

Interactive
comment

P12.L11 I thought you are trying to describe here the impact on the ice thickness evolution of two-way coupling as opposed to no coupling. In this part, you however come to the impact on the atmospheric circulation (katabatic winds) and land model changes (albedo). From line 15 on, you go back again to ice dynamic changes. Could this material be better organised to avoid jumping between the different aspects? Also, if I understand correctly, the anomalous katabatic winds created by 2-W have visible impact mainly on the narrow marginal areas of the ice sheet where anomalous cooling increases SMB. This should then be counteracted by the albedo changes described L12 and following. It is not really resolved for me how these different factors influence each other and which is the dominant mechanism in which region.

P12.L13 What is the difference between snow-free and snow free?

P12.L22 Melting itself does not necessarily contribute to SLR since melt water can be refrozen in the snow pack. Better replace "melting contribution" by "ice sheet contribution" or similar, also in the rest of the manuscript.

P12.L22 These numbers should be calculated against a control experiment to remove the contribution from remaining model drift. Has this been done?

P12.L23 I would suggest to add a panel to figure 7 with the total contributions for the three experiments and include the integrated SMB mentioned further below in this section.

P12.L25 Since you discuss 2-W against NC, the surface elevation - SMB feedback which operates all over the ice sheet should also be mentioned, not just the processes at the margin.

P12.L26 The difference of 52400 km2 is at the end of the experiment and then it increases with time? Reformulate.

P12.L27 I think all you are saying is that the high resolution ISM mask changes are translated to partial mask changes for MAR. Clarify that the ice sheet mask (Fig S2B)
is the one seen by MAR.

P12.L31 I think the point to make here is not about increase in uncertainty. You can show that when a fixed mask is used, you simply get the wrong result and overestimate the mass loss. Could you quantify the relative importance of this effect compared to the error that is made when not taking into account the surface elevation - SMB feedback?

P13.L3 Please specify the resulting SLR.

P13.L14 This is exactly the reason why median results are not very meaningful in this context. Mean absolute or root mean squared differences are easier to interpret.

P13.L15 After showing figure 8, figure 9 does not add substantial information in my view. I would remove it and continue discussion about differences between 2W and 1W based on figure 8. The only reason to show figure 9 would be if you wanted to attempt modifying the parameterisation used in 1W to incorporate the katabatic wind effect, which could be a logical next step.

P13.L27 These sentences are just stating the obvious. I'd suggest to remove them.

P14.L5 It is not clear to me why a higher resolution should lead to increase the SLR and not the opposite. Unless there are convincing arguments to support that claim, I would leave the sign of the change open. The same applies to the limitation of constant basal drag in the next sentence. With all the complexities surrounding the evolution of the basal conditions over time, I don't think there is any evidence that acceleration of ice flow has to be the dominant response. Again, putting forward some convincing arguments would be appreciated.

P14.L19 Additional limitations that should be discussed: - Ignoring the glacial-interglacial signature of past climate changes in this steady state spip-up of temperature typically makes the ice too warm. This needs to be compensated by other factors (likely a different set of basal drag parameters). - The steady state initialisation also ignores any influence of transients in the observed ice sheet evolution. - Mismatch

of the modelled ice sheet geometry and velocity structure with observations leads to uncertainties in the projected evolution.

P14.L28 Add "in this comparison" after "atmosphere-GrIS feedbacks". I hope you don't think this statement is universally true.

P14.L30 While this statement seems true for the given results, the conclusion hinges on the change in behaviour of 2W at 2110. Unless investigated in more detail, it can not be excluded that such change could happen at an earlier point in time, e.g. for a different model used as boundary condition to MAR.

P15.L5 This comparison is a bit awkward. Wouldn't it be more appropriate to compare the +0.5 cm to the total projected SLR as a relative error?

P15.L14 Remove repeated "respectively" after SLR.

P15.L19 It would be good to additionally put this number (21%) in perspective to the underestimation due to ignoring feedbacks, i.e. the difference between 2W and NC.

P15.L24 Again, I don't see any evidence for the interpretation that higher resolution and higher order physics increase the response.

P15.L29 Replace "disrupt" by "modify"

Tables: ——-

Table 1 Does "after 50 yrs" mean at year 2050? Maybe that would be a better indication. Or do you not want to assign an absolute date to your simulation? The historic and future RCP forcing is clearly linked to an absolute date, though.

Since the ablation area changes so much, it may be interesting to calculate additional diagnostics for a constant region, e.g. for the observed present day ablation zone, or backwards for the area of the ablation zone area after 150 years. This way, the convolution with a changing area could be avoided.

Table 2 Not sure how to interpret a velocity change of e.g. -3.0+-25.0. The noise being much larger than the signal, is the valid interpretation 'no significant' change?

Figures ———-

The labels in the figures are upper case (A,B,C), but the panel references in the captions are all in lower case (a,b,c). Make consistent.

Figure 2 Why are figures B and C so different? At least in the interior, one would expect a pattern very similar to the SMB anomalies in this experiment. My guess is that this is indicative of a remaining model drift. Results of a control experiment starting after step 3 with constant forcing should be shown here or in the appendix and the origin of this difference should be discussed.

Figure 3 The displayed field is ice thickness, not surface elevation as written in the caption. Since the discussion is about ELA and surface elevation - SMB feedback, it may be useful to show surface elevation instead.

Figure 4 The colour scale in A is not easy to read with small positive and small negative values sharing the exact same colour (green). This should be improved. Have you tried to plot velocity ratios instead of anomalies? Since velocity magnitudes cover several orders of magnitude, a large relative change is not visible because of the cutoff at 2 myr-1, while small relative changes at the margin appear exaggerated.

Figure 5 Why not use the same colour mapping here and in figure 4 for the velocity anomalies? That would make it easier to compare the two figures. Caption: "left panel"

Figure 6 There appears to be a slight instability in one or both of the experiments compared in figure 6A. Also Figure 8A shows signs of instability in form of a checker board pattern. While these instabilities are likely not critical for the interpretation of the large scale results presented here, they should at least be mentioned.

Figure 7 Add a panel with absolute contributions of the three experiments. Note that

results shown so far are double differences, i.e. differences in anomalous contributions since year 2000 between different experiments. Could also show sea-level contribution differences calculated from difference to a control experiment with constant forcing to remove the model drift. Same consideration holds for the absolute contributions.

There seems to be a step change around 2060 and again around 2110, where the behaviour of 2w-1w (yellow) changes dramatically. By comparison with 2w-nc it appears to be caused by the evolution of 2W. What is happening at these moments in 2w? Please investigate this further.

Caption: "Differences in Greenland ice sheet sea-level contribution between the different experiments." Then explain how it is calculated.

Figure 9 Figrue 9 is not needed in my estimation.

References:

Format of many references in the text are non-standard. A few examples are given here, but all should be re-checked.

P3.L13 add e.g. before Gagliardini

P3.L19 reformat list of reference and avoid double brackets

P4.L10 "(e.g. Fettweis et al., 2013)

P6.L12 Author is called Fox Maule. Check reference.

P6.L20 Reference Le clec'h et al. (in prep) is not in the reference list.

P8.L2 add Goelzer et al., 2017 to the reference list.

References:

Fox Maule, C., Purucker, M. E., Olsen, N., and Mosegaard, K.: Heat flux anomalies in Antarctica revealed by satellite magnetic data, Science, 309, 464-467, doi: 10.1126/science.1106888, 2005.

Goelzer, H., Nowicki, S., Edwards, T., Beckley, M., Abe-Ouchi, A., Aschwanden, A., Calov, R., Gagliardini, O., Gillet-Chaulet, F., Golledge, N. R., Gregory, J., Greve, R., Humbert, A., Huybrechts, P., Kennedy, J. H., Larour, E., Lipscomb, W. H., Le clec'h, S., Lee, V., Morlighem, M., Pattyn, F., Payne, A. J., Rodehacke, C., Rückamp, M., Saito, F., Schlegel, N., Seroussi, H., Shepherd, A., Sun, S., van de Wal, R., and Ziemen, F. A.: Design and results of the ice sheet model initialisation experiments initMIP-Greenland: an ISMIP6 intercomparison, The Cryosphere Discuss., 2017, 1-42, doi:10.5194/tc-2017-129, 2017.

Pollard, D., and DeConto, R. M.: A simple inverse method for the distribution of basal sliding coefficients under ice sheets, applied to Antarctica, The Cryosphere, 6, 953-971, doi:10.5194/tc-6-953-2012, 2012.

Quiquet, A., Ritz, C., Punge, H. J., and Salas y Mélia, D.: Contribution of Greenland ice sheet melting to sea level rise during the last interglacial period: an approach combining ice sheet modelling and proxy data, Clim. Past. Discuss., 8, 3345-3377, doi:10.5194/cpd-8-3345-2012, 2012.

---

## Author Comment (AC1) · 25 Jul 2018

**We would like to thank the reviewer J. Fyke for the evaluation of our study and the constructive comments that helped us to improve the manuscript. Please find below the reviewer's comments in black font and the author's response in blue font.**

**Responses to J. Fyke (Reviewer 1)**

*Le clec'h et al present a study that assesses the strength of Greenland ice-sheet atmosphere feedbacks over the 21st century using a regional model that is coupled to an ice sheet model. I think this is a novel experiment and valuable study and has the potential to be cited extensively as ice sheets are increasingly incorporated into various climate model architectures. My suggestions for improvement, listed in 'order of appearance', are below. My primary general concerns, which I hope the authors can address adequately, involve some apparent inconsistencies in the coupling/spinup (e.g. use of topography anomalies, and uncertainty on how land surface types change in response to ice retreat, and what happens if the ice sheet wants to expand beyond present-day margins). Finally, please feel free to counter my suggestions if you think I'm in error.*

Thank you for your constructive comments. We hope that we addressed your concerns in the following.

*P1L1: "the projected Greenland sea level rise contribution is mainly controlled by the interactions between the Greenland ice sheet (GrIS) and the atmosphere": while I tend to agree, relevant models can't yet fully assess the ocean contribution, so I think this statement is overconfident. Please moderate.*

We have considerably modified the text in the abstract and this statement has disappeared in the revised version:
*"In the context of global warming, a growing attention is paid to the evolution of the Greenland ice sheet (GrIS) and its contribution to sea-level rise. Atmosphere-GrIS interactions, such as the temperature-elevation and the albedo feedbacks have the potential to modify the surface energy balance and thus to impact the GrIS surface mass balance (SMB). In turn, changes in the geometrical features of the ice sheet may alter both the climate and the ice dynamics governing the ice sheet evolution".*

*P1L2: "in particular through the temperature and surface mass balance – elevation feedback": no, the atmospherically-driven GrIS SLR contribution is controlled by radiative excess/warming. Feedbacks reinforce this effect but is do not control it.*

Again, these lines have been modified (please see our previous comment).

*P1L2: "fine scale processes"->"fine scale dynamical processes" ?*

OK modified.

*P1L15: "Furthermore, in 2150, using a fix ice sheet mask, as in the no coupling method, overestimates by 24 % the SLR contribution from SMB compared to the use of the ice sheet mask as simulated in the two-way method" this seems counter to the previous statement that SLR from two-way coupling is 9.3% larger than the uncoupled case. Is the difference due to dynamic*

*discharge term?*

There is no contradiction but we acknowledge that the way this sentence was written was confusing. Actually, this sentence aims at quantifying the overestimation of the SLR projection inferred from changes in SMB only (and not from changes in simulated ice volume) when using a fixed ice sheet component. Therefore they ignore the albedo changes and the SMB-elevation feedbacks. By using such methods, we show that the use of a fixed ice-sheet mask leads to an overestimation of the GrIS contribution to SLR of ~6 % in 2150, and to an overestimation of ~23 % of the SMB (with respect to the use of a time variable ice-sheet mask). These estimations are referred to as $SMB_{MSK-NF}$ (fixed ice-sheet mask) and $SMB_{MSK-2W}$ (time variable ice-sheet mask) and are both based on the SMB-integrated method, traditionally used in RCM-based studies that have no interactive ice-sheet component. Conversely, when considering the two-way and the one-way coupling experiments, we find that the GrIS contribution to sea-level rise (computed from ice volume changes simulated by GRISLI) is 9.3 % higher when GrIS-atmosphere feedbacks are accounted for (i.e. in the two-way coupled method). In the revised version, this has been better presented (see section 4.4) and reformulated in the abstract:

*"As a result, the experiment with parameterised SMB-elevation feedback provides a sea-level contribution from GrIS in 2150 only 2.5% lower than the two-way coupled experiment, while the experiment with no feedback is 9.3 % lower. […]In addition, we quantify that computing the GrIS contribution to sea level rise from SMB changes only over a fixed ice-sheet mask leads to an overestimation of ice loss of at least 6 % compared to the use of a time variable ice-sheet mask".*

*P2L 4: "The atmospheric conditions control the variability" -> "Atmospheric conditions control variability and change"*

This section has been completely re-written to provide clarifications on surface melting and snowfall drivers before dealing with atmosphere-GrIS feedbacks:
*"The evolution of the Greenland ice sheet (GrIS) is governed by variations of ice dynamics and surface mass balance (SMB), the latter being defined as the difference between snow accumulation, further transformed into ice, and ablation processes (i.e. surface melting and sublimation). While surface melting strongly depends on the surface energy balance, snowfall is primarily controlled by atmospheric conditions (wind, humidity content, cloudiness…). However, various feedbacks between the atmosphere and the GrIS may lead to SMB variations that can therefore directly affect the GrIS total mass by impacting its surface characteristics, such as ice extent and thickness, with potential consequences on ice dynamics (e.g., due to change in surface slopes)."*

*P2L7: "SMB directly affect the GrIS total ice mass by impacting its characteristics such as thickness, ice volume and ice extent" - this can occur both directly and via impacts on ice dynamics. Explicitly state the latter (dynamics) for clarity.*

Again this part has been drastically reformulated with clarity in mind (see previous comment).

*P2L9: there are more foundational references regarding the dynamical GrIS impact on atmospheric flow. Suggest to use these in addition/instead. As just one arbitrary example: http://onlinelibrary.wiley.com/doi/10.1034/j.1600-0870.1996.00014.x/abstract*

Thank you for the reference. We have added the following:
*"These changes may in turn alter both local and global climate. As an example, changes in near-*

*surface temperature and surface energy balance may occur in response to changes in orography (temperature-elevation feedback) or in ice-covered area (albedo feedback; see Vizcaino et al., 2008, 2015; Lunt et al. 2004). On the other hand, topography changes may alter the atmospheric circulation patterns (Doyle and Shapiro, 1999, Petersen et al. 2003, Moore and Renfrew, 2005) causing changes in heat and humidity transports."*

*P2L11: "different processes and feedbacks"->"different processes and feedbacks that regulate transient ice sheet change"*

Thanks for the suggestion, we have modified the text accordingly.

*P2L16: "The climate models usually represent" -> "For example, CMIP5 climate models unanimously represented"*

We modified as: *"For example, the CMIP5 climate models unanimously represent the ice sheet component with a fixed and constant topography, even under a warm transient climate forcing".*

*P2L24: Suggest citing recent Lofverstrom et al. discussion study on resolution dependence of ice sheet conditions in GCMs: https://www.the-cryosphere-discuss.net/tc-2017-235/*

The suggested reference has been added as well as the following paragraph:

*"Using the AGCM NCAR-CAM3 run at different spatial resolutions (T21 to T85) and coupled to the SICOPOLIS ice-sheet model, Löfverström and Liakka (2017) investigated how the atmospheric model resolution influences the simulated ice sheets at the Last Glacial Maximum. They found that the North American and the Eurasian ice sheets were properly reproduced with the only T85 run. According to the authors, this is likely due to the inability of the atmospheric model to properly capture the temperature and precipitation fields (used to compute the SMB) at lower horizontal resolutions, as a consequence of the poorly resolved planetary waves and smooth topography".*

*P2L35: "the authors only consider a strict linear relationship between topography and SMB changes" - please note more clearly either here or in next paragraph why this a handicap to these methods, leading to why your approach is better*

We have added the following:

*"However, in both parameterisations by Franco et al. (2012) and Edwards et al. (2014b), the authors only consider a strict linear relationship between topography and SMB changes. Although changes in temperature can be derived from a linear vertical lapse rate, other processes governing the SMB such as those related to energy balance, precipitation or atmospheric circulation do not follow a linear relationship with the altitude. While this approach may be valid at the local scale for small elevation changes, it may lead to a misrepresentation of the SMB-elevation feedbacks for substantial changes in altitude, especially at the ice-sheet margins."*

*P2L9: "The second fundamental requirement is to represent the ice sheet topography changes in the atmospheric model by using an ISM instead of the fixed geometry usually used" This sentence is tautological since by definition a fixed geometry will not capture topography changes. Reword sentence.*

The sentence has been reworded as: *"The second fundamental requirement to describe the interactions between atmosphere and GrIS is to represent the ice sheet topography changes in the*

*atmospheric model by using an ISM (instead of the fixed geometry typically used) to take into account the effects of ice dynamics on the ice sheet topography changes".*

*Throughout text: "developped" -> "developed"*

OK, modifed.

*P4L7: 16 km high, from surface? Sea level?*

This part of the text has been changed in *"The MAR horizontal resolution is 25 km x 25 km covering the Greenland region (6600 grid points), from 60 °W to 20 °W and from 58 °N to 81 °N, and 24 vertical levels to describe the atmospheric column in sigma-pressure coordinates (Gallée and Schayes, 1994)".*

*P4L12: "hydrological cycle" -> "atmospheric hydrological cycle"?*

Yes, modified.

*How does Crocus differ/integrate with SISVAT? Please clarify. In the case where the ice sheet expands or contracts, how is under-snow (or snow free) ice sheet surface exchanged for bare land surface (or vice versa)?*

Crocus is a 1D snow model, while SISVAT is the surface model embedded in MAR. In SISVAT, each grid cell is assumed to be covered by at least 0.001% of two major surface types, namely tundra and snow (including ice sheet). Tundra is considered by SISVAT as a vegetation zone with an albedo ranging from 0.1 to 0.2 as a function of surface water and plant type. On the contrary, the Crocus snow model is used to compute the albedo of ice covered areas. In the 2W method, the percentage of tundra/snow evolves following the ice-sheet model advance and retreat. We now provide more information about the MAR model in Sec. 2.1 and we hope that the interplay between Crocus and SISVAT appears now clearer:

*"MAR is a regional atmospheric model fully coupled with the land surface model SISVAT (Soil Ice Snow Vegetation Atmosphere Transfer model, see Gallée and Duynkerke, 1997) which includes the detailed one-dimensional snow model Crocus (Brun et al., 1992) which simulates fluxes of mass and energy between snow layers and reproduces snow grain properties and their effect on surface albedo [...].Each grid cell is assumed to be covered by at least 0.001 % of tundra and snow. At each time step SISVAT computes the albedo of each surface type and the characteristics of the snowpack which are weighted and averaged as a function of the snow and vegetation coverage in each grid point, and then exchanged with MAR."*

In addition, we have included more details on the 2W coupling methodology (in Sec. 3.3):

*"At the end of a MAR model year, MAR is paused and GRISLI is forced by the downscaled SMB and ST fields with the method of Franco et al. (2012) as in PF (Eq. 7). Then, GRISLI computes a new GrIS topography and a new ice extent at 5 km which are aggregated at the yearly time scale onto the 25 km MAR grid. The aggregated ice extent is used to update the fraction of tundra relative to ice/snow covered surface type for the subsequent MAR run. To account for the differences between MAR and GRISLI topographies, the surface elevation which is aggregated onto MAR is computed from GRISLI surface elevation anomalies added to the present-day observed topography (Eq. 7). It is then used as the updated surface elevation in MAR. As previously mentioned, topography changes are negligible before 2020. Hence, changes in ice-sheet geometry are fed to MAR only*

*after this date. Compared to the NF and PF approaches, this two-way coupled method is the most accurate to represent the GrIS-atmosphere feedbacks".*

*P4L20: "The topography of the GrIS as well as the surface types (ocean, tundra and permanent ice) are provided by Bamber et al. (2013)" -> clarify this is for the NC is experiment (presumably)*

We made this clarification in the text:

*"Except for the experiment presented later in this study in which MAR is coupled to an ice-sheet model, the topography of the GrIS as well as the surface types (ocean, tundra and permanent ice) are taken from the Bamber et al. (2013) dataset aggregated on the 25 km grid."*

*P5L10: "we have repeated the MIROC5 year 2095 (representative of the years 2090s) for 50 additional years" - this repetition is certainly not representative of this time period due to lack of continued change, and also lack of internal variability. While I don't think this is a fatal flaw of the study, the authors should clearly note this caveat here and later during discussion of results, so readers clearly realize the effects of this artificial 'extension' (probably, fairly strongly reduced overall change, making the results presented here conservative).*

We acknowledge the fact that, in our approach, we discard the role of interannual variability within the GCM after the year 2100. This could indeed results in conservative estimates due to non-linearities of SMB (in particular ablation). However, the GCM imprint of the year 2095 may also increase regional changes in term of GrIS response. We present these limitations in the revised version of the manuscript in the discussion section:

In section 3, we mentioned that the use of a constant forcing from 2100 to 2150 *"implies that both climate changes and large-scale inter-annual variability are neglected beyond 2100".*

In the Discussion section*: "A second question concerns the impact of a constant MIROC5 climate used to force MAR beyond 2100. As outlined in section 3, this results in discarding the continued change that the climate will likely undergo beyond 2100 suggesting that our SLR projections are underestimated. The second consequence is that inter-annual variability is neglected after 2100. This can lead to conservative estimates of Greenland melting contribution to sea level rise in the future due to non-linearities of the SMB. On the other hand, the imprint of the 2095 MIROC5 climate may amplify regional changes of the GrIS response"*

*P6L11: Why is the annual mean bottom snowpack temperature not used as the boundary condition for the ISM instead?*

It would be indeed possible, but probably would have very low impact on the ice temperature profile simulated by GRISLI. Indeed, the annual temperature at the bottom of the snowpack is very similar to the annual mean surface temperature. Because GRISLI has a yearly time step, it can not see annual temperature variability simulated by Crocus. Therefore, we have used the annual mean temperature as a boundary condition for the ISM.

*P6L19: also just due to the long timescale of ice sheet responses?*

Yes, we agree with this comment. It seems more appropriate to only deal with the long time-scale response of the ice sheet. Our motivation has been reformulated as: *"Due to the long time scale response of the ice sheet to a given climate forcing, a proper initialisation of the model is required before performing forward experiments".*

Moreover, our spin-up procedure also includes the calibration of unknown parameters (basal drag coefficient) and our inversion procedure can be seen as a way to correct model deficiencies. That being said, we have clarified and simplified the presentation of the spin-up procedure (see Section 2.2.2) as we now directly refer to the paper published in the discussion forum of the Geoscientific Model Development journal. In the revised manuscript, this paper refers to as : Le clec'h et al. (2018).

*P7L1: what is meant by 'vertical fields'? Please clarify.*

We meant temperature and ice velocity profiles from Gillet-Chaulet et al., 2012. It is now specified in the revised manuscript.

*Spin-up procedure: How does this procedure deal with ice growth outside the observed ice sheet extent? Figure 2 suggests this ice is simply removed? If so, how does this effective strong artificial sink of ice impact all subsequent sensitivity experiments? Please explain the impacts of this clearly in the text, if this is the case.*

You are right, in our framework we apply an artificial strong negative SMB outside the observed present-day ice sheet mask. We do not think that it is a major flaw in our methodology as our spin-up procedure aims at reducing the mismatch between observed and simulated ice thickness. Assuming that MAR produces a realistic SMB on the ice sheet and because the simulated ice thickness is close to observations, we can hypothesise that our simulated ice flow is realistic. As such, in theory, the ice sheet should not grow outside the observed present-day ice sheet imprint. The artificial strong negative SMB outside the present-day ice sheet mask can be seen as a way to correct both the atmospheric model bias (e.g. positive / not enough negative SMB over the tundra) and the spin-up procedure bias (too strong ice export towards the margin). This has been explained at the end of the spin-up description (section 2.2.2).

*P8L21: why not simply start the coupling at 2005 (i.e. the end point of the 1976-2005 initialization/spin-up period)?*

Sure it would have been possible. However, as stated in the manuscript, the results would have been similar as the SMB changes through 2005-2020 does not produce any significant topography changes in GRISLI.

*P9L13: The use of topography anomalies is concerning since it implies the SMB/ST field received by GRISLI is inconsistent with GRISLI's height (for example, the ELA on the GRISLI grid would exist at a different elevation than if the GRISLI elevation was directly used). Can the authors comment on why this approach does not introduce problems with their experimental design? As it stands, this is not justified adequately. An alternate approach that would have avoided this problem would have been to use the spun-up GRISLI topography as the 'fixed' topography instead of the Bamber topography.*

The issue here is that GRISLI tends to produce steeper slopes than what is observed. This has important consequences for the climate simulated by MAR due to, in particular to katabatic winds. This is why we made the choice to maintain the realism of the simulated present-day climate (computed on the Bamber et al. (2013) topography) and the consistency between the climate simulated by MAR and the climate used to force GRISLI, downscaled at the 5 km resolution using the method developed by Franco et al. (2012).

In Section 3.2, we mentioned that *"Due to the topography differences between MAR and GRISLI, this approach has been chosen to avoid large inconsistencies between the SMB and ST fields computed by MAR and the ones corrected to account for the GRISLI topography"*.

We also discussed the impact of the anomaly method in Section 5:

*"A second limitation is related to the 2000-yr relaxation GRISLI experiment, run at the end of the spin-up procedure to reduce the model drift in terms of ice volume, that produces residual differences with the observed topography (Bamber et al. 2013) used in the MAR simulations. This has important consequences on the MAR simulated climate. In particular, the steeper slopes existing in the GRISLI topography (i.e. $S_{ctrl}$) tend to produce unrealistic katabatic winds. Therefore, we choose to use an anomaly method of the surface elevation onto which the SMB and ST fields are downscaled at the 5 km resolution grid (Eq. 7). The objective of this approach was first to maintain the realism of the simulated present-day climate computed on the observed topography (Bamber et al. 2013) and, secondly, to avoid inconsistencies between the climate simulated by MAR and that used to force GRISLI. However, this implies that the forcing climate is not fully consistent with the GRISLI topography. This should be taken into consideration in a future work to improve the quality of our results"*.

Figure 2 and other figures: 5 years is likely not long enough to generate robust climatologies. Suggest using at least 10 years instead.

We have followed your suggestions and used 10 years to compute climatologies.

*P11L10: the finding of very strong marginal cooling due to increased katabatics is very interesting and pertinent, and deserves a further explaining. It would be very useful the authors plotted overlaid near-surface wind anomaly vectors plus ST changes in'zoomed-in' plot of a good illustrative portion of the margin.*

*We provided further explanations to justify the role of katabatic winds in the marginal cooling (see section 4.2.1):*

*"Over the ice sheet, the steeper surface slopes simulated in 2W in 2150 (discussed in Sec. 4.1.2) lead to a slight increase in katabatic winds (Fig. 9). However, at the ice sheet margin, i.e. where the ice mask in MAR is below 100%, there is a substantial decrease in surface winds. This is because the change in surface elevation as seen by the atmospheric model is computed from the aggregated changes in GRISLI at 5 km. As such, a non-zero fraction of tundra, which presents no change in surface elevation, results in smaller elevation changes compared to grid cell in the same region with permanent ice cover only. This induces artificially lower surface slopes at the margin with respect to the interior and a decrease in surface winds in these regions. Altogether, the slight increase in katabatic winds over the ice sheet and their reduction at the margin lead to a cold air convergence towards the ice sheet edge (Figs. 8b and 9 and Fig. S8-S9)"*.

To support these explanations, we added a new figure (Figure 9) displaying the 2W near-surface wind vectors at the end of the 2W experiment as well as the wind strength anomaly between 2W for NF. A zoom-in plot showing near-surface wind anomaly vectors overlaid to ST changes is provided in the Supplementary Materials as Figure S7.

*Similar to above point: it would be excellent to see a quiver plot of wind anomalies over the entire*

*ice sheet, given their importance. Also would it be possible to visualize the increased mixing in the boundary layer, leading to warming in the 2-W coupled case?*

A similar plot as Figure 9 in the main text is also given in the Supplementary Materials (see Fig. S9).

*P11L23: do authors mean "Following the increase of the ST"?*

Yes, this is what we meant. However, due to modifications in the structure of the revised manuscript, this part of the text has been removed.

*P11L25: ", there is a decrease of 112 Gt yr−1 25 of ice " -> "112 Gt/yr extra ice ablates"*

The sentence has been changed in: *"This process is faster in 2W than in NF and PF. In 2150, the ablation zone is 14 % (resp. 11.7 %) larger in 2W than in NF (resp. PF) causing 112 Gt yr$^{-1}$ of extra ice ablation in 2W (w.r.t NF)".*

*P11L30: "14 % larger in 2-W" - can an estimate be made of the uncertainty in this value (and others) due to interannual variability? Put another way, can the authors confirm that the changes they see are significant in the face of background noise in ablation area (for example)?*

As specified in sections 2 and 5, the use of a constant climate forcing for MAR after 2100 (here the MIROC5 climate simulated for year 2095) implies that the inter-annual variability is neglected beyond 2100. As such, the relative changes in ablation areas after 2100 mentioned in the text are necessarily statistically significant, at least within the framework of our experimental setup. However, we acknowledge that a better approach would be to perform similar simulations with a prolonged RCP8.5 scenario (not available at the time of this study).

*P12L5: "lower surface temperature over these regions" - suggest reinforcing to readers once more here that this is \*relative\* to the NC experiment.*

Thanks for this remark. We paid attention to clarify the text when dealing with relative changes.

*P12L8/9: what does the +/- indicate here?*

This is the mean value +/- the root mean square error over the region. In the revised manuscript, the mean values do not longer appear with the +/- root mean square error. We chose to present the mean value results with the 5$^{th}$ and 95$^{th}$ percentiles when necessary.

*P12L13: "become ice or snow-free or snow free, exhibiting bare ice " this is confusing. What happens if the entire GRISLI ice column disappears? Does tundra emerge?*

This part of the text has been modified and changes in ice-sheet extent are now only discussed in Section 4.4.

The point which which is addressed here has been clarified (see section 2.1):

*"[In MAR], each grid cell is assumed to be covered by at least 0.001% of tundra and snow. At each time step SISVAT computes the albedo of each surface type and the characteristics of the snowpack which are weighted and averaged as a function of the snow and vegetation coverage in each grid point, and then exchanged with MAR".*

In both the NF and the PF experiments, the ice-sheet mask, as seen by GRISLI is not updated in the atmospheric model and MAR sees the present-day observed ice-sheet mask throughout the simulation. In the 2W experiment, the ice extent computed by GRISLI is then aggregated to MAR to update the fraction of tundra relative to ice/snow covered surface type for the subsequent MAR run. As a result, if the entire ice column disappears, MAR sees in each grid cell a fraction of tundra of 99.999 % and modifies the albedo accordingly.

*P12L25: Previous studies have highlighted a strong decrease in ice discharge across outlet glacier grounding lines as a consequence of increased surface melting. E.g. Gillet-Chaulet 2012, Goelzer 2013 and others. Is this same effect seen here?*

At the end of the 2W experiment (2140-2150), there is a decrease of surface velocities compared to the 2000-2010 mean period (Figs. 6a, 7c), suggesting that ice discharge across outlet glaciers is reduced. Moreover, the negative anomaly of ice flux divergence (Fig. 5b) shows an upstream ice accumulation (i.e. ice accumulates faster than it discharges through outlet glaciers). These results strongly suggest a decrease of ice discharge across outlet glaciers, similarly to what was found by Gillet-Chaulet et al. (2012) and Goelzer et al. (2013).

*P12L25: Is it completely correct to say the entire SLR contribution is caused by the 'melting contribution'?*

In our model, there is only a very few number of grid points in contact with the ocean. Therefore, calving is negligible and melting remains the dominant contribution to sea-level rise. However, to avoid confusions, we removed all expressions such as the Greenland melting contribution to SLR in the revised manuscript and simply use the Greenland ice sheet contribution instead.

*P12L25: Can the authors quantify the reduction in marine margin extent in 2-W?*

As explained in our previous response, the number of grid points in contact with ocean is negligible in our model. This is likely due to the too coarse GRISLI resolution (5 km) that prevents from properly resolving the complex topographic features of marine terminating glaciers. As a result, it is not possible to quantify accurately the marine margin extent. To illustrate the limitations induced by the coarse ice-sheet model resolution, we added the following paragraph in the Discussion section:

*"Regarding the ice-sheet model, a 5 km horizontal resolution does not permit to capture the complex ice flow patterns of smallest outlet glaciers, whose characteristic length scale can be less than 1 km (Aschwanden et al., 2016) and to quantify accurately the ice discharge at the marine front. This may have large implications in the sea-level rise estimates. Using a 3D ice-sheet model with prescribed outlet glacier retreat, Goelzer et al. (2013) found an additional SLR contribution from outlet glaciers of 0.8 to 1.8 cm in 2100 and 1.3 to 3.8 cm in 2200, with the influence of their dynamics on SLR projections decreasing with time and with the increasing importance of the atmospheric forcing. This is in line with the fact that ice dynamics act to counteract ice loss from surface melting (see Section 4.2), as previously outlined by several authors (Edwards et al., 2014b, Goelzer et al., 2013, Huybrechts and de Wolde, 1999). However, despite the possible decreasing influence of marine terminating glaciers, at the centennial time scale, it seems to be preferable to evaluate more accurately the impact of ice dynamics and to better capture the complex geometry of fjords surrounding the marine-terminating glaciers".*

*P13L1: "This higher integrated SMB, obtained when using no updated ice sheet mask" - do the authors mean "lower"..? This sentence seems to directly contradict the previous sentence. If I'm mistaken here, a clearer description of the processes here is needed.*

Yes, you're right. In the revised manuscript (Section 4.4), we tried to better explain the issues related to the integrated-SMB method. We hope that the text has been clarified enough:

*"A widely used method to estimate the projected GrIS to global sea-level rise is to compute the GrIS mass loss as the time-integral of the SMB computed by an atmospheric model over a fixed ice-sheet mask (Fettweis et al., 2013, Meyssignac et al., 2017, Church et al., 2013). In the present study, we go a step further since the ice mass variations related to SMB changes are computed over a changing ice-sheet mask as simulated by GRISLI. However, in both the NF and the PF experiments, the atmospheric model does not account for the variations in the ice-sheet extent simulated in GRISLI and the ice-sheet mask, taken from the observations (Bamber et al., 2013) is kept constant throughout the simulation. Taking the changes in ice-sheet mask into account may have strong impacts on the computed GrIS contribution to sea-level rise. To illustrate the influence of the ice sheet mask, we used the SMB outputs from the NF experiment at the MAR resolution and applied the integrated SMB method over the fixed observed ice-sheet mask ($SMB_{MSK-NF}$) and over the updated 2W mask ($SMB_{MSK-2W}$). Results reported in Table 2 indicate differences in SMB values exceeding 23 % in 2150. In the same way, compared to a time variable ice-sheet mask, the use of a fixed ice-sheet mask overestimates the sea-level rise by ~6 % in 2150. Though a bit lower, this number is far from being negligible compared to the errors made when the SMB-elevation feedbacks are not taken into account (i.e. 7.6 %) and when all the feedbacks are ignored (i.e. 9.3 %). This strongly suggests that realistic SLR projections cannot neglect the evolution of the ice-sheet extent, only accounted for through the use of an ice-sheet model".*

*General: The authors should consider quantifying actual feedback factors associated with the inclusion of elevation feedbacks (see Roe 2009, Reviews of Geophysics). This would be a good benchmark number to produce, for other works to compare to.*

We agree that a formalised way to quantify the elevation feedback would be very interesting, in particular for inter-comparison exercises. However, the definition of such a metric has yet to be done. For now, we only compare our SLR projections with and without the elevation feedback to other papers available in the literature a similar approach has been followed (e.g. Vizcaino et al., 2015; Calov et al., 2018).

*P14L11: "As for the ISM, increasing the grid resolution of MAR" - do you mean "as for the regional climate model"..?*

No, we think that an increase in both ISM and RCM resolutions could better constrain the SLR contribution from Greenland ice sheet. These aspects have been detailed in the Discussion section (in the revised manuscript).

*P14L35: "underestimated by simulating." Unclear.*

"By simulating" should be removed. This error was probably due to an improper "copy-paste"

*P15L1: "surface albedo and strength of katabatic winds." -> "surface albedo and strength of katabatic winds, with a demonstrably strong return influence on SMB"*

The Discussion section has been entirely re-written and this sentence has been removed from the

original text.

*P15L27: "optimal resolution of the ice sheet and the atmospheric model, for ISM-RCM coupling." While an interesting-sounding statement, I find it also a bit vague: by optimal, do the authors mean something like "of high enough respective resolutions to resolve both important atmospheric and important ice sheet dynamical processes"?*

The point here is to find "high enough ISM and RCM resolutions to resolve both important atmospheric and important ice sheet dynamical processes", while keeping a reasonable computational time. In the revised manuscript, the sentence has been modified by: "*However, a compromise must be reached between the additional computing resources and the required degree of accuracy of sea-level projections*".

*P15L30: "The next step of this study. . ." as described, this is extremely ambitious, with many challenges that outstrip the effort to implement atmospheric coupling. If it is truly a planned next step; great! But if not, I'd suggest not claiming to plan to do this.*

In the revised manuscript (Section 5), we rather gave a few examples to illustrate the importance of having a description of the ocean-atmosphere-GrIS coupled system describing the coupled ocean (see paragraph below), but we followed your recommendation and avoided expressions such as "the next step of this study":

"*There is a growing number of evidence for attributing the acceleration of outlet glaciers to the intrusion of warm waters from adjacent oceans in the fjord systems or in the cavity of floating ice tongues (e.g. Straneo et al., 2012; Johnson et al., 2011, Rignot et al., 2015) that can destabilise the glacier front and/or favour the ice-shelf breakup, decreasing thereby the buttressing effect and increasing the ice calving. In turn, the released freshwater flux in ocean may impact sea-surface temperatures, oceanic circulation and sea-ice cover. Moreover, atmosphere-ocean feedbacks also have an impact on the GrIS. As an example, Fettweis et al. (2013) showed that the disappearance of Arctic sea ice in summer induced by ocean warming enhances surface melting in northern Greenland through a decrease of surface albedo and the subsequent atmospheric warming. Thus, the absence of the oceanic component in our modelling setup appears as a limiting factor, although, the direct impact of ocean via sub-shelf melt at the ice sheet margin will likely be limited in the future as a result of inland retreat of GrIS*".

Note however, that MAR has already been coupled to a regional configuration of the oceanic model NEMO (e.g. Jourdain et al., 2011), but applied to the Ross Sea sector in Antarctica. We can therefore reasonably envisage that in the coming years, we will be able to develop a coupled atmosphere-ocean-ice-sheet model.

Jourdain, N. C., Mathiot, P., Gallée, H., Barnier, B. : Influence of coupling on atmosphere, sea ice and ocean regional models in the Ross Sea sector, Antarctica, Clim. Dynam., 36, 1523-1543, doi: 10.1007/s00382-010-0889-9, 2011.

*General: while the writing is 100% understandable and clear, a final proof-read by a native English speaker would be useful as a final stage, if possible, to clear up remaining small grammar issues.*

We are aware of the fact that many English mistakes and syntax errors appeared in the submitted manuscript. We made a huge effort to improve English writing.

---

## Author Comment (AC2) · 25 Jul 2018

**We would like to thank the reviewer for the evaluation of our study and the constructive comments that helped us to improve the manuscript. Please find below the reviewer's comments in black font and the author's response in blue font.**

**Responses to Reviewer 2**

*The paper claims to be focused on assessment of the future of the GrIS through 2150. But in fact, it seems more focused on assessment of a new technique for RCM-Ice Model coupling. Throughout the paper, focuse shifts back and forth between the two. The experiment is to run a future simulation of the GrIS using MAR coupled with GRISLI in three different ways, and then compare/analyze the results from each other.*

*The coupling method is interesting, but the GrIS is more interesting. I believe the paper would be better if it would keep its focus firmly on the GrIS, while keeping the methods separate. I ultimately want to know, what do we learn about Greenland? Unfortunately, the figures do not really support that. Figures 1-4 do in a way; but the rest of the figures only really tell us about technical differences between coupling technique.*

In the revised version, we have largely restructured the manuscript. We now start the result section with a thorough analysis of the GrIS evolution simulated with the most comprehensive method, i.e. the two-way (2W) coupling. The primary focus being now the GrIS evolution, we hope that we have addressed your concerns on this point.

*The experiments in the paper show that the different coupling techniques provide different answers. Unfortunately, it is hard to know which answers are closer to the truth, because there are no controls. I came into this believing that the most sophisticated coupler would produce the most melt and also be more accurate; but I had no proof on the accuracy part. This paper has reinforced my prior assumptions, without providing any additional evidence on accuracy. I am therefore hard pressed to say what it has added to my understanding of coupling technique.*

All methods, and more generally all models, have their flaws. As stated in the manuscript, both the NF (No Feedback) and PF (parameterized altitude feedbacks) methods (corresponding respectively to NC and 1W in the first version of the manuscript) do not account for the change in atmospheric circulation induced by the change in ice-sheet orography and albedo. The PF method intends to represent the non-linearities of SMB changes with linear corrections based on vertical SMB gradients. Finally, it is fair to say that, compared to the NF and PF methods, the 2W method is the most physically based approach. The two approaches (NF and PF), are inaccurate by construction but have been widely used in the community because of the complexity of including a dynamical ice sheet model in RCMs. Related to your concern on accuracy, we acknowledge the fact that there is only minor constraints to test the validity of our projections: the satellite era covers a relatively short period for which the change in ice sheet topography is small. Thus, although we can state firmly that the 2W method has a stronger physical realism we cannot however guarantee the accuracy of the projections.

*I did learn some things about the future Greenland itself, in spite of the figures not really helping with this. I learned:*

1. *Expect a steeper slope and stronger katabatic winds, in addition to the expected smaller ice sheet. This will result in colder (not warmer) temperatures near the coast.*
2. *In parts of Greenland, the ELA could be as high as 3000m by the year 2150. I find that idea astounding, at 77 degrees North latitude. Some discussion of this result would be really interesting.*
3. *Expected sea level rise contribution of Greenland in 150 years is 20cm; and the rate of melting will be continuing to rise at that point.*
4. *Ice loss and SMB are highly correlated over the next 150 years; so much so that plots of the two look highly similar. Unfortunately, the paper does not try to quantify the correlation.*

Thanks for mentioning that. In the revised version of the manuscript, these points appear more clearly along with the description of the 2W results in Sec. 4.1. Katabatic winds are discussed in Section 4.2.1.

Concerning the ELA, we did not mention in the original text that it could be situated as high as 3000 m. In the revised manuscript, we added more information about the shift of the ELA towards higher altitudes (see Fig. 3 and section 4.1.1):

*"The equilibrium line altitude (ELA, i.e. altitude for which SMB = 0) increases significantly between the beginning and the end of the 2W experiment, as a consequence of increased runoff for areas below 2000 m. As an example, at around 73.5 °N, on the eastern side of the ice sheet, the ELA moves from ~1000 m to ~2500 m (Fig. 3). In other regions, at the end of the 2W experiment, the ELA is generally situated between 1500 and 2000 m high, except in the northern part where it is between 1000 and 1500 m. This shift of ELA towards higher altitudes represents an increase of 24 % of the ablation area between the beginning and the end of the experiment".*

Concerning the correlation of SMB with total mass loss, we added more discussion on the role of ice dynamics (see Section 4.1.3). As now shown in this section, ice dynamics act to counteract ice loss from surface melting ( see Figs 4 and 5). This was also noticed in previous studies (e.g., Goelzer et al., 2013, Edwards et al., 2014a). In turn, ice dynamics is impacted by changes in ice-sheet geometry (see Fig. 6a).

*For the record, here's what I learned about coupling techniques:*

1. *Integrating SMB over a fixed ice mask over time is a poor way to calculate total SLR contribution, due to the changing ice mask.*
2. *The 2w case melts more than the 1w or NC case in the RCP8.5 scenario.*
3. *Full Stokes solvers might yield better results.*

*Overall... I think this paper has done some interesting modelling runs, but so far has mostly failed to draw interesting conclusions from those runs, and to focus the reader's attention on those conclusions. I would suggest the authors think through the question "What have we learned about Greenland;" and then re-do the figures and commentary to support that learning, and focus the reader's attention on it. The paper will also need significant disucssion of these Greenland results, in comparison with other papers that have looked at the future of Greenland; for example, Vizcaino et al 2015. Especially interesting would be places where this*

*paper predicts something DIFFERENT from those other papers, and why? In this way, the reader needs to be drawn to focus on the most interesting things — the surprises! — first, without having to dig for them.*

In the revised version of the paper we did our best to organise the ideas following your suggestion, emphasizing on the fate of the GrIS in our projections with the 2W method. We have also added a thorough discussion with existing literature (see in particular Section 4.4 and Section 5). These sections, as well as the Conclusions (Section 6) have been entirely re-written.

*Once the paper has focused primarily on Greenland, I would then think about how to add discussion of a new coupling technique, without taking away from the main scientific focus of the paper. But in the absence of any solid provable way to prove that one coupling technique is better than another, I would avoid making too many claims about the 2w coupling; just that you think it is better, and it certainly melts more ice.*
*In the parts (bulk) of the paper focused on Greenland, I would use whatever coupling technique you think is most realistic.*

Here, we do not agree. The 2W method is definitively more physically based than the two other methods and explicitly represents feedbacks that are lacking in NF and PF. For example, the change in albedo in response to ice sheet retreat exerts a major control on local SMB changes that is completely discarded with the two simple coupling techniques. A simpler approach can provide similar estimates for GrIS melt but not always for physical reasons.

*A secondary issue: the paper reports many numbers, and only a few of them have error bars. Where did those error bars come from, and why are error bars not reported for other numbers? Would it be possible to get error bars for other numbers?*

Because we only have one scenario for each coupling technique we cannot assess statistically the uncertainty in our projections. The +/- signs that you saw for some numbers in the original manuscript stand for the spatial average of the standard deviation for a given variable. For example, in Sec. XX for deltaH=XX+/-YY, the YY is simply the standard deviation in ice thickness change (i.e. the XX value) from the initial condition for a given temporal snapshot. However, in the revised manuscript we now provide the 5th and the 95th percentile values to indicate the range of a given variable.

*p.2l.24: Studies by Vizcaino et al (and also at GISS; see Fischer Nowicki 2014) use elevation classes to develop an SMB. Elevation classes are mathematically equivalent to custom-designed gridcells that follow elevation contours. They are therefore able to offer high resolution in the direction of the slope gradient, while continuing with low resolution perpindicular to the gradient.*

We agree. Their technique is a way to downscale the SMB from their coarse GCM grid. The technique of elevation classes to downscale SMB is explained in Vizcaino et al. (2013), not in Vizcaino et al. (2015) that was cited in the first version of the manuscript. Following your suggestion, we also mentioned the study by Vizcaino et al. (2013) in the revised manuscript in the Introduction section:

*"To circumvent the low resolution, some authors have used the method of elevation classes and are therefore able to offer high resolution in the direction of the slope gradient (e.g. Vizcaino et al., 2013)".*

*p.8 l.25: I have traditionally used different labels for the different coupling strategies described. Your "NC", I have traditionally called "1-way coupling." Your "2w", I would call "serial 1w coupling". Your "1w coupling," I would call "corrected 1w coupling." Given the differences in terminology, it's probably best to describe what each of your schemes is (which you do), but don't assume that others would use the same names. BTW, none of the coupling schemes here conserve energy, in the sense that two-way couplers (say) between the ocean and atmosphere typically do conserve energy. Therefore, I would be reluctant to call any of them true "two-way coupling."*

Following your advice in agreement with the two other reviewers, we renamed the coupling experiments. The experiment with no feedback representation is now called NF (for no feedbacks). The experiment which represents the elevation feedbacks by correcting the MAR outputs is called PF (for parameterized feedbacks). The two-way experiment name remains identical (2W). Since the 2W method does not account for the ocean and since it is based on topography anomalies, we removed all the occurrences of "full two-way coupling" and "fully coupled" replaced them by "two-way coupling".

*p.9 l.7: Why is the 2w scheme more expensive? I see that you have to run the GCM and ice model together, rather than separately. But is any more expense actually involved?*

In case you do not have an existing MAR simulation, it is true that the 2W is not drastically more expensive than the two other methods since the only difference is the additional time needed by the ice sheet model (negligible compared to the atmospheric model). However, the major advantage of the NF and PF methods is that we can use existing MAR simulations. In this case, we can run multiple sensitivity experiments since only the ice sheet model is run. We have clarified this point in the text:

*"[The PF] method offers the possibility to account artificially for the elevation feedbacks when using existing RCM simulations in which the topography is kept constant. As such, it is also transferable to any ice sheet model".*

*p.9 l.21: Fig. 1 does not support the text. Now I see Fig. 1 is reporting anomalies; but I think it would be more interesting (and no less informative) if it would report actualy Temperature.*

Following your suggestions to focus on what happens to GrIS, we start the Result section with a description of the results obtained with the 2W experiment. Therefore, Figure 1 has been changes. It now displays (in absolute values, not anomalies) the evolution of SMB and its components integrated over the whole ice sheet. The spatial distribution of the surface temperature anomaly (2140-2150 vs. 2000-2010) is now given in Fig. 2d.

*p. 10 l.2: Cause-and-effect is backwards. Actually, the lower SMB is the CAUSE of the ELA shift.*

We totally agree with the reviewer. More precisely, the ELA shift is mainly due to increased runoff (see Fig. 2c). This has been clarified:

*"The equilibrium line altitude (ELA, i.e. altitude for which SMB = 0) increases significantly between the beginning and the end of the 2W experiment, as a consequence of increased runoff for areas below 2000 m".*

*p. 10 section 4.1.2: This is the one section of the paper with error bars. How were those error bars computed, it didn't say? Unfortunately, some of the values reported are not statistically significant; and many others are barely. A more clear way to report the reports in this section would be something like "we saw no statistically significant change in the GRISLI ice sheet in the years 2000-2050." This conclusion is already pretty apparent in the figure: the "interesting stuff" happens further out in time, especially with the more advanced coupling.*

As stated earlier in response to one of your comment, the values given in the section 4.1.2 of the original manuscript were the spatial averages of the standard deviation. The idea behind these numbers was to have an idea on how geographically different is the variable of interest. However, we agree that these numbers, averaged over the entire ice sheet do not illustrate statistically significant changes. In the revised manuscript, the results are most often discussed as a function of the altitudinal locations. Therefore it does no longer make sense to provide quantitative results averaged over the entire ice sheet. Instead, we often used the 5th and the 95th percentile values, as previously mentioned.

*p.10 l.12-24: This looks like an explanation for the increased slope; but I'm not following it.*

The increased slopes are simply due to larger and negative SMB changes at the margin relative to the interior. Changes in surface slopes have consequences on ice dynamics with increased slopes leading to increased velocities. We have made substantial text modifications in this paragraph that now reads:

*"The changes in local ice dynamics between the first and the last 10 years of the 2W experiment are also related to changes in surface slope and ice thickness, particularly at the margins. To investigate the ice dynamics changes at the local scale, we used the examples of the Jakobshavn (western coast) and the Kangerlussuaq (eastern coast) glaciers for which the fine scale structures of the ice velocity, obtained after the GRISLI initialisation procedure, are relatively well reproduced compared to the observations (Figs 7ab).*

*For the Jakobshavn glacier, and for altitudes above 1500 m, the vertically-averaged ice velocities increase by more than 15 m yr$^{-1}$ (i.e. +10 %) as a result of increasing surface slopes, and slow down by more than 200 m yr$^{-1}$ (i.e. +29 %) for altitudes below 1000 m due to the decreasing ice thickness (Fig. 7c). For altitudes above 500 m, the vertically-averaged velocity is mainly driven by the SIA velocity (Figs. 7c-e). On the contrary, below 500 m, basal sliding velocities are large due to low basal drag coefficient (see Fig. 3 in Le clec'h et al., 2018) and the SSA velocity component dominates the ice flow (Figs 7c and 7g). However, while basal drag is lower in locations below 500 m, the ice flow is limited by the strongly reduced ice thickness (Fig. 4).*

*The Kangerlussuaq glacier is located in regions where the bedrock is characterised by a succession of valleys surrounded by mountains merging in a canyon where the deepest part is located 100 km away from the coast (Morlighem et al., 2017). The ice flow of the Kangerlussuaq is therefore divided in different branches with increasing ice velocities towards*

*the ice sheet margin and becoming even larger when merging in the canyon (Fig. 7b). As for the Jakobshavn glacier, the ice flow accelerates at the end of the 2W experiment as a consequence of the increase in surface slope for high altitudes (~2000-2500 m, see Fig. 4). Conversely, a strong decrease of the ice flow is found in most of margin regions (Fig. 7d) directly related to the ice thinning (Fig. 4). Contrary to the case of the Jakobshavn glacier that presents large basal sliding velocities only below 500 m, the Kangerlussuaq shows low basal drag coefficients in the entire glacier (see Fig. 3 in Le clec'h et al. 2018) and thus the ice flow is mainly governed by the SSA component (Fig. 7h)".*

*p. 10: In general, please report ice loss in dual units: both Gt, and mm of sea level rise.*
*If this were done consistently, then section 4.2.3 would barely be needed. Section 4.2: Now, the paper stops telling us about Greenland, and analyzes minute differences between the coupling techniques Not so interesting.*

Concerning the units, we adopted the following conventions: Integrated SMB values (over the whole ice sheet) are given in Gt yr$^{-1}$, while spatially-discretized SMB values are given in m yr$^{-1}$ for consistency reasons with units of ice thickness variations from which our GrIS contributions to sea-level rise are inferred. As such SLR units are in cm.

Section 4.4 in the revised manuscript replaces the former Section 4.2.3. This new section has been largely modified. In particular, we added an extended discussion to compare the different experiments and the impact of feedbacks.

*p.11 l.20: The word "probably" is used. This indicates a hypothesis; how can that hypothesis be tested?*

We observe a strong snowfall increase in the northeastern part of the ice sheet, mainly occurring in autumn (see Fig. S2d), explaining the SMB increase in this region.

*p.12 section 4.2.2: Ice thickness and SMB maps are highly correlated throughout this paper. For that reason, section 4.2.2 says pretty much the same thing as section 4.2.1. It would be better to (a) talk about the correlation explicitly, even quantify it, and then (b) keep ice thickness and SMB together in one section every time it is discussed in the results.*

We agree with you. Ice thickness and SMB maps show that both are highly correlated as shown with Figures 2a and 4 in the revised manuscript and with the plot displaying the SMB anomalies vs the ice thickness anomalies between the last ten years (2140-2150) and the first ten years of the 2W experiment (see below). The values of the regression coefficients also emphasizes the high correlation between both variables (e.g. for 2W: $R^2$ = 0.92).

[Figure]

*Caption: Surface mass balance anomalies vs the ice thickness anomalies simulated in the 2W experiment. The anomalies are taken between the 2140-2150 and the 2000-2010 mean periods. Solid lines represent the linear regression lines for the 2W (blue, $R^2$ = 0.92), PF (red, $R^2$ = 0.93) and NF (yellow, $R^2$ = 0.92) experiments.*

The high correlation between ice thickness anomaly and SMB anomaly shows that climate change due to the imposed RCP forcing is the major control on the Greenland ice sheet geometry change. However, we find important to keep two sections presenting on one hand the changes in SMB and, on the other hand, the changes in ice thickness because it allows to better constrain the role of ice dynamics. Indeed, in our revised version, we show that ice dynamics counteracts the SMB signal (see Section 4.1.3, Fig. 5 and the following paragraph):

*"To quantify the role of ice dynamics on the GrIS geometry (Fig. 4), we plotted the ice flux divergence integrated over 150 years (2000-2150, see Fig. 5b). In particular, over the central plateau, the cumulated SMB (Fig. 5a) reaches about +50 m, 40 m of which are transported away by the ice dynamics (Fig. 5b). As a result, the ice thickness anomaly is reduced to only ~10 m in this region (Fig. 4). An opposite behaviour is found near the western coast, where the ice melting is partly compensated by ice convergence, resulting in a less negative ice thickness anomaly than that related to the SMB forcing. This shows that ice dynamics act to counteract ice loss from surface melting, as previously noticed by several authors (Huybrechts and de Wolde, 1999, Goelzer et al., 2013, Edwards et al., 2014b). As a consequence, it appears to be essential to account for ice dynamics to estimate accurately the mass balance of the whole ice sheet".*

*p.12 l.30: I appreciate that doing wrong calculations will give the wrong answer. I'm glad that you are not doing that. But is this worth half a section to explain? It seems you are going out of your way because someone else did something fishy.*

Our study is the first one to provide the GrIS melting projection that makes use of a RCM coupled to an ice sheet model. This means that all the previous studies based on RCMs, did

not consider the change in ice sheet mask. We therefore think that this section is particularly relevant for the ice-sheet surface mass balance community. To emphasize the importance of the results, this section has been re-written:

*"A widely used method to estimate the projected GrIS to global sea-level rise is to compute the GrIS mass loss as the time-integral of the SMB computed by an atmospheric model over a fixed ice-sheet mask (Fettweis et al., 2013, Meyssignac et al., 2017, Church et al., 2013). In the present study, we go a step further since the ice mass variations related to SMB changes are computed over a changing ice-sheet mask as simulated by GRISLI. However, in both the NF and the PF experiments, the atmospheric model does not account for the variations in the ice-sheet extent simulated in GRISLI and the ice-sheet mask, taken from the observations (Bamber et al., 2013) is kept constant throughout the simulation. Taking the changes in ice-sheet mask into account may have strong impacts on the computed GrIS contribution to sea-level rise. To illustrate the influence of the ice sheet mask, we used the SMB outputs from the NF experiment at the MAR resolution and applied the integrated SMB method over the fixed observed ice-sheet mask ($SMB_{MSK-NF}$) and over the updated 2W mask ($SMB_{MSK-2W}$). Results reported in Table 2 indicate differences in SMB values exceeding 23 % in 2150. In the same way, compared to a time variable ice-sheet mask, the use of a fixed ice-sheet mask overestimates the sea-level rise by ~6 % in 2150. Though a bit lower, this number is far from being negligible compared to the errors made when the SMB-elevation feedbacks are not taken into account (i.e. 7.6 %) and when all the feedbacks are ignored (i.e. 9.3 %). This strongly suggests that realistic SLR projections cannot neglect the evolution of the ice-sheet extent, only accounted for through the use of an ice-sheet model".*

*p. 13 l.2: the last sentence of this paragraph is the most important. Don't "bury the leded"... put it up at the front.*

We agree with you. In the revised manuscript, an entire paragraph is devoted to the role of the katabatic wind feedback as simulated in our model. We also added the new Figure 9 to support our findings (see Section 4.2.1):

*"Over the ice sheet, the steeper surface slopes simulated in 2W in 2150 (discussed in Sec. 4.1.2) lead to a slight increase in katabatic winds (Fig. 9). However, at the ice sheet margin, i.e. where the ice mask in MAR is below 100%, there is a substantial decrease in surface winds. This is because the change in surface elevation as seen by the atmospheric model is computed from the aggregated changes in GRISLI at 5 km. As such, a non-zero fraction of tundra, which presents no change in surface elevation, results in smaller elevation changes compared to grid cell in the same region with permanent ice cover only. This induces artificially lower surface slopes at the margin with respect to the interior and a decrease in surface winds in these regions. Altogether, the slight increase in katabatic winds over the ice sheet and their reduction at the margin lead to a cold air convergence towards the ice sheet edge (Figs. 8b and 9 and Figs. S8-S9). Another consequence of the katabatic winds increase due to increased surface slopes in the GrIS interior, is to enhance the atmospheric exchanges along the slope of the ice sheet. The area with lower atmospheric pressure generated by the stronger katabatic winds is filled in by the warmer air coming from higher atmospheric levels in the boundary layer. The warming of the upper part of the boundary layer combined with the lower surface elevation, explains the ST increases in the interior of the GrIS".*

*p.14 l.15: I don't believe this argument on ice-ocean feedback. We know that tidewater glaciers retreat VERY quickly once they become imbalanced. How many tidewater glaciers will be left for us to simulate in the year 2050, 2100 or 2150? And what about going beyond that — when the REALLY interesting things start to happen? I just don't believe that ocean coupling is very important for GrIS.*

We agree with you concerning the high probability of having a decreasing influence of outlet glaciers in the future as a result of increased melting in margin areas. We have outlined this in the discussion section (see section 5). However, it remains difficult to accurately evaluate the time scale at which the influence of outlet glaciers on the whole Greenland ice sheet will be negligible. At the centennial time scale, it is therefore highly desirable to have a good representation of tide water glaciers because they have important consequences on inland ice dynamics. A strong change in ice dynamics could in turn strongly modify the SMB signal and the projected sea-level rise contribution. This process cannot represented in our model because of the too coarse GRISLI resolution. As an example, Goelzer et al. (2013) found an additional SLR contribution from outlet glaciers of 0.8 to 1.8 cm in 2100 and 1.3 to 3.8 cm in 2200, as mentioned in Section 5. In addition, ocean may exert a strong influence on ice dynamics through the intrusion of warm waters in the fjord system that can accelerate the destabilization of marine terminating glaciers and the subsequent ice discharge. This leads to a release of freshwater flux in the ocean, modifying oceanic circulation, sea-surface temperatures and sea-ice cover and the exchanges at the atmosphere-ocean interface, resulting in fine in SMB changes (due to changes in external forcings). These ideas have been developed in Section 5:

*"There is a growing number of evidence for attributing the acceleration of outlet glaciers to the intrusion of warm waters from adjacent oceans in the fjord systems or in the cavity of floating ice tongues (e.g. Straneo et al., 2012; Johnson et al., 2011, Rignot et al., 2015) that can destabilise the glacier front and/or favour the ice-shelf breakup, decreasing thereby the buttressing effect and increasing the ice calving. In turn, the released freshwater flux in ocean may impact sea-surface temperatures, oceanic circulation and sea-ice cover. Moreover, atmosphere-ocean feedbacks also have an impact on the GrIS. As an example, Fettweis et al. (2013) showed that the disappearance of Arctic sea ice in summer induced by ocean warming enhances surface melting in northern Greenland through a decrease of surface albedo and the subsequent atmospheric warming. Thus, the absence of the oceanic component in our modelling setup appears as a limiting factor, although, the direct impact of ocean via sub-shelf melt at the ice sheet margin will likely be limited in the future as a result of inland retreat of GrIS".*

*\*\* Any idea what happens beyond the year 2150? I know it's outside the scope of this paper. But this paper opens up more tantalizing questions by simulating a non-steadystate process just a little bit of the way — to a point where the changes are continuing to accelerate. What does this simulation look like in 500 years? 1000 years? 5000 years? How important are the feedbacks on that timescale?*

Unfortunately, running the MAR model over such long time scales is out of reach for the time being because of the considerable computational resources it would require. However, from the ice-sheet perspective, we can reasonably expect:

a/ an amplification of the SMB-elevation feedbacks, as suggested by the results presented in this paper (see Section 4.4);

b/ a smaller ice-sheet extent (possibly combined with a larger ablation area) with therefore a growing influence of the albedo effect amplifying warming and surface melting (see Section 4.4);

c/ increased surface slopes favouring thereby (see Section 4.1.3):

      i/ the convergence of cold air in margin areas through the effect of katabatic winds, acting therefore against warming;

      ii/ the increase of surface ice velocities in the interior regions.

d/ decreased ice thickness leading to a reduction of ice velocities (see Section 4.1.3)

e/ inland retreat of outlet glaciers resulting in their limited influence on ice dynamics (see Section 5);

f/ Multiplication of melt ponds at the surface of the ice sheet, possibly even in high altitude areas leading to:

      i/ surface albedo reduction;

      ii/ increased lubrification and basal sliding.

Of course all the processes listed above should be investigated with a coupled climate-ice-sheet model to investigate their relative influence at different time scales. In addition, atmosphere-ocean-ice sheet feedbacks should also be considered (see our response to your previous comment).

*Fig 6A: Why is there a vertical-stripe pattern in western Greenland? That makes me suspicious of the model. Please explain...*

This pattern is due to the interpolation method between the coarse MAR grid and the finer GRISLI grid

*** Figures: Please make sure of the following in figures:*

*a) Avoid the rainbow color scale in most cases (Fig 4). There are better choices.*

*We kept the rainbow scales, but we paid attention to the choice of the colour to better illustrate our purpose.*

*b) If you do use the rainbow, avoid splitting green at zero (Fig 4A). One figure has green for both positive and negative numbers; not cool.*

*We agree. This has been changed.*

*c) Avoid a color scale that's read on one end and violet on the other; because then the smallest and largest values look almost the same.*

*Once again, we agree with you. Colour scales have been changes accordingly.*

*d) When using color scales with red on one end and blue on the other, make sure that red always corresponds to places that are melting / getting warmer / losing mass; and blue corresponds to the opposite. Reverse the color scale if needed, in order to keep this consistent.*
*All the colour sales are now consistent: Blue colours correspond to a decrease of the displayed variable, and red colours represent the opposite.*

*e) The figures in this paper all use different color scales and conventions, for no apparent reason. It looks like they don't belong together. Please make them more uniform, unless there's a good reason for the difference.*
*We followed your recommendation.*

*f) Please put a title on top of every plot, in font large enough to read. Make sure that every plot has units on every axis (either the color scale, or the x-y axis. Most fonts on most figures need to be larger.*
*We have put a title on most figures and the fonts are now larger.*

---

## Author Comment (AC3) · 25 Jul 2018

**We would like to thank the reviewer for the evaluation of our study and the constructive comments that helped us to improve the manuscript. Please find below the reviewer's comments in black font and the author's response in blue font.**

**Responses to Reviewer 3**

*The two-way coupling between a regional climate model and an ice sheet model is an important development that marks a clear step forward to improve the projections of the future contribution of the Greenland ice sheet to sea-level change. The manuscript compares results of the two-way coupling to former methods of representing the interactions between ice sheet and atmosphere and comes to important conclusions concerning the errors implicit to those simpler approaches. The manuscript is of clear interest to the readers of The Cryosphere but still needs to be substantially improved before being acceptable for publication. I recommend major revisions along the comments outlined below.*

*General comments*

*The language of the manuscript needs substantial improvement, because many formulations give rise to misinterpretations of the scientific content. While many mistakes could clearly be avoided with a better command of the English language, a large number of typographical errors and mistakes in the referencing suggest that the authors could have made a better effort to deliver a readable manuscript for the review process.*

We apologize for the language of the submitted manuscript and the large number of typographical errors. In the revised version, we did our best to avoid English mistakes and typographical errors. We hope that the new version has become more readable.

*The text itself reveals that the models are not actually fully coupled (use of anomaly method) and also gives indications why a full coupling is so much more difficult to achieve. I suggest to adjust the title and modify other occurrences of "fully coupled" in the text to "two-way" coupled to take this into account. A discussion item on this point and next steps that need to follow to work towards truly full coupling between RCMs and ISMs should be included.*

Following your recommendation, we changed the title of the manuscript by replacing "fully coupled" with "coupled". Now, the title is "Assessment of the Greenland ice sheet atmosphere feedbacks for the next century with a regional atmospheric model coupled to an ice sheet model". For consistency reasons, we also removed the other occurrences of "fully coupled" in the text. The "fully coupled" experiment (in the first version of the manuscript) is now simply referred to as the "two-way" experiment (2W). In the discussion section (i.e. section 5), we also discussed the limits of this experiment with respect to a real fully coupling method between RCM and ISM.

*The ice sheet initialisation procedure is somewhat non-standard and therefore requires a much better explanation. As it is heavily based on former work that is in part not well documented, an additional effort is required to describe the method in a way reproducible for other modellers.*

Since the submission of the present paper, the ice sheet initialisation procedure used in this study has been published in *Geoscientific Model Development Discussions* (see https://doi.org/10.5194/gmd-2017-322). The GMDD paper describes in detail the different steps of the method, its sensitivity to various parameters as well as its limitations. We have therefore simplified the description of the ISM initialisation method in the present paper (Section 2.2.2) by only providing the basic principles of the procedure, and we propose to the readers willing to focus on the spin-up procedure to focus on the GMDD paper. However, in our response to reviewers, we tried to reply as clearly as possible to all questions related to the spin-up procedure.

*Finally, the evaluation of the method appears to be based on an experiment that is not closely related to the model state actually used for the projections, which may be possible to resolve with an additional control experiment.*

We apologize for this misunderstanding. The evaluation of the initialisation procedure is actually based on a 2000-year forward control experiment with constant forcing. In the revised paper, we have clarified the points related to this control experiment. Below, we also provide a detailed answer to the comments associated with this experiment.

*The thermodynamic aspect of the model is not well represented, arguably because it plays a minor role for the present work. Nevertheless, substantial computing time is spent during initialisation to equilibrate the temperature and the role of bottom and surface boundary conditions is mentioned. Therefore, the model description in 2.2 requires at least a short description of this model component.*

Following your recommendation, a short description of the thermodynamic aspect of the ice-sheet model has been added in section 2.2 with the following paragraph:

"*Basal melting occurs when the basal temperature is at the pressure melting point. The ice temperature plays a crucial role in the dynamics of the ice sheet because it also affects the viscosity, and thus the ice flow in the entire ice column (Ritz et al., 1997, 2001). In turn, heat released by internal ice deformation and basal dragging over the bedrock modifies the temperature. The temperature field is computed by solving a time-dependent heat equation both in the ice and in the bedrock accounting for advection and vertical diffusion processes. At the surface, the boundary condition is provided by the prescribed surface temperature. At the base of the ice sheet, the boundary condition is given either by the geothermal heat flux or by the temperature melting point at the ice-bed interface.*"

*The experiment names are not specific enough and should be improved. For my understanding, what is presented as the method "no coupling" is in fact a one-way coupling, where the ice sheet is responding to changes computed by the RCM (with "no feedback"). 2-W is correctly described, but 1-W is somewhere between one-way and two-way coupling because it parameterises the feedback. Maybe you could use "no feedback", "parameterised feedback" and "two-way" instead.*

We followed your recommendation and changed the name of the experiments referred to as NC and 1W in the first version of the manuscript. These experiments are now referred to as NF (for "No Feedback") and PF (for "Parameterised Feedbacks"). NF corresponds to the experiment in which GRISLI is forced by the MAR climate, and PF corresponds to the experiment in which both SMB and ST fields simulated by MAR are corrected to account for topography changes simulated by GRISLI. In our revised manuscript the name of the two-way coupling method (2W) has remained unchanged.

*The most important question in the comparison of results after 100 years and after 150 years is left open: why does the behaviour of 2-W suddenly change around 2010. For this it may be instructive to also look more detailed at around 2060, where a similar shift is possibly visible.*
In the first version of the manuscript we insisted on the behaviour of the 2W experiment after year 2100. However, a closer examination of the results clearly shows that the evolution of the Greenland ice sheet to sea-level rise diverge from one experiment to the other as soon as 2025-2030, namely only a few years after GrIS-atmosphere feedbacks are accounted for in the 2W experiment and in the parameterized feedback experiments (SMB-elevation feedbacks in this case). As a result, while the effect of the feedbacks on sea-level rise becomes significant by the end of the 21$^{st}$ century, it starts to operate much earlier and is amplified over time. To illustrate this point, we added a –zoom-panel in the new Figure 12 displaying the evolution of the anomalies (2W-NF, PF-NF and 2W – PF) of the GrIS contribution to SLR.

*Otherwise, I find the comparison redundant because the bottom line in most cases is 'like after 100 years, only stronger'.*
We fully agree with this remark. In the revised version of the manuscript, we mostly present the results at the end of the simulations (i.e. 2150) and we only discuss the temporal evolution, including the results by 2100, only in Sect. 4.4.

*The integration of SMB anomalies already discussed in the manuscript could be added as an additional experiment, possible even two, if masking would be additionally taken into account. This would facilitate the comparison and place the discussion of the effect of masking on firmer ground.*
Estimates of sea-level rise from the time integral of SMB anomalies were already discussed in the submitted manuscript. In the revised version they are referred to as $SMB_{MSK-NF}$ and $SMB_{MSK-2W}$. Both are based on the SMB outputs from the NF experiment (at the MAR resolution), but the time integral of SMB anomalies is made either on the fixed present-day ice-sheet mask ($SMB_{MSK-NF}$) or on the time variable ice-sheet mask simulated in the 2W experiment. The results are discussed at the end of section 4.4. However, since these estimates are inferred from diagnostics of already performed experiments (i.e. NF and 2W), we think it is not appropriate to present them as additional experiments.

*Title I would argue that the models are not "fully", but rather "two-way" coupled because an intermediate down-scaling step is necessary and, more importantly, an anomaly method is used.*
To avoid any confusion, we changed the title in:

**"**Assessment of the Greenland ice sheet - atmosphere feedbacks for the next century with a regional atmospheric model coupled to an ice sheet model".

*P1.L1 Better "the projected sea-level contribution from the \*Greenland ice sheet\*". Also mention a typical time scale here to make clear this is about the centennial time-scale.*
The first sentence now reads as:
*"In the context of global warming, a growing attention is paid to the evolution of the Greenland ice sheet (GrIS) and its contribution to sea-level rise at the centennial time scale".*
In the main text we use *"the GrIS contribution to sea-level rise"* or, following your suggestion, *"the projected sea-level rise contribution from the Greenland ice sheet"*.

*P1.L2 Be more precise about the mechanisms and feedback(s). The next sentence ("these feedback\*s\*") suggests that "temperature and surface mass balance – elevation feedback" refers to at least two feedbacks. What are these precisely? "surface mass balance – elevation feedback" is clear, but what is the role of temperature? Note also that melting is clearly related to temperature increase, but the SMB is ultimately controlled by the energy balance.*
The abstract has been extensively modified. The sentence you mentioned has been changes in:
*"Atmosphere-GrIS interactions, such as the temperature-elevation and the albedo feedbacks have the potential to modify the surface energy balance and thus to impact the GrIS surface mass balance (SMB). In turn, changes in the geometrical features of the ice sheet may alter both the climate and the ice dynamics governing the ice sheet evolution".*

*P1.L5 A bit confusing to mention start date as 2020. It is understood later that before 2020 elevation changes are considered too small to make a difference. But at this place it may be better to give the period of the entire simulation (2006 - 2150). Note also that the RCP is not defined beyond 2100, so it is better to mention "prolonged RCP 8.5 scenario".*
Recommendation followed.

*P1.L5 It seems confusing to call this simple method "no coupling", since it represents a one-way coupling. See also general comment on naming the experiments.*
As advised we have change the name of the experiment: No coupling (NC) becomes No Feedbacks (NF) experiment and one-way coupling (1-W) becomes Parameterised Feedbacks (PF) experiment.

*P1.L6 Could mention that this one-way coupling methods attempts to incorporate or parametrise two-way interaction. It represents an intermediate method between one-way and two-way coupling. See also general comment on naming the experiments.*
In the abstract, we first present the two-way coupled approach, and then the one-way coupling experiment (i.e. NF). The parameterised feedback experiment is then defined as an *"alternative one-way coupling approach in which the elevation changes feedbacks are parameterised in the ice-sheet model"*.

*P1.L7 I suggest to omit "offline". The correction may be offline to MAR, but it is online to the ice sheet model, as the correction is updated every time step and dependent on the current ice sheet*

*elevation. Could add what is happening with the extent, since it has been explicitly mentioned for the former method.*

We agree, it's offline to MAR but not for GRISLI. The ice sheet extent is not updated in the one way parameterised coupling method (PF). Only ice sheet topography changes computed by GRISLI relative to the observations are used to correct the SMB fields.

In the revised version of the manuscript we do not mention explicitly what happens with the ice sheet extent in the PF experiment, but we explain that only the surface mass balance - elevation feedbacks are parameterised, which implies that the ice-sheet extent in the atmospheric model is kept constant as in the NF experiment.

*P1.L9 Clearer to replace "ice sheet elevation feedback" by "surface mass balance – elevation feedback".*

"Ice-sheet elevation feedback" has been removed from the entire text in favour of SMB-elevation feedback, melt-elevation feedback or temperature-elevation feedback, depending on the context.

*P1.L9 Maybe ", the one-way and two-way coupling methods ..." since the amplification occurs in both cases.*

We agree. In the revised manuscript, this part of the abstract has been completely reorganised but we paid attention to make clear that SMB-elevation feedbacks are amplified over time both in the PF and the 2W experiments.

*P1.L11 Some ice sheet margins are not in the coastal region. Replace by "ice sheet margins" or similar. This should be followed throughout the document for other occurrences.*
Recommendation followed.

*P1.L15 "52 400 km^2 smaller"*
In the revised version of the manuscript, we discuss the relative changes (in %) in ablation area (rather than changes in ice-sheet extent) between NF vs 2W and PF vs. 2W.

*P1.L16 "fixed ice sheet mask"*
OK, modified

*P1.L20 "always" is only true for the end of the simulation. In the first decades or so the volume loss difference cannot be significant. Maybe give an estimate for a time scale where this is true similar to the comparison one-way vs. two-way.*
In the revised version of the abstract, we just mention that the effect of feedbacks is amplified over time. However, in Section 4.4, we specify that the feedbacks make the three simulations diverging from each other only a few years after taking into account the feedbacks (i.e. after 2020). This is illustrated in Figure 12. However, we also mention that the effect of the feedbacks becomes significant only after the end of the 21$^{st}$ century.

*P2.L6 "the ablation" (singular) <–> "are processes" (plural). Reformulate*
OK, this has been reformulated.

*P2.L6 Some risk for confusion here. It is a bit simplistic for a paper discussing an RCM as an important component to reduce the interaction to changes in SMB and temperature. It is understood that these are the two variables used to force the ice sheet model, but a bit more detail is required. How does the change of ice extent change the albedo and therefore the energy balance?*

In the revised version, our arguments have been a bit more developed. The first paragraph of the Introduction has been modified as follows:

*"The evolution of the Greenland ice sheet (GrIS) is governed by variations of ice dynamics and surface mass balance (SMB), the latter being defined as the difference between snow accumulation, further transformed into ice, and ablation processes (i.e. surface melting and sublimation). While surface melting strongly depends on the surface energy balance, snowfall is primarily controlled by atmospheric conditions (wind, humidity content, cloudiness…). However, various feedbacks between the atmosphere and the GrIS may lead to SMB variations that can therefore directly affect the GrIS total mass by impacting its surface characteristics, such as ice extent and thickness, with potential consequences on ice dynamics. These changes may in turn alter both local and global climate. As an example, changes in near-surface temperature and surface energy balance may occur in response to changes in orography (temperature-elevation feedback) or in ice-covered area (albedo feedback; see Vizcaino et al., 2008, 2015; Lunt et al. 2004). On the other hand, topography changes may alter the atmospheric circulation patterns (Doyle and Shapiro, 1999, Petersen et al., 2003, Moore and Refrew, 2005) causing changes in heat and humidity transports".*

*Does temperature enter the correction method and how?*

Yes, both ST and SMB, used as forcings of the ice-sheet model are corrected in the PF experiment, using the Franco's et al. (2012) method. In the 2W experiment, they are explicitly computed as a function of the evolving topography (computed by GRISLI), following the same procedure as in the PF experiment.

*OK, accumulation and ablation are sensitive to ST, but why and how?*

Processes have been clarified in the modified paragraph reported above

*Also, what is the role of ST other than its influence on SMB, as boundary condition to ice thermodynamics? Does it have an impact on the simulations at all (I don't expect it, but would be good to say something about why not and being able to exclude it).*

The surface temperature (ST) applied as a boundary condition of the ISM allows to compute the vertical temperature profile (i.e. the surface conditions diffuse from the surface to the base of the ice sheet and modify the ice flow by changing the viscosity of the ice). Using ST as a climate forcing is therefore a pre-requisite to run the ice-sheet model. However, at the century timescale, ST has not time enough to diffuse farther than the surface layer. Thus, in the present study, changes in ST during the 150-yr experiment have only a very limited impact on ice dynamics.

*P2.L6 Maybe already intended, but make really clear that the changes in ST have no direct effect on thickness volume and extent. Reformulate.*

We acknowledge that this sentence was not clear. The overall paragraph has been reformulated (see above).

P2.L9 Replace "disrupt" by "modify"
OK, modified

*P2.L19 More detail needed. Amplification of mass loss by what process under what forcing and compared to what other (control) experiment?*
We clarified these points in the revised version:

*"Compared to a control experiment in which the ISM is forced off-line by the atmospheric model run with the fixed present-day GrIS topography, they found an amplification of ice mass loss of 8–11 % and 24–31 % in 2100 and AD 2300 respectively, when the elevation feedbacks are taken into account (i.e. in the coupled experiment). This results from the combination of the positive elevation-SMB feedback in low lying areas, the negative feedback related to the elevation-desertification effect in accumulation areas, and the changes of surface slopes resulting from high mass loss in ablation areas and slight snowfall increase in the accumulation zone, enhancing the ice transport from the central regions to the ice margins".*

*P2.L22 The beginning of this sentence suggests (and I agree) that increased resolution would help to improve the modelling compared to observations, while "more detailed physics" is at least for the ice sheet model typically associated with 'less approximation', i.e higher order physics. Could you add some detail to distinguish these.*
This sentence has been completely reformulated. In particular, we specified that the ISM (SICOPOLIS) used in Vizcaino et al (2015) is based on the shallow ice approximation and is therefore not able to properly capture fast flowing of outlet glaciers. As suggested by Reviewer 1, we also mentioned the study of Löfverström and Liakka (2017) who confirmed the importance of the spatial resolution in coupled climate – ice sheet experiments in a paleo-climatic context. They explain that ISM results are limited by the capacity of the climate model to simulate atmospheric temperature and precipitation at low spatial resolution as a consequence of the poorly resolved planetary waves and smooth topography.

*P2.L26 Should introduce RCMs and add references to MAR, RACMO, HIRHAM ... already here, as that is the obvious choice to increased resolution. Introducing the Franco and Edwards methods is already a step further as it is based on RCM output.*
As recommended we have firstly introduced RCMs with references for MAR (Fettweis et al. 2017), RACMO2 (Noël et al., 2015), Polar MM5 (Box et al. 2013) and HIRHAM5 (Langen et al. 2015). We then mentioned the altitude corrective methods of Franco et al. (2012) and Edwards et al. (2014b).

*P3.L3 Sentence misses references for examples of RCMs.*

We added the same references as those mentioned in our response to the comment P2.L26 (see just above).

*P3.L9 Specify again for what it is a requirement.*
The sentence has been modified as follows:
*"The second fundamental requirement to describe the interactions between atmosphere and GrIS is to represent the ice sheet topography changes in the atmospheric model by using an ISM (instead of the fixed geometry typically used) to take into account the effects of ice dynamics on the ice sheet topography changes".*

*P3.L10 Reformulate "usually used" to "typically used" or similar.*
This has been reformulated (see the sentence reported just above in our previous answer).

*P3.L11 Add reference to Goelzer et al. 2017 here, since it is specifically on GrIS models.*
Sorry for this omission. The reference has been added.

*P3.L18 Remove "high resolution" or specify explicitly at what resolution GRISLI is run.*
We removed "high resolution" from the sentence and specified at what resolution MAR and GRISLI are run in section 2.

P3.L21 "two-way"
OK modified

*P3.L25 I would consider the three methods part of the experimental setup and therefore name initialisation and experimental setup first.*
The paragraph describing the organisation of the paper has been reformulated according to the new structure of our revised version. Section 2 (entitled Models) describes the atmospheric and the ice-sheet models together with their respective spin-up procedures and boundary conditions. Section 3 (entitled Coupling methods) is now focussed on the description of the three coupling methods.

*P4.L4 "developed". Correct also throughout the manuscript.*
OK corrected everywhere

*P4.L4 "SISVAT" requires a reference and description of the acronym.*
OK specified.

*P4.L16 "ice albedo that has been improved by parametrising the impact of melt ponds on the albedo."*
OK corrected.

*P4.L19 Replace "provided by" by "taken from"*
OK corrected

*P4.L20 "forced with 6-hourly atmospheric fields". See also P8.L6*
OK corrected.

*P4.L24 Remove "forcing"*
OK removed

*P4.L27 Suggest reformulation to "... because it has been shown by Fettweis et al. (2013), to be the best choice from the CMIP5 data-base to reproduce the present-dayclimate compared to results of MAR forced by reanalyses."*
OK, modified.

*P5.L1 Heading "Climate model initialisation and experiment"*

Section 2 (and its related subsections) has been reorganised following the recommendations of the three reviewers and heading is now "Models" This section is still divided in two subsections 2.1 and 2.2 devoted to the description of the MAR and GRISLI models respectively. Section 3 is devoted on the description of the three "coupling" experiments.

*P5.L2 What is the difference between "spurious drifts" and "unwanted trends" or are they one and the same? Reformulate.*
We apologise for this misunderstanding. We used two different expressions to deal with "unwanted trends". The sentence has been reformulated as:
*"Before starting our experiments, MAR needs to be properly initialised to limit unwanted trends in the results".*

*P5.L3 Replace "SISVAT requires more than 6 years", by "SISVAT requires less than 7 years" to make clear that the chosen 7-year period is long enough. Or otherwise explain why 7 years is considered OK.*
OK modified.

*P5.L5 Replace "provided by" by "taken from". Add explanation how the data was interpolated to the coarse MAR grid.*
We replaced "provided" by "taken from" as suggested.
In the revised version, we specified that the GrIS topography from the Bamber et al. (2013) dataset is aggregated on the MAR grid.

*P5.L5 Be consistent in if SISVAT is written in italic or not.*
OK corrected

*P5.L6 Replace "following year 1976" by "from 1977 onward".*
Sorry for this misunderstanding. In the revised version we clarified that MAR is initialized with MIROC5 climatic fields from 1970 to 1975 included. The MAR simulations start in 1976, but the results presented in this paper are for the period 2000-2150. This has been clarified in the revised manuscript. Therefore we changed the sentence in:

*"Here, MAR is initialised with the atmospheric forcing fields from MIROC5 from 1970 until 1975 and the MAR simulations start in 1976. However, in this paper, the MAR results will be analysed for the period spanning from years 2000 to 2150".*

*P5.L10 Need to explain in more detail why 2095 can be considered representative for the 2090s. Is it e.g. the year that is closest to the decadal mean? Are trends so linear that the middle of the decade are representative for the average? Typically one would use the decadal mean to represent the long-term average and not one individual year, unless it doesn't matter for some reason.*

We chose to force MAR with the 2095 climate from 2101 to 2150 because, averaged over the entire GrIS, the 2095 climate is one the closest to the decadal 2090-2100 mean climate. We acknowledge that, in the absence of a MIROC5 simulation run under a prolonged RCP8.5 scenario it would have been more appropriate to repeat the ten years (2090 -2100) until 2150, but it would have been more complex to set up.

*P5.L12 Better to omit "coupled" here, since it is not clear what is coupled to what and it is further detailed later.*

OK, "coupled" has been removed.

*P5.L15 "the northern hemisphere ice sheet\*s\* (NH references) and the Greenland ice sheet (GrIS references)". or "the northern hemisphere ice sheet\*s\* and the Greenland ice sheet (all references)".*

OK modified according to the suggestion.

*P5.L16 "... covering Greenland with ...", since the coverage extends outside of the ice sheet mask. Add information about the vertical.*

We have also specified that GRISLI has 21 vertical evenly spaced levels.

*P5.L17 Need to specify what "hybrid" means.*

The word "hydrid" has been removed from this part of the text and introduced after having explained the basic principles of both the shallow-ice and the shallow-shelf approximations: *"Using a hybrid model (i.e. based on both SIA and SSA approximations) allows to better represent the different deformation regimes found in an ice sheet".*

*P5.L19 Need to add explanation on the thermodynamic aspect of the model.*

We added new information on the thermodynamic aspects:

*"Basal melting occurs when the basal temperature is at the pressure melting point. The ice temperature plays a crucial role in the dynamics of the ice sheet because it also affects the viscosity, and thus the ice flow in the entire ice column (Ritz et al., 1997, 2001). In turn, heat released by internal ice deformation and basal dragging over the bedrock modifies the temperature. The temperature field is computed by solving a time-dependent heat equation both in the ice and in the bedrock accounting for advection and vertical diffusion processes. At the surface, the boundary condition is provided by the prescribed surface temperature. At the*

*base of the ice sheet, the boundary condition is given either by the geothermal heat flux or by the temperature melting point at the ice bed interface".*

*See also P5.L27 SIA velocity is even stronger controlled by ice thickness.*
Both the ice surface slopes and the ice thickness occur in the computation of the SIA and the SSA velocity with the same exponent. We therefore modified the sentence as:
*"The ice thickness and the ice-sheet surface slopes control the SIA and the SSA velocity components, but the SSA is also governed by basal dragging"*

*P5.L28 "SSA component is mainly controlled by the ice flux" is confusing because ice flux is velocity x ice thickness. Clarify!*
This has been clarified and corrected in the revised version (see our previous response).

*P5.L29 "rheologies" is the wrong term here. Maybe "deformation regimes".*
Yes, you are right. We replaced by "deformation regimes"

*P5.L30 Replace "ice melting point" by "pressure melting point"*
OK, corrected

*P6.L3 Replace "floating criterion" by "floatation criterion"*
Ok, corrected

*P6.L4 What does "characteristics of the Greenland bedrock" mean? Explain*
We acknowledge that this expression was too vague. It referred to the nature of the bedrock (i.e. water-saturated sediment or not). However, this part of the manuscript has been re-written and the sentence has been deleted.

*P6.L5 Does that mean the enhancement factor differs for different regions? Explain.*
Alike most ice-sheet models, GRISLI considers the ice as a non-Newtonian viscous fluid that follows the Glen's flow law (with the coefficient n generally fixed to 3). However, a particularity a GRISLI is also to account for a Newtonian contribution (i.e n =1) for low deformation rates leading to a polynomial Glen's flow law in which we apply an enhancement factor in SIA areas to favour longitudinal deformations. In addition, a fixed ratio between the SIA and the SSA enhancement factor is used. The polynomial Glen's flow law is expressed as :

$$\frac{1}{\eta} = (E_1 \, B_1 \, (T) + E_3 \, B_3 \, (T) \cdot \tau^2 \,) \tau'_{ij}$$

where $\eta$ is the ice viscosity, $\tau$ is the shear stress tensor and $\tau'_{ij}$ is the deviatoric stress tensor, B1(T) and B3(T) are temperature-dependent coefficients following and Arrhenius equation for coefficients n= 1 and n= 3 respectively and E1 and E3 are the corresponding enhancement factors. As a result there are theoretically four enhancement factors (2 for the SIA component of the velocity with n= 1 and n=3 and 2 for the SSA component of the velocity). In practice, for the simulations presented in this paper, we used $E_{1\_SIA} = E_{3\_SIA} = 1$ and $E_{i\_SIA}/E_{i\_SSA} = 0.125$.

After a careful examination of the paper and reviewers comments, we do believe that any mention to the enhancement factor does not provide any added value to the manuscript. We therefore removed the corresponding sentence.

*P6.L6 "ice loading changes"*
OK, corrected.

*P6.L7 Add a reference for the used isostatic model.*
OK, Le Meur et al. (1996) added for the ELRA model.

*P6.L8 Add a reference describing the thermodynamic model.*
As previously mentioned, we added a new paragraph to describe the thermodynamic aspects of the GRISLI model and added the references Ritz et al (1997, 2001).

*P6.L10 This whole paragraph needs to be reworked. Be more specific. What is considered a boundary condition, what is input data and what is considered a forcing? What variables are concerned for ice flow, ice thermodynamics and isostasy?*
We acknowledge that this paragraph was very confusing. In the revised version, subsections 2.2.1 is now devoted to the description of the GRISLI ice-sheet model and section 2.2.2 is focused on the spin-up procedure. Following your recommendation, the paragraph concerning climate forcing, initial conditions and input data has been entirely re-written. Now it reads as: *"The climatic forcing is given by the mean annual SMB and the mean annual ST. Because seasonal variations of surface temperature are rapidly dampened, ST is considered as a good approximation of the bottom snowpack temperature. The initial GrIS surface and bedrock topographies come from Bamber et al. (2013) and the geothermal heat flux is taken from Fox Maule et al. (2009)".*

*P6.L11 Is there are a difference between "The annual mean near surface air temperature" and ST? If yes, explain, if not, use TS instead.*
No, there is no difference: ST represents the mean annual near surface air temperature. This has been clarified in the new version of the manuscript.

*P6.L13 What data are these 'boundary conditions' and which variables are taken from which data set? Surface elevation, bedrock elevation and ice thickness are not boundary conditions to the equations that GRISLI solves in the proper sense. You could call this "input data" instead.*
As mentioned above, we clarified the text.

*P6.L14 "The climatic forcings". Say what they are! TS and SMB?*
The climatic fields used as GRISLI forcings are the SMB and the ST. This has been clearly specified in the new version.

*P6.L15 If basal drag were a boundary condition, it could hardly be computed. Reformulate to make this clearer.*

The basal drag coefficient is only adjusted during the initialization. In forward experiments, it remains constant through time and its spatial distribution is fixed to that obtained at the end of the initialisation. It can be thus considered as an input data, at least for transient experiments.

*P6.L18 Heading "Ice sheet model initialisation and experiments"*
As previously mentioned this sub-section has been canceled and the text has been moved to the main section 2.2

*P6.L19 The motivation is not quite correct. I would argue that to equilibrate the model to a steady state is not a necessity given the approximations, but rather a choice. One could envision a transient spinup as initialisation with the exact same model.*
Yes, we agree with this comment. It seems more appropriate to only deal with the long time-scale response of the ice sheet. Our motivation has been reformulated as: *"Due to the long time scale response of the ice sheet to a given climate forcing, a proper initialisation of the model is required before performing forward experiments"*

*P6.L20 Again, more precision needed. What the ice sheet model equilibrates to is rather the climate forcing held constant for this particular initial steady state experiment.*
The text has been modified as follows:
*"the aim of the initialisation is to start the simulations from a present-day ice sheet geometry as close as possible to the observed one while ensuring consistency between internal properties of the ice-sheet (e.g. basal sliding velocities and vertical profile of temperature) with the climate forcing".*

*P6.L20 Replace "sensitivity" by "forced" or "forward".*
We replaced "sensitivity" by "forward".

*P6.L20 Reference Le clec'h et al. (in prep) is not in the reference list. If you are referring to the present manuscript, say that instead of using an external reference.*
No, we did not refer to the present manuscript. We simply omitted to add the reference *Le clec'h et al. (in prep)* in the reference list. This paper describes in details the initialization procedure. Since the submission of the present manuscript, *Le clec'h et al. (in prep)* has been published in GMDD. In the following (as well as in the revised manuscript) it is referred to as *Le clec'h et al. (2018)*. As a result, in the revised version of the present manuscript, this reference appears as *Le clec'h et al. (2018)*. Moreover, we made the choice to only present the basic principles of the initialisation procedure to avoid redundancy with the GMDD paper.

*P6.L22 Replace "avoid" by "reduce", since the method is not perfect. Also I'd suggest the formulation ". reduce an initial adjustment of the model during the first years of the simulation due to factors not related to the climate forcing alone." or similar.*
In the revised version, we no longer speak about *"an initial adjustment of the model". Instead, we explain that the aim of the initialization procedure is to "reduce the difference between the observed and the simulated ice thickness".*

*P6.L25 Reformulate "just over the bedrock". Maybe "basal conditions".*
We agree with this suggestion. However, as a result of the simplified description of the initialisation procedure, the corresponding sentence has been removed in the new version of our manuscript.

*P6.L26 If basal conditions are "likely to change in time" your method to define spatially variable but \*temporally fixed\* basal drag coefficient could never be successful. Should add here that your method assumes them to be constant over the 150 years of your experiment.*
This is actually a limitation of the method and this is why it cannot be applied for long-term transient experiments. However, over the 150 years of the experiments, we assume that basal conditions do not change so much and that the best guess for the basal drag coefficient obtained at the end of the spin-up procedure is a good approximation of the basal dragging at the century time scale. Because of the simplified description of the spin-up procedure, this part of the text has been removed. However, we specified in the revised manuscript that "$\beta$ is a time constant but spatially variable basal drag coefficient".

*P6.L27 Suggest to remove sentence "As a result any error in the basal velocity computation can spread vertically in the ice and generate slowdown or acceleration of ice sheet motion." In its present form this sentence is generally true in any case and doesn't support your chose of assimilation method.*
We followed this suggestion and removed the sentence.

*P6.L29 It is not clear to me at what point in the procedure observed velocities are actually used. Which observational data set is used? Reference needed.*
The observed velocities (Joughin et al., 2010) are only used as input data for the first iteration. The actual target is to reduce as best as possible the mismatch between the observed and the simulated ice thickness.

*P6.L30 "three main steps:" Make a numbered list (possible with lists of sub-steps) to facilitate navigation of the different steps.*
We acknowledge that is part of the text was not well written and contained several misleading formulations. As explained above, the presentation of the spin-up method has been reduced to its basic principles (see section 2.2.2) in this revised paper because the full description of the method can be found in Le chlec'h et al. (2018).

*P6.L31 Replace "not necessary consistent between them" by "not necessarily mutually consistent".*
Recommendation followed.

*P6.L32 It looks to me like the first guess of basal drag mentioned here is a very good first guess and further adjustment of the basal drag coefficient is very much based on it. At any rate, a full description of the procedure used to arrive at that stage should be included, otherwise the method is not reproducible with another model (and not even with GRISLI itself). See also general comment on initialisation.*

The first guess of the basal drag coefficient comes from a preliminary version of the spin-up procedure summarized in the present paper and fully detailed in Le clec'h et al. (2018). This former procedure was set up for Ice2Sea simulations carried out for with exactly the same GRISLI model version and the same initial Greenland ice-sheet topography (Bamber et al. 2013) as those used in the present study, but with a different climate forcing, implying the need for adjusting the basal drag coefficient. Furthermore Le clec'h et al. (2018) have shown that the final value of the basal drag coefficient (i.e. used for forward experiments) obtained after the spin-up procedure is very poorly dependent on the initial guess (see Fig. 3 in the GMDD paper).

*P6.L32 Edwards et al is a multi-model intercomparison and does not give specific details on the assimilation technique for GRISLI. The model reference there is given as Quiquet et al., 2012), which does not provide information on spatially variables tuning. Again, the method to produce the first guess basal drag needs to be made transparent for other modellers to be able to reproduce the results.*

The reviewer is right concerning the reference Edwards et al (2014a). This was cited to inform the reader that the initial guess of the basal drag coefficient was coming from the Ice2Sea project. We acknowledge this was not appropriate since this paper does not contain any detail about the assimilation technique. In the revised manuscript, we provide a piece of information about the method used to obtain this first guess (see our previous response).

*P6.L32 "surface and bottom". I think you mean surface elevation and bedrock topography. Be more specific!*

This is right. This part of the text has been reformulated (see our response related to the main steps of the initialization procedure).

*P7.L1 "vertical fields" Be more specific!*

We dealt with the vertical and temperature profiles. Once again, this part has been reformulated.

*P7.L2 If I understand correctly, you calculate something here in the first step to be used in the second step. Maybe you should say that. Confusing to mention already here "to have an ice flux as close as possible to observation" when diagnostically calculating something here will not have any influence on the match of the ice flux with observations in this step. This could be mentioned in the second step or as a general motivation for your method before.*

Completely reformulated to make clearer the description. Indeed, the basal drag coefficient computed during the 1$^{st}$ step (see new description) is used in the 2$^{nd}$ step. This has been specified in the new version.

*P7.L3 Not clear to me how to derive a factor (a/b) from a difference (H1-H0). Please provide an equation or better explanation what the underlying idea is, what is done here, and how it is calculated?*

The revised manuscript includes equations supporting the spin-up description. We hope this will help to avoid any ambiguities.

*P7.L4 You are mixing topography differences and ice thickness differences. Possibly similar or identical in absence of bedrock adjustment, but is it necessary to distinguish them?*

Since bedrock adjustment is negligible in the present study (owing to the addressed time scales), surface elevation differences and ice thickness differences are supposed to be very similar. However, to avoid any confusion, we only use the term "ice thickness" throughout the revised manuscript.

*P7.L4 Again "the factor allows to decrease (resp. increase) the surface ice velocity" is confusing, because this is not happening in this first step. Also "If \*locally\* the topography difference \*is\* positive ..."*

The previous description was misleading. In the revised manuscript, the first step consists in computing a new value of the basal drag coefficient from the value obtained at the previous iteration and from the ratio of the sliding velocity over the corrected sliding velocity. This ratio represents the corrective factor to reduce the mismatch between observed and simulated ice thicknesses. The way the corrected sliding velocity is computed is now fully described in the revised manuscript. For the very first iteration (i.e. just after the 5-year relaxation), the first step is skipped because there is no difference between observed and simulated ice thicknesses and the procedure starts at the second step. This has been also specified in the new version of the paper.

*P7.L5 How does deltaH translate into deltaV?*

The relationship between the ratio of $H^G/H^{obs}$ and the ratio of the vertically averaged velocity is given by Equation 4.

*How does the new velocity compare to observed surface velocities?*

There was a confusion in the revised manuscript when dealing with surface ice velocity. Actually, surface ice velocity have to be replaced by "vertically-averaged velocity". As a result, we do not compare the new surface ice velocities to the observed ones in the present paper. However, this comparison can be found in the GMDD paper (see Fig. 8 herein): we show that the overall patterns of the simulated ice surface velocities are generally in good agreement with observations (particularly in regions of fast ice flows), despite slight differences in the central plateau where the ice velocities are low.

*P7.L11 How is the new coefficient calculated? Explain in detail.*

The description of the method has been clarified in the revised manuscript and the way the new basal drag coefficient is calculated has been explained in detail (see the new section 2.2.2 in the revised paper).

*This is reminiscent of the method of Pollard and DeConto 2012, could you describe the similarities and differences to their approach?*

The reviewer is right. In the revised manuscript, we specified that our spin-up method is based on the same basic principles as that of Pollard and Deconto (2012) in that their basal sliding coefficient is adjusted so as to reduce the difference between simulated and observed ice-sheet

topography. We also mentioned the main differences between their method and ours. The new paragraph is:

*"Based on the same basic principles as that of Pollard and DeConto (2012), our method consists in the adjustment of the spatially-varying basal drag coefficient (and thus of the basal sliding velocities, see equation 3) so as to reduce the difference between the observed and the simulated ice thickness. However, while the study by Pollard and DeConto (2012) requires long (multi-millennial) integrations for the method to converge, we suggest instead an iterative method of short (decadal to centennial) integrations starting from the observed ice thickness".*

*Surprisingly your adjustment goes very fast (in total less than 2000 years). This makes me believe that the original basal drag was already a good guess and you only need minor adjustments. Is that correct? How different is the final basal drag field from the initial one? Can we see a figure for this comparison?*

As previously mentioned, we have shown in the GMDD paper that the convergence of our spin-up method is only poorly dependent on the choice of the original basal drag coefficient. Sensitivity tests performed with a uniform $\beta$ coefficient ($\beta=1$) and with the same spin-up parameters (i.e. 20 years for the duration of each iteration, 200 years for the free-evolving simulations and Nbcycle = 8) results in negligible differences in the final basal dragging compared to that inferred from our "standard method" (i.e. a first guess for $\beta$ coming from Ice2Sea simulations), and in an ice thickness root mean square error (+ 62 m) fully comparable to that obtained in the present study after 8 cycles (+ 63 m). These results are illustrated in Figure 3 in the GMDD paper. Hence, they are not reported in the present study.

*P7.L15 Replace "minimum gap" by "error".*
This part has been reformulated. Throughout the manuscript we use "mismatch" or "differences" between simulated and observed ice thicknesses.

*P7.L15 You additionally need to convince the reader here that this method is optimal in the parameter choices (adjustment time 20 y, relaxation time 200 years) and to make clear in how far the results are (not) dependent on these choices.*
The results (in terms of time of convergence and ice thickness root mean square error) are obviously dependent on the choice of the adjustment and relaxation time and of the number of cycles. As explained below (see response concerning the stopping criterion) they have been chosen to minimise the ice thickness RMSE. This has been mentioned in the revised paper. Numerous sensitivity studies with different sets of parameters have also been carried out and presented in Le clec'h et al. (2018). In the revised paper, we specify that: *"The overall process is stopped when the ice thickness root mean square error is not significantly improved. This ensures a good compromise between the reduction of the mismatch between observed and simulated ice thickness and the rapidity of the convergence of the spin-up method. In the present paper, the number of cycles that provides the best fit with observations (RMSE = + 63 m) is $Nb_{cycle} = 8$".*

*P7.L20 After each step you have "a new set of initial conditions" for the next step. Maybe better to only name the final result of your initialisation your initial state as input for the forward experiments.*
Completely reformulated

*P7.L21 After 30 kyr, T is in equilibrium with the climate \*and with the fixed geometry\*, but not the other way around. In the next step of retuning basal drag, you further evolve the geometry and the ice temperature? Could you quantify, give an estimate how far from equilibrium you are now? Why could you not run (part of the initialisation) with freely evolving temperature?*
We apologise for the confusion. Actually, there is no temperature equilibrium in the spin-up procedure used for the MAR-GRISLI experiments. However, this issue has been examined in the GMDD paper in which the 30,000-yr temperature equilibrium run appears as a sensitivity experiment. In the present paper, the temperature evolves freely at any stage of the initialisation procedure. Initial conditions inferred from the relaxation run are just restored before starting a new iteration.

*What is your stopping criterium at this point and the reason for not iterating further?*
Our target is to obtain the mininum ice thickness root mean square error (here RMSE = + 63 m). We stopped the iterations when the RMSE is not significantly improved (here after 8 cycles). This ensures a good compromise between the reduction of the mismatch between observed and simulated ice thickness and the rapidity of the convergence of the spin-up method. This has been clearly explained in the revised manuscript.

*P7.L27 It is not clear why evaluation of the initial state should be based on an experiment which includes further relaxation steps. The control experiment that offers itself naturally and should be used for that purpose is just running the model after step 3 forward with constant forcing. This would give a good indication of the match with observations (at t=0 or t=25) and the remaining model drift (after 150 years), since this is the model state actually used as initial state for the forward experiments. It anyhow seems strange to impose the observed geometry, when the model has been relaxed to a different geometry in step 3.*
After the last step 2 (i.e. after the end of the 8[th] cycle), a 2000-yr free evolving GRISLI run is carried out under conditions identical to those used in step 2 in terms of climate forcing, initial vertical temperatures and velocity profiles. As such, the value of the basal drag coefficient is that obtained at the end of the 1[st] step of the 8[th] cycle. This has been specified in the revised manuscript.

*P7.L32 It is a bit unusual to specify errors in ice thickness as median values, given that errors locally could be positive or negative. Why not specify the absolute error or root mean squared error augmented with the quantiles given already. A map of the mismatch with observations should be given (possibly in the appendix), but then for the model state after step 3, which is assumed as the initial state for the projections.*
We agree with you that median computed from ice thickness errors with respect to observations is not always informative because of both positive and negative values. For this reason, the description of ice thickness changes has now been given as a function of different

surface elevations (see section 4.1.2) in order to aggregate regions that present similar tendencies.

As explained just above, the 2000-yr GRISLI simulation has been performed to reduce the ice volume drift. The state obtained at the end of this run is used as initial state of the forward experiments. In the revised version we mention both the new ice thickness RMSE (= + 132 m), which is different from that obtained at the end of the last step 2 (+ 63 m), the 5th and the 95th quantiles, and also the sea-level equivalent model drift ($\sim 10^{-5}$ mm yr$^{-1}$). We added in the Supplementary Materials a figure (Fig. S1) showing the differences between the observed and the GRISLI topographies, with the GRISLI topography taken at the end of the 2000-yr relaxation run.

*P8.L1 The model state that has been compared to other models in the initMIP exercise appears to be different from the state used in the forward experiments, because it includes re-imposing the observed geometry and additional relaxation for 2000 years. This should be made very clear, especially in light of the claim that the model is one of the best in the model comparison.*

*This statement in particular requires further qualification and needs to specify what criteria to consider, since the Goelzer et al paper does not provide any explicit ranking of the models and goes into length about how different criteria for evaluating models are not independent. Please use such community efforts to improve your model, but don't misuse them to gain credibility for your model.*

To avoid confusion and misleading interpretations we removed the comparison to other models in the initMIP exercise.

*P8.L5 SLR contribution as the most abstract change could be named last.*

Corrected in the text following the recommendation. The new sentence reads as: *"The aim of this study is to assess to what extent accounting for the atmosphere-GrIS interactions influences the GrIS evolution in terms of changes in SMB, ST, ice thickness and SLR".*

*P8.L16 Is the elevation difference used for the correction calculated between Bamber (at 5 km) and Bamber (at 25 km) bi-linearly interpolated to 5 km? Please describe.*

The horizontal interpolation is made using an inverse distance weighting method, as it is now specified in the revised manuscript. Moreover, to account for the differences in surface elevations between the 25 and 5 km Bamber et al. (2013) topographies, we also apply a vertical correction following Franco et al. (2012) who derived a local vertical gradient of each SMB component as a function of altitude.

*P8.L21 This seems to imply that at least until 2020, NC is an appropriate approximation to the full problem. This should enter the discussion and the abstract, following an earlier comment. Is there any reason why the modification starts at 2020 and not at 2000? It would seem like a cleaner comparison to start the interaction from the moment it is possible (i.e. 2000).*

For all the experiments, the "coupling" starts in 2020 when the SMB simulated by MAR at a given time is enough different from the SMB simulated at the beginning of the simulation to induce significant changes in the GrIS topography. Thus, in the PF and 2W experiments, GRISLI is forced by MAR outputs from 2005 to 2020, following the same procedure as in the NF

experiment. It would have been possible to start the coupling in 2000 or 2005, but the results would have been similar to those presented here as the SMB changes through 2005-2020 do not produce any significant topography changes in GRISLI. This has been specified in the revised abstract and in the main text.

*P8.L23 Another "coupling method" that is already discussed in the text and could be formally listed here as well is the one where MAR SMB anomalies alone are used to generate a changing ice sheet geometry (in the absence of an ice sheet model). This experiment can be performed with or without taking into account the surface elevation - SMB feedback and with or without fixed ice sheet extent.*

As previously specified in our response to the General Comment", the GrIS sea-level rise estimated from SMB integrations over fixed and time variable ice sheet masks have been discussed in Section 4.4. We decided not to present them as additional experiments since the SLR estimates are inferred from diagnostics stemming from NF and 2W experiments.

*P9.L13 This section reveals that the models are not actually fully coupled and also gives indications why a full coupling is so much more difficult to achieve. See general comment.*

We agree with this comment. In the revised manuscript, we removed expressions such as "fully coupled" (see also our response to the general comment). We also explained why we have chosen the anomaly method (see section 3.2):

*"Due to the topography differences between MAR and GRISLI, this approach has been chosen to avoid large inconsistencies between the SMB and ST fields computed by MAR and the ones corrected to account for the GRISLI topography."*

In the discussion section (i.e. section 5), we also discussed the limits of this experiment with respect to a real fully coupling method between RCM and ISM:

*''A second limitation is related to the 2000-yr relaxation GRISLI experiment, run at the end of the spin-up procedure to reduce the model drift in terms of ice volume, that produces residual differences with the observed topography (Bamber et al. 2013) used in the MAR simulations. This has important consequences on the MAR simulated climate. In particular, the steeper slopes existing in the GRISLI topography (i.e. $S_{ctrl}$) tend to produce unrealistic katabatic winds. Therefore, we choose to use an anomaly method of the surface elevation onto which the SMB and ST fields are downscaled at the 5 km resolution grid (Eq. 7). The objective of this approach was first to maintain the realism of the simulated present-day climate computed on the observed topography (Bamber et al. 2013) and, secondly, to avoid inconsistencies between the climate simulated by MAR and that used to force GRISLI. However, this implies that the forcing climate is not fully consistent with the GRISLI topography. This should be taken into consideration in a future work to improve the quality of our results. As an example, a reasonable compromise to avoid the use of anomaly method would be to use the topography obtained at the end of the spin-up iterative process (rather than $S_{ctrl}$) as initial GRISLI topography to keep the mismatch with the observed topography as low as possible, and to initialise and perform MAR simulations with this spin-up topography"*

*P10.L2 The mean decrease in SMB explains the shift in the ELA not the other way around. The ELA is an abstract concept, the SMB change is 'real'.*

The sentence has been modified as:

*"The equilibrium line altitude (ELA, i.e. altitude for which SMB = 0) increases significantly between the beginning and the end of the 2W experiment, as a consequence of increased runoff for areas below 2000 m."*

*P10.L5 I am not sure reporting the changes in ice thickness changes as mean and standard deviation makes much sense, given the bipolar nature of thickening in the centre and thinning at the margins. More useful would be for me to describe the changes for specific regions.*

In order to clarify the text, the description of ice thickness changes has been given as a function of different surface elevation (see section 4.1.2). Besides the mean ice thickness anomaly values and the corresponding standard deviations, we have also reported the 5[th] and the 95[th] percentiles to indicate the range of ice thickness changes:

*"The ice thickness anomaly (Fig. 4) also presents two distinct patterns. For surface elevations higher than 2000 m in the northern part, and higher than 2500 m in the central and southern parts of the ice sheet, the ice thickness increases by +5 m on average, with the increase ranging from +1.5 m (5[th] percentile) to +17 m (95[th] percentile). On the other hand, in regions whose surface elevation is lower than 2000 m, the ice thickness decreases from -248 m (5[th] percentile) to -3 m (95[th] percentile) with a mean value equal to -100 m".*

*P10.L12 What exactly is the impact of ice temperature on ice dynamics? Are you implying that changes of the surface boundary conditions modify the temperature structure of the ice and its deformation?*

This was a shortcut. Actually, as the model was forced by a warming scenario (i.e. the RCP8.5 and extension of year 2095 to 2150), we simplified by "warming scenario" instead of simply explaining that the ice dynamics was also impacted (in addition to ice thickness). In the revised manuscript, changes in ice velocities are related to changes in ice thickness. As a result, the new sentence has been changed in:

"The ice dynamics is also impacted by changes ice sheet geometry as illustrated by the mean surface velocity anomaly (Fig. 6a)".

*P10.L13 Are you talking about velocity or velocity anomalies here? Figure 4A shows anomalies! Please clarify.*

We are talking about surface velocity anomaly. We have clarified the text (see section 4.1.3).

*P10.L15 This statement calls for a figure comparing modelled and observed velocities! Add a panel to substantiate this point.*

The panel showing the observed surface velocities has been added (see Fig. 7 in the revised manuscript).

*P10.L18 Add "in this area" after "ice velocities" and remove it in the sentence after.*

The comments related to the new figure 7 have been reorganized and the sentence you refer to has been removed from the revised manuscript. The examples of the Jakobshavn and the Kangerlussuaq glaciers are now distinguished, and the new paragraph (section 4.1.3) reads as:

*"For the Jakobshavn glacier, and for altitudes above 1500 m, the vertically-averaged ice velocities increase by more than 15 m yr$^{-1}$ (i.e. +10 %) as a result of increasing surface slopes, and slow down by more than 200 m yr$^{-1}$ (i.e. +29 %) for altitudes below 1000 m due to the decreasing ice thickness (Fig. 7c). For altitudes above 500 m, the vertically-averaged velocity is mainly driven by the SIA velocity (Figs. 7c-e). On the contrary, below 500 m, basal sliding velocities are large due to low basal drag coefficient (see Fig. 3 in Le clec'h et al., 2018) and the SSA velocity component dominates the ice flow (Figs 7c-g). However, while basal drag is lower in locations below 500 m, the ice flow is limited by the strongly reduced ice thickness (Fig. 4)".*

*"The Kangerlussuaq glacier is located in regions where the bedrock is characterised by a succession of valleys surrounded by mountains merging in a canyon where the deepest part is located 100 km away from the coast (Morlighem et al., 2017). The ice flow of the Kangerlussuaq is therefore divided in different branches with increasing ice velocities towards the ice sheet margin and becoming even larger when merging in the canyon (Fig. 7b). As for the Jakobshavn glacier, the ice flow accelerates at the end of the 2W experiment as a consequence of the increase in surface slope for high altitudes (~2000-2500 m, see Fig. 4). Conversely, a strong decrease of the ice flow is found in most of margin regions (Fig. 7d) directly related to the ice thinning (Fig. 4). Contrary to the case of the Jakobshavn glacier that presents large basal sliding velocities only below 500 m, the Kangerlussuaq shows low basal drag coefficients in the entire glacier (see Fig. 3 in Le clec'h et al. 2018) and thus the ice flow is mainly governed by the SSA component (Fig. 7h)".*

*P10.L30 "amplification of all the changes" is a bit too general here. Better "amplification of the changes".*
This has been reformulated. Following the recommendations of Reviewer 2, the Results section (Section 4) to emphasize the 2W experiments. As a consequence Section 4 has been reorganised and the changes occurring in 2100 are now discussed in Section 4.4

*P10.L31 A figure showing the absolute sea-level changes for the different experiments would be in place, possible as additional panel in figure 7.*
Following your suggestion, we added a figure showing the absolute sea-level changes (Fig. 12a in the revised manuscript). We also made a zoom-figure displaying the sea-level anomalies between 2000 and 2100 to better illustrate the divergence of the three experiments as soon as 2025-2030 (Fig. 12b).

*P11.L3 ST is already defined*
Thanks for this remark. ST is now defined once, in Section 1.

*P11.L4 Replace "is strongly colder" by "sees a strong cooling" or similar.*

This has been corrected

*P11.L10 "Thus, the \*stronger\* ST decrease in 2-W compared to NC ...", assuming there is decrease in both cases. To check also in other places that you discuss differences in changes, not changes itself.*
We made our best to remove all ambiguities related to changes and differences in changes. We hope the text is now clearer.

*P11.L10 Not sure where "the middle of the slope is". Clarify!*
We replaced "middle of the slope by "along the slope"

*P11.L14 Costal regions don't exist inland from the ice edge.*
We reformulated in "in the interior of the GrIS".

*P11.L28 Replace "SMB anomalies increases by a factor of 10" by "SMB anomalies decreases by 10 cm yr-1"*
This sentence was referred to Table 1 which does no longer appears in the revised manuscript. The new Table 1 provides values of the GrIS contribution to sea-level rise in 2050, 2100 and 2150 for the three experiments. Moreover, the section describing the SMB differences between the 2W and the NF experiments has been re-written (see Section 4.2.1)

*P12.L1 Again mixing discussion of surface elevation and ice thickness here. Revise.*
This has been revised and corrected in the entire revised manuscript

*P12.L1 Add "difference" after "surface elevation change" and reformulate to "follow the patterns of SMB anomaly differences (Fig. 6B)".*
Replaced by: *"The ice thickness anomaly pattern is essentially mimicking the SMB differences between 2W and NF (Fig. 8a)"*

*P12.L5 Do you mean lower surface temperature in 2-W is the cause for higher SMB and therefore increasing ice thickness, or is the lower surface temperature directly impacting ice thickness (i.e. not through its effect on SMB)? In the first case, lower surface temperature and its effect on SMB should be mentioned first and higher SMB as a consequence. More precision needed here.*
This part of the text has been clarified and precisions have been added:
*"The main SMB differences between both experiments, averaged over the 2140-2150 period, highlight lower SMB values in 2W compared to NF for altitudes below 2000 m, with the exception of some margin locations in the eastern part (Fig. 8a). This SMB anomaly behaviour is driven by a snowfall reduction in low altitude areas (Fig. S6) and by the runoff increase in 2W with respect to NF (Fig. S7). This increased runoff results from warmer temperatures over the whole GrIS (up to 0.8°C in the western and northern parts, Fig. 8b), except in the region at the edge of the GrIS, which sees a strong cooling (as low as -10°C, Fig. 8b). The warming can be explained by the temperature-altitude feedback being active in 2W, resulting in lower altitudes*

*(section 4.1.2 and Fig. 8c) and therefore warmer temperatures. The cooling over the very edge of the ice sheet occurs despite the ice sheet thinning over these regions. It can be explained by changes in atmospheric circulation".*

*P12.L6 "in areas of lower ST"? In my eyes, ST and ST differences (Fig. 6A) are both high and positive in regions of negative thickness anomaly. Clarify that statement.*
The ST differences between the 2W and the NF experiments are positive in most of GrIS areas. However, at the very edge of the ice sheet, these differences are negative, showing that the 2W surface temperatures are lower than the NF ones as a result of the effect of katabatic winds. In the revised manuscript we changed the ST color scale, making the negative ST differences more visible.

*P12.L11 I thought you are trying to describe here the impact on the ice thickness evolution of two-way coupling as opposed to no coupling. In this part, you however come to the impact on the atmospheric circulation (katabatic winds) and land model changes (albedo). From line 15 on, you go back again to ice dynamic changes. Could this material be better organised to avoid jumping between the different aspects?*
Following your suggestion, this part has been reorganized. In this section, we only emphasize the effect of katabatic winds. The reduction of the ice-sheet extent simulated in the 2W experiment (and thus the effect on albedo changes) is now discussed in section 4.4.

*Also, if I understand correctly, the anomalous katabatic winds created by 2-W have visible impact mainly on the narrow marginal areas of the ice sheet where anomalous cooling increases SMB. This should then be counteracted by the albedo changes described L12 and following. It is not really resolved for me how these different factors influence each other and which is the dominant mechanism in which region.*
Taking into account the effect of katabatic winds leads to a cooling in 2W with respect to NF at the very edge of the ice sheet. Since the 2W temperature is lower than the NF one the predominant effect is that induced by the katabatic winds, not by albedo changes.

*P12.L13 What is the difference between snow-free and snow free?*
Sorry, this was a typo error

*P12.L22 Melting itself does not necessarily contribute to SLR since melt water can be refrozen in the snow pack. Better replace "melting contribution" by "ice sheet contribution" or similar, also in the rest of the manuscript.*
We followed your suggestion and the occurrences of "melting contribution" have been replaced by "GrIS contribution"

*P12.L22 These numbers should be calculated against a control experiment to remove the contribution from remaining model drift. Has this been done?*

Yes ; it has. Our control experiment is the 2000-yr GRISLI relaxation run. As specified in the revised manuscript, the remaining model drift in terms of ice volume is only $10^{-5}$ mm $yr^{-1}$, fully negligible with respect the GrIS contribution to sea-level rise.

*P12.L23 I would suggest to add a panel to figure 7 with the total contributions for the three experiments and include the integrated SMB mentioned further below in this section.*
This has been done. Figure 7 has become Figure 12.

*P12.L25 Since you discuss 2-W against NC, the surface elevation - SMB feedback which operates all over the ice sheet should also be mentioned, not just the processes at the margin.*
We paid attention to describe through the entire revised manuscript the processes operating in the interior of the ice sheet. In particular, the results are most often presented as a function of surface elevation: we distinguished regions of low to medium altitudes from regions of high altitudes (See in particular Section 4.1 for the description of the results inferred from the 2W method and Section 4.2 for the effects of the katabatic winds which strongly differ from central regions to margin areas.

*P12.L26 The difference of 52400 km2 is at the end of the experiment and then it increases with time? Reformulate*
As mentionned above, absolute changes in ice-sheet extent are no longer discussed in the revised manuscript. Now, this aspect is only addressed in terms of relative changes between NF vs 2W and PF vs. 2W. This has been therefore reformulated in: *"Compared to the NF and the PF experiments for which the ice-sheet mask is fixed to observations from 2000 to AD 2150, the 2W ice sheet extent is reduced by ~2.8 % in 2150 as a result of increased ablation".*

*P12.L27 I think all you are saying is that the high resolution ISM mask changes are translated to partial mask changes for MAR. Clarify that the ice sheet mask (Fig S2B) is the one seen by MAR.*
This part has been re-written:
*"Compared to the NF and the PF experiments for which the ice-sheet mask is fixed to observations from 2000 to AD 2150, the 2W ice sheet extent is reduced by ~ 2.8 % in 2150 as a result of increased ablation. As MAR sees the ice sheet retreating over time in 2W concomitantly with the increase in bare ground or tundra fractions (Fig. S5b), the albedo feedback takes place favouring further the ice melting. Although the ice sheet retreats, the extent of the ablation zone increases with time. This process is faster in 2W than in NF and PF. In 2150, the ablation zone is 14 % (resp. 11.7 %) larger in 2W than in NF (resp. PF) causing 112 Gt $yr^{-1}$ of extra ice ablation in 2W (w.r.t NF). As a consequence, the ELA is located further inland in 2W compared to NF with a maximum inland retreat of 120 km located in northeastern Greenland (Fig. 3)."*

*P12.L31 I think the point to make here is not about increase in uncertainty. You can show that when a fixed mask is used, you simply get the wrong result and overestimate the mass loss. Could you quantify the relative importance of this effect compared to the error that is made when not taking into account the surface elevation - SMB feedback?*
In Section 4.4, we quantified 1/the error made when the SMB-elevation feedbacks are ignored (i.e. 7.6%, deduced from the comparison between the SLR contributions in NF and PF

experiments) 2/ the error made when all the feedbacks are ignored (i.e. 9.3 %, deduced from the comparison between the SLR contributions from NF and 2W) and 3/ the error made when using a fixed ice-sheet mask (i.e. 6 %). To follow the suggestion of the reviewer, we added the following sentence at the end of the section:

*"[…] compared to a time variable ice-sheet mask, the use of a fixed ice-sheet mask overestimates the sea-level rise by ~6 % in 2150. Though a bit lower, this number is far from being negligible compared to the errors made when the SMB-elevation feedbacks are not taken into account (i.e. 7.6 %) and when all the feedbacks are ignored (i.e. 9.3 %)".*

*P13.L3 Please specify the resulting SLR.*
The resulting sea-level rises obtained with the integrated-SMB methods have been explicitly mentioned in the text of the revised manuscript (Section 4.4) and reported in the new Table 2.

*P13.L14 This is exactly the reason why median results are not very meaningful in this context. Mean absolute or root mean squared differences are easier to interpret.*
In the revised version, we no longer mention the median values. We provide instead the ice thickness root mean square errors as well as the 5$^{th}$ and the 95$^{th}$ percentiles in ice thickness differences for regions showing similar patterns (margins vs. interior).

*P13.L15 After showing figure 8, figure 9 does not add substantial information in my view. I would remove it and continue discussion about differences between 2W and 1W based on figure 8. The only reason to show figure 9 would be if you wanted to attempt modifying the parameterisation used in 1W to incorporate the katabatic wind effect, which could be a logical next step.*
We acknowledge that part of the information provided in Figure 9 (Figure 11 in the revised manuscript) can be found in Figure 8 (Figure 10 in the revised manuscript). However, we believe that the new Figure 11 better illustrates the differences between the three experiments in the simulated spatial variability as a function of the altitude. This is why we finally kept this figure in the revised paper.

*P13.L27 These sentences are just stating the obvious. I'd suggest to remove them.*
We agree with this statement: the sentences have been removed.

*P14.L5 It is not clear to me why a higher resolution should lead to increase the SLR and not the opposite. Unless there are convincing arguments to support that claim, I would leave the sign of the change open.*
The reviewer is totally right. We recognize this was an overstatement. In the revised manuscript, the Discussion section has been extended and we better explained the possible influence of outlet glaciers on projected SLR. We also mention the possible decreasing influence of the outlet glacier dynamics with time:
*"Regarding the ice-sheet model, a 5 km horizontal resolution does not permit to capture the complex ice flow patterns of smallest outlet glaciers, whose characteristic length scale can be less than 1 km (Aschwanden et al., 2016) and to quantify accurately the ice discharge at the marine front. This may have large implications in the sea-level rise estimates. Using a 3D ice-*

*sheet model with prescribed outlet glacier retreat, Goelzer et al. (2013) found an additional SLR contribution from outlet glaciers of 0.8 to 1.8 cm in 2100 and 1.3 to 3.8 cm in 2200, with the influence of their dynamics on SLR projections decreasing with time and with the increasing importance of the atmospheric forcing. This is in line with the fact that ice dynamics act to counteract ice loss from surface melting (see Section 4.2), as previously outlined by several authors (Edwards et al., 2014a, Goelzer et al., 2013, Huybrechts and de Wolde, 1999). However, despite the possible decreasing influence of marine terminating glaciers, at the centennial time scale, it seems to be preferable to evaluate more accurately the impact of ice dynamics and to better capture the complex geometry of fjords surrounding the marine-terminating glaciers".*

*The same applies to the limitation of constant basal drag in the next sentence. With all the complexities surrounding the evolution of the basal conditions over time, I don't think there is any evidence that acceleration of ice flow has to be the dominant response. Again, putting forward some convincing arguments would be appreciated.*

Again, we fully agree with the reviewer and this sentence has been removed from the revised manuscript. In particular, we discussed the limitations related to the time constant basal drag coefficient and to the lack of any infiltration scheme in our ice-sheet model:

*"Our spin-up method adjusts the basal drag coefficient in such a way that the departure between the observed and the initial GRISLI topographies is reduced. The resulting coefficient is spatially varying but is constant in time. This assumption may likely be valid for short-term forward simulations but is probably overly simplistic. On the one hand, the basal drag tends to be smaller towards the margins with respect to the interior. As the ice sheet retreats inland, it can be expected a reduction in basal drag for a specific location, due for example to a decreasing effective pressure. On the other hand, changing basal hydrological conditions can also alter the basal drag. This can occur as a result of rainfall or surface meltwater infiltration that can refreeze at depth or propagate all the way to the bottom of the ice sheet and increase basal lubrication (Kulessa et al., 2017). Therefore, a time constant basal drag coefficient inferred under present-day conditions may underestimate the ice flow acceleration. A few models describing the vertical inflow exist (e.g. Banwell et al., 2016, Clason et al., 2015; Koziol et al., 2017) but are generally run at the regional scale and at very high spatial resolution (a few tens to a few hundreds of meters at most). Implementing such models in large-scale ice-sheet models is currently outside the realm of possibilities. However, as there is a growing interest in performing ice-sheet projections over multi-centennial time scale, the GRISLI-like models would undoubtedly benefit from the implementation of simplified infiltration schemes (e.g. Goelzer et al., 2013) so as to account for the impact of ongoing changes in surface meltwater on ice dynamics".*

*P14.L19 Additional limitations that should be discussed: - Ignoring the glacial-interglacial signature of past climate changes in this steady state spin-up of temperature typically makes the ice too warm. This needs to be compensated by other factors (likely a different set of basal drag parameters). - The steady state initialisation also ignores any influence of transients in the*

*observed ice sheet evolution – Mismatch of the modelled ice sheet geometry and velocity structure with observations leads to uncertainties in the projected evolution.*

We added the following paragraph in the Discussion section (Section 5):

*"An additional limitation related to the choice of our spin-up procedure is that the glacial-interglacial signature of past climatic changes is ignored. Neglecting the climate history of the Greenland ice sheet implies too warm ice temperatures. This may have an impact on the future GrIS evolution and on its contribution to sea-level rise. Indeed, the basal drag coefficient inferred from the inverse method may be too high so as to compensate the errors induced by the artificial warm bias. However, using a higher-order ice flow model, Seroussi et al. (2013) showed that at the centennial time scale the basal conditions and the GrIS projections are only poorly sensitive to the initial vertical temperature profile but are critically dependent on atmospheric conditions".*

*P14.L28 Add "in this comparison" after "atmosphere-GrIS feedbacks". I hope you don't think this statement is universally true.*

Due to the huge changes made to the original text, this issue has been presented differently and mentioned in Section 3.3 devoted to the description of the 2W experiment: *"Compared to the NF and PF approaches, this two-way coupled method is the most accurate to represent the GrIS-atmosphere feedbacks".*

*P14.L30 While this statement seems true for the given results, the conclusion hinges on the change in behaviour of 2W at 2110. Unless investigated in more detail, it cannot be excluded that such change could happen at an earlier point in time, e.g. for a different model used as boundary condition to MAR.*

In the new version, we showed that the results from the three experiments start to diverge from each other as soon as 2025-2030, that is a few years only after the start of the coupling. This means that the feedbacks that are accounted for in the PF (SMB-elevation feedbacks) or in the 2W experiment start to operate as early as this period. However, we also explain that the influence of the feedbacks increases over time and that they become dominant at the end of the 21$^{st}$ century (See in particular Section 4.4). Moreover, in the Discussion section, we clearly explain the possible dependence of our results with the GCM forcing used to force MAR:

*"Whatever the experimental design, the large spread in SLR projections raises the question as to whether the ice-sheet response simulated in our 2W experiment relative to that of the NF and PF experiments would be similar, amplified or mitigated with a different GCM climate forcing having a different sensitivity from MIROC5. [...].There is therefore a strong need for iterating the present study with different global climate simulations run under an extended RCP8.5 scenario and used as a MAR forcing, to assess more accurately the impact of the different GrIS-atmosphere feedbacks and to better evaluate the uncertainty associated with the projected sea-level rise contribution from GrIS".*

*P15.L5 This comparison is a bit awkward. Wouldn't it be more appropriate to compare the +0.5 cm to the total projected SLR as a relative error?*

We acknowledge that this comparison was not fully appropriate and we removed it from the revised manuscript.

*P15.L14 Remove repeated "respectively" after SLR.*
The sentence has been changed.

*P15.L19 It would be good to additionally put this number (21%) in perspective to the underestimation due to ignoring feedbacks, i.e. the difference between 2W and NC.*
This comparison has been done (see our response to comment P12.L31).

*P15.L24 Again, I don't see any evidence for the interpretation that higher resolution and higher order physics increase the response.*
We agree with you. This comparison was not appropriate (See our response to comment P14.L15).

*P15.L29 Replace "disrupt" by "modify"*
OK, all occurrences of "disrupt" have been removed.

*Table 1 Does "after 50 yrs" mean at year 2050? Maybe that would be a better indication. Or do you not want to assign an absolute date to your simulation? The historic and future RCP forcing is clearly linked to an absolute date, though. Since the ablation area changes so much, it may be interesting to calculate additional diagnostics for a constant region, e.g. for the observed present day ablation zone, or backwards for the area of the ablation zone area after 150 years. This way, the convolution with a changing area could be avoided.*
In the revised version, most results have been discussed as a function of altitude. We mainly distinguished two type of areas: areas of high altitude (generally higher than 2000 or 2500 m) and areas of low to medium altitude (< 1000 or 1500 m). As a result it does no longer make sense to present SMB and ST values (Table 1) or GrIS thickness or ice velocities (Table 2) at the scale of the whole ice sheet, as it was done in the first version of the paper. Moreover, the results computed at the ice-sheet scale are not really informative because of the large spatial variability in the 2W-NF anomaly. In addition, the changes in ablation area and in ice-sheet extent have been discussed in the revised paper in terms of relative changes (see Section 4.4). To our opinion, they don't need to appear in a table. We therefore removed both the former Tables 1 and 2 from this new version of the manuscript. However, we replaced these tables by new ones providing the GrIS contribution to sea-level rise inferred from NF, PW and 2W experiments (new Table 1) and from the SMB-integrated method (Table 2). We also replaced "after 50 years" by the absolute dates.

*Table 2 Not sure how to interpret a velocity change of e.g. -3.0+-25.0. The noise being much larger than the signal, is the valid interpretation 'no significant' change?*
We fully agree. See our previous response to comment related to Table 1.

Figures ——-

*The labels in the figures are upper case (A,B,C), but the panel references in the captions are all in lower case (a,b,c). Make consistent.*

OK, the labels in the figures and in the captions are now identical

*Figure 2 Why are figures B and C so different? At least in the interior, one would expect a pattern very similar to the SMB anomalies in this experiment. My guess is that this is indicative of a remaining model drift. Results of a control experiment starting after step 3 with constant forcing should be shown here or in the appendix and the origin of this difference should be discussed.*

To address the comments raised by Reviewer 2, the organization of the paper has been modified. We now start the result section with a thorough analysis of the GrIS evolution simulated with the most comprehensive method, i.e. the two-way (2W) coupling. The NF results are only presented in terms of differences with the 2W method. Moreover, we think that the differences you mentioned between both plots can be attributed to the choice of the color scale. The new figures (Figs 2a and 4) present similar patterns for SMB and ice thickness anomalies simulated in the 2W experiment. However, the discussion requested to explain the differences between the SMB and the ice thickness patterns has been provided for the 2W method (see Figs 2a, 4 and 5 in the revised manuscript and section 4.1.3). These differences are explained by the ice dynamics. In Section 4.1.3, we added the following paragraph to support this argument:

*"The ice thickness anomaly is due to the complex combination of changes in surface atmospheric conditions (SMB, Fig. 5a), ice dynamics (ice flux divergence, Fig. 5b) and basal melting (not shown), following the continuity equation (Eq. 2). To quantify the role of ice dynamics on the GrIS geometry (Fig. 4), we plotted the ice flux divergence integrated over 150 years (2000-2150, see Fig. 5b). In particular, over the central plateau, the cumulated SMB (Fig. 5a) reaches about +50 m, 40 m of which are transported away by the ice dynamics (Fig. 5b). As a result, the ice thickness anomaly is reduced to only ~10 m in this region (Fig. 4). An opposite behaviour is found near the western coast, where the ice melting is partly compensated by ice convergence, resulting in a less negative ice thickness anomaly than that related to the SMB forcing. This shows that ice dynamics act to counteract ice loss from surface melting, as previously noticed by several authors (Huybrechts and de Wolde, 1999, Goelzer et al., 2013, Edwards et al., 2014b). As a consequence, it appears to be essential to account for ice dynamics to estimate accurately the mass balance of the whole ice sheet".*

*Figure 3 The displayed field is ice thickness, not surface elevation as written in the caption. Since the discussion is about ELA and surface elevation - SMB feedback, it may be useful to show surface elevation instead.*

Figure 3 has been modified. Now it displays the surface elevation

*Figure 4 The colour scale in A is not easy to read with small positive and small negative values sharing the exact same colour (green). This should be improved. Have you tried to plot velocity ratios instead of anomalies? Since velocity magnitudes cover several orders of magnitude, a large relative change is not visible because of the cutoff at 2 myr-1, while small relative changes at the margin appear exaggerated.*

The surface velocity anomalies are now plotted in the new figure 6 1/ for the 2W experiment (instead of NF) between the end (2140-2150) and the beginning (2000-2010) of the simulation and 2/ for the anomalies between 2W and NF and between 2W and PF at the end of the

simulation. The colour scale (in log10) has been also modified to better illustrate positive and negative changes.

*Figure 5 Why not use the same colour mapping here and in figure 4 for the velocity anomalies? That would make it easier to compare the two figures.*
Figure 5 (in the first manuscript) is now Figure 7 and uses the same colour scale as Fig. 4.

Caption: "left panel" The figure caption has been modified

*Figure 6 There appears to be a slight instability in one or both of the experiments compared in figure 6A. Also Figure 8A shows signs of instability in form of a checker board pattern. While these instabilities are likely not critical for the interpretation of the large scale results presented here, they should at least be mentioned.*
Figures 6A and 8A are now Figures 8b and 10b. The features the reviewer refers to are related to the method used to correct for the altitude difference between the MAR and the GRISLI topographies.

*Figure 7 Add a panel with absolute contributions of the three experiments. Note that results shown so far are double differences, i.e. differences in anomalous contributions since year 2000 between different experiments. Could also show sea-level contribution differences calculated from difference to a control experiment with constant forcing to remove the model drift. Same consideration holds for the absolute contributions.*
Figure 7 appears now as Figure 12. We added a panel showing the absolute contributions (Fig. 12a). In all the simulations presented here, the model drift has been taken into account but is fully negligible (see our response to comment P7.L32).

*There seems to be a step change around 2060 and again around 2110, where the behaviour of 2w-1w (yellow) changes dramatically. By comparison with 2w-nc it appears to be caused by the evolution of 2W. What is happening at these moments in 2w? Please investigate this further.*
The figure showing the anomalies of sea-level contributions has been replaced by a zoom-figure. Thanks to this new panel, we show that the three experiments start to diverge from each other as soon as 2025-2030 and not only around 2060. We do not observe any significant change in slope in 2060 nor in 2110.

*Caption: "Differences in Greenland ice sheet sea-level contribution between the different experiments." Then explain how it is calculated.*
We provided further details in the figure 12 caption.

*Figure 9 is not needed in my estimation.*
See our response to comment P13.L15. Note also that Figure 9 now appears as Figure 11.

References:

*Format of many references in the text are non-standard. A few examples are given*

*here, but all should be re-checked.*
The format of the references has been re-checked.

*P3.L13 add e.g. before Gagliardini*
OK, added

*P3.L19 reformat list of reference and avoid double brackets*
P4.L10 "(e.g. Fettweis et al., 2013
OK, reformatted

*P6.L12 Author is called Fox Maule. Check reference.*
OK, modified

*P6.L20 Reference Le clec'h et al. (in prep) is not in the reference list.*
This reference now appears as Le clec'h et al. (2018).

*P8.L2 add Goelzer et al., 2017 to the reference list.*
OK, added in the reference list

*References:*
*Fox Maule, C., Purucker, M. E., Olsen, N., and Mosegaard, K.: Heat flux anomalies in Antarctica revealed by satellite magnetic data, Science, 309, 464-467, doi: 10.1126/science.1106888, 2005.*
The reference has been corrected

---

## Author Response (AR2)

Dear Editor,

As requested by the Editor Prof. Valentina Radic, we have revised the paper in time. We are very grateful to you for accepting the successive deadline extensions and we apologized to the time takes to submit this revised version. This time have been helpful for us to largely improve the manuscript following the numerous reviewers' comments and detailed concerns
during the peer-review process.

Following the J. Fike and the referee #3 reviews, we have even more clarified sections dealing with initialisation of the atmospheric model MAR (Section 2.1) and of the ice sheet model GRISLI (Section 2.2).

We also redid all the figures to fit with the revised result Section 4 and following the referee #3 comments.

Finally, we made our best to improve the English language.

Best regards,

Sébastien Le clec'h (on behalf of all co-authors)

**We would like to thank the reviewer J. Fyke for the evaluation of our study. Please find below the reviewer's comments in black font and the author's response in blue font.**

**Responses to J. Fyke (Reviewer 1)**

*I find this paper to be much improved. Thanks to the authors for working to address reviewer comments. I have two remaining areas of concern that I think require greater 'caveating' in the final paper.*
Thank you for this comment.

*P6L4: "SISVAT takes only 7 years to reach equilibrium". I am slightly surprised that the full SMB field over GrIS takes 7 years to equilibrate (for example, there must be a few locations where longer equilibration is necessary to establish the simulated ELA..?). Perhaps in other words, how is the treatment of bare ice ablation zones treated and is snow (as represented in SISVAT) initialized with spatially varying thicknesses? I think readers (especially those familiar with the multi decade timescale of firn, for example) will be surprised by the 7 year spin-up, and will want further information on snow initialization procedure. Please include.*
Thanks for the suggestion. In fact, the snow model SISVAT is first initialised with the averaged snowpack coming from previous MAR simulations carried out under present-day conditions (1960-1999). Using this snowpack equilibrium MAR is then initialised from 1970 to 1975 using MIROC5 forcing fields.
In the new version of the manuscript, we modified the paragraph:
 *"Because the snowpack in the land model requires generally longer time scale than MAR to reach an equilibrium with the atmospheric forcing, here MAR is spun-up for 6 years forced at its lateral boundaries by outputs from MIROC5 from 1970 until 1975 and by an initialised snowpack coming from a previous MAR simulation carried out under present-day conditions (1960-1999)."*

*P9L5: Thank you for noting the application of negative SMB values outside the present-day ice sheet mask. While I accept that this is the philosophy/methodology chosen for this study (which is mirrored elsewhere, e.g. ISMIP6 standalone experiments I believe) I think this needs a greater caveat in the context of a two-way coupling manuscript. This is because in GCMs, the use of flux corrections (which this functionally is, in the form of a flux-based bias correction) has been essentially entirely abandoned, because it technically violate true coupling (the conservative transfer of fields between components, as in the real world). This introduces hard-to-understand ambiguity into final results (e.g. impact of coupling-induced feedbacks).*
We acknowledge that our methodology is not suited for a true coupling. In addition to the flux correction you rightly mention, we also follow an anomaly method between the ISM topography changes and the atmospheric model. Thus, in our model framework, we can't have mass and energy conservation between the ISM and the atmospheric model. This caveat has been further emphasized in the revised manuscript (Sec. 5):

*"Moreover, the use of an anomaly method to account for the change in topography is incompatible with a conservative coupling between the ice-sheet model and the climate model. This is further amplified by the fact that we use a flux correction outside the present-day ice margin to force ice removal. This methodology has been followed to limit the impact of biases from the atmospheric model and from the initialization procedure, but the imposed ice removal outside the present-day ice mask may bias locally the model response towards increased ice thinning. Since our simulations are run under the RCP8.5 forcing scenario, this has probably a negligible impact. However further studies of future climate with alternative scenarios and/or GCM forcing, and even more paleoclimate studies, should ideally avoid using this kind of flux correction".*

*Finally, I'm not sure I agree with the final sentence of the paragraph referenced here: it could be that remaining SMB biases (which are probably mostly related to MIROC biases) would drive GrIS expansion, even during the warming future. From a paleoclimate perspective, this approach also limits the model configuration from expanding in colder climate states (though I recognize that's not the point here).*

We fully agree with your comment and we do not recommend the use of an artificial SMB correction outside the present-day ice mask (i.e. strongly negative SMB) for paleoclimatic experiments. MAR uses MIROC outputs only at its lateral boundaries. Inside its domain, MAR generates its own boundary layers climatic fields and is less impacted by near-surface MIROC biases. In our study, and under the RCP8.5 scenario, the GrIS margin region is only marked by ice thinning. However, under a different RCP scenario and/or a different GCM forcing we could obtain locally an ice expansion.

*In general, please more clearly describe the potential caveats of the imposition of an artificial SMB 'moat' around the ice sheet based on the observed ice sheet shape, so that readers are clearly aware of the possible implications to the main feedback-quantifying results.*

Please see the text addition suggested in our response to your first comment. We hope this will help the reader to better understand the caveat of using such a flux correction.

**We would like to thank the reviewer #3 for the evaluation of our study. Please find below the reviewer's comments in black font and the author's response in blue font.**

**Responses to Reviewer 3**

**Summary:**

*The authors have dramatically improved the quality of the paper since the last version and most of my comments have been sufficiently responded to. Since the paper has changed a lot, I still have a number of additional minor comments that the authors should consider before publication.*
Thank you very much for this comment.

*Because Figure 9 does not appear to be displayed in the manuscript, I feel I have to tick major revisions. If this problem should be resolved and the figure is checked by at least one of the reviewers, I don't need to see the manuscript again.*
After reading your comments, we realized that Fig. 9 did not appear in the pdf file. In the new revised version, we will pay a close attention that the problem does not occur again.

***General comments:***

*It is not clear to me why the sea-level contribution (and some other quantities like surface elevation, ice thickness, ice masks ...) should be averaged over 10 years periods in this reporting. In consequence, a forcing period of 100 years (2000-2100) is practically reported as a 90 year difference in the results, which is confusing. The sea-level contribution is physically a time integrated quantity and does not exhibit inter-annual variability that has to be averaged out. In most cases (instead where strong inter-annual variability exists (SMB, temperature) I would omit the averaging and calculate direct differences. If not, the labels in the table would have to be adjusted to the centre of the averaging period in all tables and plots instead.*
Following the reviewer 1's suggestion, we averaged our model results over a 10-year period (in the first revised version) instead of a 5-year period (in the initial manuscript). This is important for climatic variables (e.g. SMB, temperature, winds) in order to build robust climatologies. However, we acknowledge that for integrated quantities such as sea level, ice volume and ice thickness, it is more relevant not to average the values. To account for your comment, we did the computations of a few diagnostics (e.g. ice thickness change) using the variable at 2150. As a result, Figures 3, 4, 8c, 10c and 11 have been modified accordingly. However, we verified that the new results are very close to those obtained from a 10-year mean (within 1 or 2%) showing that the message given in the previous version of the manuscript was not altered.
Concerning, the sea-level: We acknowledge that there was an error in the captions of Fig. 12 and of Table 1 caption (corrected in the new revised manuscript) and that the values reported in Table 1 correspond to those obtained in 2050, 2100 and 2150 (not averaged over a 10-year

time period).The situation is different in Table 2 (now combined with Table 1 following your suggestion) since the sea-level is here inferred from SMB variations between 2100 and 2000 or between 2150 and 2000. Since SMB is a climatic variable (that may be subject to inter-annual variability) we think that the use of a 10-year mean is more appropriate.

*In the reorganisation of the manuscript (that I appreciate), the 2W experiment is now discussed first as the reference and differences to NF and PF later. Since you introduce 2W as the standard experiment, I would plot and discuss the differences NF-2W and PF-2W (instead of e.g. 2W-NF). It just changes the sign, but seems more consistent with the rest of the document. I think it would also make sense to explain the model setup/coupling method in that order, starting with 2W.*

Thank you for the proposed reorganization. We prefer to keep the presentation of the coupling methods in our initial order. As it is presented in the manuscript, the methods are shown from simple to more complex and we think that it is easier to follow since the reader has simply to understand the additions from one method to the other. However, it is true that it is somehow subjective and that we can change the order if you strongly suggest us to do so.

**Minor comments**

*I believe the affiliation count is incorrect with Fettweis and Wyard linked to Brussels and Ritz linked to Liege.*
This has been corrected in the new version.

*P1 L9 Start a new sentence after MAR: "They are fed ..."*
Modified

*P1 L16 Remove "important". There is already an "important" in the line before.*
Removed and replaced by "significant".

*P1 L18 Remove "tend to" before "favour" to make this statement less vague.*
Removed

*P1 L20 Does it "reduce the SMB signal" or rather re-distribute the additional mass. Clarify.*
Replaced by "counteracts the SMB signal"

*P1 L22 This should probably be "ice volume above floatation".*
Thank you for this precision. This has been specified in the revised manuscript.

*P2 L1 The results this conclusion is based on have not been described so far in the abstract. This must arise from a comparison to the uncoupled experiments. Suggest to mention those before this conclusion.*
This has been clarified:
"*The comparison between the coupled and the two uncoupled experiments suggests that the effect of the different feedbacks is amplified over time with the most important feedbacks being the SMB-elevation feedbacks*".

*P2 L14 Replace "polar" by "Arctic".*
Replaced.

*P2 L18-21 Complicated sentence, consider splitting in two and revising.*

We have splitted the sentence in two, as follows:

*"However various feedbacks between the atmosphere and the GrIS impact the ice-sheet surface characteristics such as ice extent and thickness. This has potential consequences on ice dynamics (e.g., due to changes in surface slopes) and may lead to SMB variations that can therefore affect the total ice mass of Greenland".*

*P2 L23 Add "changes" before "in ice-covered area" to make the relation clearer.*

Clarified as suggested.

*P2 L23 I think with "albedo feedback" you mean here the change in land surface type from ice to tundra, which leads to warming and further ice sheet retreat. This is typically a slow process, I would call "planetary albedo feedback". There is also a more immanent "melt-albedo feedback" as melting snow at the surface absorbs more short-wave radiation, leading to more melting and increasing albedo. These feedbacks are quite different in nature and time scale and should ideally be distinguished.*

Yes, we fully agree with you and replaced "albedo" by "planetary albedo"

*P2 L27 Replace "predict" by "project".*

Replaced.

*P2 L29 In the manuscript you use different ways to order the references. Here it goes from newest to oldest. The TC guideline is open ("In terms of in-text citations, the order can be based on relevance, as well as chronological or alphabetical listing, depending on the author's preference."), but I would at least try to be consistent in all cases throughout the manuscript to not confuse the reader.*

Thank you for this remark. We have chronologically ordered the references throughout the revised paper.

*P3 L32 The approach of Edwards et al. was only used to correct for the SMB-height feedback. Here it sounds like it was also used for downscaling ("An alternative approach [to Franco]"). Please reformulate.*

We agree with the possible confusion and have reformulated the sentence to clarify the approach of Edwards et al. (2012) as follows:

*"An alternative approach to correct the SMB field from surface elevation changes is based on statistical relationships between altitude and SMB (Edwards et al., 2014b). Also been derived from MAR, this approach computes a SMB-elevation feedback gradient for regions below and above the equilibrium line altitude in the northern and southern parts of GrIS, with limited additional computing resources".*

*P4 L9 Replace "surface energy balance" by "surface mass balance".*

A better representation of the surface mass balance requires necessarily a more complex representation of the surface energy balance. To clarify our idea we have changed the sentence as follows:

*"Additionally, the use of a detailed snow model such as that implemented in MAR (Fettweis et al., 2017) or RACMO2 (Noël et al., 2015) allows a more accurate description of the surface*

*properties (e.g., snow cover, albedo, surface melting) and therefore a better representation of the surface energy balance and hence of surface mass balance".*

*P4 L27 Replace "second" by "third".*
Replaced.

*P4 L31 "to the other, uncoupled experiments".*
Modified.

*P5 L11-13 "and 24 vertical levels" misses a verb. Suggest "MAR has a horizontal resolution of ... and 24 vertical levels ..."*
The sentence has been modified as follows:

*"MAR has a horizontal resolution of 25 km x 25 km and 24 vertical levels to describe the atmospheric column in sigma-pressure coordinates (Gallée and Schayes, 1994). The MAR domain covers the Greenland region (6600 grid points), from 60°W to 20°W and from 58 °N to 81°N".*

*P5 L17 Maybe "covered by at least 0.001 % tundra and at least 0.001 % snow" to make clear both surface types are represented. Also add an explanation why that is done.*
Thank you for the suggestion. In the new revised version we explained why MAR used a minimun percentage of tundra and snow in each grid cell:

*"Each grid cell is assumed to be covered by at least 0.001 % of tundra and at least 0.001 % of snow so that the retreat or the expansion through time of snow (reciprocally tundra), especially at the margin, can be explicitly represented outside the original MAR ice-sheet mask".*

This has been done with the aim of coupling MAR with an ice sheet model. Note also that when a grid cell is covered by 50 % of snow or ice it is considered as a permanent ice sheet grid cell, as specified in Fettweis et al. (2017).

*P5 L21 Add why these parameter changes were needed? What was improved?*
This point has been clarified in the revised paper:

*"The differences with previous MAR versions (e.g., Fettweis et al. 2013) are only related to adjustments of some parameters in the representation of cloudiness and bare ice albedo. These new parameterisations allow to better account for the positive feedback that cloud cover exerts on surface melting (Van Tricht et al., 2016) and to represent the impact of melt ponds that strongly reduce surface albedo (Alexander et al., 2014)".*

*P6 L12 Regular grid? What projection? Refer to Bamber?*
We apologize for the missing information. GRISLI uses a 5 x 5 km regular grid projected on a polar stereographic projection with a standard parallel at 71°N and a central meridian at 39°W. This has been added in the new version of the manuscript.

*P6 b_melt is only defined for grounded ice, I suppose? What is done for shelf melting? Clarify.*
This equation is valid for both grounded and floating ice, and so, $b_{melt}$ is defined for both grounded ice and the ice shelves.

*P6 L19 Remove "also" before "affects".*
Removed.

*P6 L27 Consider defining "The shallow ice approximation (SIA)" instead.*
Modified as recommended for both SIA and SSA.

*P7 L17 Section title should be "Initialisation procedure". "Spin-up" should be reserved for a consistent long-term transient run as often done with ice sheet models.*
This suggestion has been followed.

*P7 section 2.2.2 At the time of the first review of this manuscript, the GMDD paper of the same author describing the initialisation procedure was not available. It was therefore not possible to understand the procedure from the limited information given in this manuscript. I have therefore asked for more detail to be included. Now that the GMDD paper is published, it would be enough to give a broad overview of the method here and refer to the other paper. At the moment, a large part of 2.2.2 is a copy-and-paste from the other paper with a fair amount of technical detail. I don't think this is a problem in terms of plagiarism, but I would encourage the authors to rewrite and shorten this part to summarise the most important aspects.*
Based on your suggestion, the presentation of the initial procedure has been entirely revised. We now only present the basic principles of the method without any equation, and we ask the reader to refer to the GMDD paper. Please see the revised version.

*What I miss so far (also in the GMDD paper) is a clear idea of the basic principle of the method. While similar to PD2012, the main difference is that beta is modified in function of the thickness ratio, rather than the thickness difference. This has not been clearly state and is lost in the complicated formulation.*
There are in fact two important differences with PD2012. On the one hand, as you rightly point out, the thickness ratio is used instead of the ice surface elevation difference. On the other hand, our procedure iterates from present-day ice thickness with multiple cycles short (i.e. decadal to centennial-scale) simulations. This drastically reduces the computation time relative to PD2012. We hope that the difference between the PD02012's method and ours is now better explained in the revised manuscript:
*"This method is based on the same basic principles as that of Pollard and DeConto (2012) except that their basal drag coefficient is adjusted as a function of the difference between mod-elled and observed ice surface elevation while we use the ice thickness ratio instead. Moreover, while the method suggested by Pollard and DeConto (2012) requires long (multi-millennial) integrations for the method to converge, we use an iterative method of short (decadal to centennial) integrations starting from the observed ice thickness allowing a more rapid convergence".*

*I also find the formulation through $U^{corr}$ very confusing, since the modelled velocities are never directly corrected nor compared to observations. When the equations 4,5 and 6 are put together, one can arrive at a simple expression for the way beta is adjusted in the method (see below). I think it would add substantially to the process-understanding to include it here and/or of still possible in the GMD paper. In the method the basal drag coefficient beta is updated iteratively by multiplying the old beta with a factor rbeta = beta_{new}/beta_{old} (was $U^{slid}/U^{corr}_{slid}$ in Eq 6 before). By combining Eq 6,5 and 4 one can show that the inverse of rbeta is beta_{old}/beta_{new} = rH + U_{deformation}/U_{sliding} * (rH-1), where rH = $H^{G}/H^{obs}$, is the ratio of modelled and observed ice thickness, and U_{deformation} and U_{sliding} are the modelled velocities due to deformation and sliding, respectively. This means that the adjustment of beta is in the end a function of the thickens ratio with stronger adjustment in regions dominated by deformation.*

Thank you for this suggestion. As mentioned above, we only provide the basic principles of the initialization procedure without giving any equation.

*P8 L5-31 It seems to me that you start by skipping step 1 and then end by skipping step 2. If that is the case it would be clearer to exchange 1 and 2 and start the description with what is now called step 2.*

We hope that the different steps of the procedure are more clearly presented in the new version.

*P8 L20 Maybe add that beta_new = beta_old in the first iteration for clarity.*

These notations are no longer used in the new version

*P8 L28 I don't think NB_cycle is used afterwards. Maybe it is possible to avoid using the symbol all together.*

Yes, you are right. Your suggestion has been followed.

*P9 L7 I would say the "impact" of this condition is strong because it keeps ice from building up where there is none in reality. What you want to say is that it has a negligible impact on the projected SL contribution.*

Applying a strong negative SMB outside the present-day observed ice sheet extent could have strong impact for colder climate projection. However, using RCP8.5 forcing scenario, the ice sheet expansion would be probably very limited, and we believe that it would have only a little impact at the global scale, and in particular on the GrIS contribution to sea-level rise. As also suggested by the Reviewer 1, we added a discussion of this caveat in Section 5. Moreover, the sentence you refer to has been changes in:

*"This avoids ice growth where there is none in reality and allows to correct for both the potential atmospheric model biases (e.g., positive SMB values over tundra areas) and the initialisation procedure biases (i.e. too strong ice export towards the margins). However, for GrIS projections run the RCP8.5 forcing scenario, this condition has only a limited impact on GrIS contribution to sea-level rise since the ice extent will likely to keep on retreating over the next centuries".*

*P9 L8 Remove "quite" before "negligible".*

Removed.

*P9 L23 This sentence repeats most of what is already in the sentence line 20. Remove?*

We agree with this suggestion. The sentence has been removed.

*P9 L27 I don't think the SMB can be said to be consistent with the 5 km topography. Maybe "generate a 5 km resolution SMB for the Bamber et al. (2013) topography"?*

By consistent we mean that the new 5 km resolution SMB inferred from the method of Franco et al. (2012) is now adapted to the fine scale features of the 5 km Bamber et al. (2013) topography. We have clarified this point in the revised paper:

*"Thus, this method allows to generate a 5 km resolution SMB entirely adapted to the fine scale features of the 5 km Bamber et al. (2013) topography".*

*P9 section 3 You chose to show results for 2W first and discuss differences to the other experiments afterwards, I agree with that. But wouldn't it then make sense to also explain the model setup/coupling method in that order, starting with 2W?*

See our response to your general comment. While we consider the 2W experiment as a reference, we think that the presentation of the different methods going from lower to higher complexity makes the understanding easier for the reader.

*P10 L18 It is not clear what "aggregated" means. Some process to go from 5 km to 25 km, but what is it exactly? Please explain that better.*

We acknowledge that this sentence was confusing. We tried to clarify this point in the revised version. The text has been changed in:

*"Then, GRISLI computes a new GrIS topography and a new ice extent at a 5 km resolution. This new GrIS topography is then aggregated (i.e. geographically averaged) at the yearly time scale onto the 25 km MAR grid. The number of ice covered GRISLI grid points within a MAR grid cell relative to the number of ice-free GRISLI grid points is used to compute the new ice extent in MAR and to update the fraction of tundra relative to ice/snow covered surface type for the subsequent MAR run".*

*P10 L28 First sentence of 4.1.1 difficult to read. Move "in the 2W experiment" before (Eq. 1).*

Modified as recommended.

*P11 L5 You say there is no inter-annual variability anymore after 2100, but where does the +- 13 Gt/yr come from then? I suppose MAR still has inter-annual variability, but not the GCM boundary condition. This should be clarified.*

The other numbers mentioned in this paragraph (280 Gt yr$^{-1}$ and - 638 Gt yr$^{-1}$) are associated with mean standard deviations equal to 95 Gt yr$^{-1}$ and 271 Gt yr$^{-1}$, representing errors of ~34 % and ~42%. This must be compared to the error of 13 Gt yr$^{-1}$ which represents an error of only 1.6%. This much smaller uncertainty (compared to 34 and 42 %) illustrates the small inter-annual variability due to the repeated 2095-year MIROC5 as atmospheric forcing. Note also that the inter-annual variability simulated by MAR is driven by the inter-annual variability of the GCM (here MIROC5). However, MAR generates its own surface boundary layer fields which are not impacted by MIROC5 forcing.

*P11 L8 I have problems with the formulation "two distinct patterns". You chose to analyse the results separated in two different regions. If you would have made four regions, you would probably observe four different distinct patterns. Can this be formulated better?*

The point which is addressed here is that most grid points having surface elevation higher than 2000 m are characterized by a positive SMB anomaly, whereas lower altitude grid points have a negative SMB anomaly. This is the reason why we thought that "two distinct patterns" was relevant. However, we have reformulated the sentence in the revised version:

*"The SMB anomaly between the beginning and the end of the 2W experiment is displayed in Fig. 2a. 65 % of the grid points having surface elevations higher than 2000 m are characterized by a positive SMB anomaly [...]"*

*P11 L26 There should also be a \*shift\* of the ELA, which should be important to note here.*

The ELA shift is discussed explicitly in the previous paragraph. Therefore, we did not find necessary to mention again this point in the paragraph dedicated to ST.

*P11 L26-27 The relationship between the first and the second part of this sentence has escaped me.*

We apologize for misunderstanding. We have modified the sentence to be clearer. This new sentence explains that even if the northern part of the GrIS is marked by the strong temperature

increase between 2150 and 2000, the ablation processes over this region are counteracted by the increasing snowfall, as shown in Fig. 2b.

*P11 L29 The ice thickness anomaly at what time? Clarify.*
Clarified to specify that the the ice thickness anomaly we consider here is computed between the beginning and the end of the 2W experiment.

*P11 L29 Reformulate "two distinct patterns".*
Reformulated as recommended following your P11L8 comment.

*P12 L8-9 Fig 5a and 5b look nearly identical, because you plot the integrated total SMB (5a) and integrated total ice flux divergence. It would be much clearer to compare integrated SMB anomalies (and integrated flux divergence anomalies) instead.*
Here we disagree. The aim of this section is to evaluate the respective roles of atmospheric forcing (i.e. SMB) and ice dynamics (i.e. ice flux divergence) on the ice thickness variations between 2000 and 2150. These variations are directly given by the integral of dH/dt (i.e. left-hand term of equation 2), which also represents the ice thickness anomaly between 2000 and 2150. According to Equation 2 and assuming that the integrated basal melting is negligible, they can therefore be directly inferred from the integral of SMB and the integral of the ice flux divergence. Conversely, taking the integrated ice flux divergence and SMB anomalies cannot be directly compared to the ice thickness variations between 2000 and 2150 for the reasons mentioned above.

*P12 L10 Replace "ice melting" by "runoff".*
Replaced.

*P12 L15 I think "ice dynamics" is a too broad term here, you may want to say that "ice flow" is impacted by changes in the geometry instead.*
We replaced "ice dynamics" by "ice flow" or "ice flux" when we deal with ice flow but kept the term "ice dynamics" for mechanistic processes.

*P12 L19 "by the combination" of what. There is only one thing mentioned in the following: larger surface slope. Clarify.*
Yes, you are right. We clarified the sentence in the revised version and removed "by the combination of".

*P12 L23 What do you mean by "fully consistent with the decrease of ice thickness"? The velocities are dependent on both thickness and surface slope (and maybe on changes in basal sliding), as you state yourself in the next sentence. This is confusing.*
We replaced the sentence you refer to with:
*"Compared to the 2000—2010 period, this decrease ranges from -213 m yr$^{-1}$ (5$^{th}$ percentile) to -0.2 m yr$^{-1}$ (95$^{th}$ percentile), and favours the decrease in ice thickness".*

*P12 L29 Is it really that clear cut? Changes in surface slope above 1500, ice thickness changes below? Maybe the wording should be modified to "dominated by" or "governed by", allowing for transitions between the two.*
This has been reformulated in:
*"For the Jakobshavn glacier, the ice-sheet areas located above 1500 m, are mainly characterised by an increase of more than 15 m yr$^{-1}$ (i.e. 10 %) of the vertically-averaged ice velocities*

*as a result of increasing surface slopes (Fig. 7c). Conversely, areas below 1000 m are domi-nated by a slow down of the ice flow of more than 200 m yr$^{-1}$ (i.e. 29 %) due to the decreasing ice thickness (Fig. 7c)".*

*P13 L30 "in the same region with full ice cover."*
Modified.

*P13 L31 Remove "artificially".*
Removed.

*P14 L7 "positive elevation-SMB feedback".*
Modified as suggested.

*P14 L8 "and thus increased runoff".*
Modified.

*P14 L21 You should explain why the SMB is lower in PF compared to 2W here. The SMB change in PF is only related to elevation change but in 2W wind changes are dominant.*
SMB computed in both PF and 2W experiments accounts for the surface elevation changes simulated by GRISLI through parameterised SMB-elevation feedbacks in PF and explicitly computed feedbacks in 2W. The main difference between both experiments lies in the fact that, in 2W, simulated ice-sheet changes feedback onto the MAR simulated climate. This leads to a cold air convergence, as explained in Section 4.2.1, slowing down ice melt, hence a higher SMB in 2W compared to PF. This has been specified in Section 4.3:

*"[...] the SMB simulated in PF has even become lower compared to 2W due to a much complex representation of the ice-sheet climate interactions. Indeed, as mentioned in section 4.2.1, in the 2W experiment, the GRISLI topography feedbacks onto the MAR simulated climate leads to a cold air convergence at the ice-sheet margins and thus to a higher simulated SMB".*

*P14 L29 Could you explain why there is mainly negative values in PF? It seems strange to me that there are not more positive values. Or are the positive changes so small that they are not visible on the graph. If so, it would be good to mention that.*
First, we draw your attention that Fig.11a has been changed to represent the NF-2W and NF-PF ice thickness differences (instead of 2W-NF and PF-NF as in the previous version of the manuscript) as a function of the GrIS surface elevation. Thus, NF-PF values are now positive. The only difference between PF and NF relies on the parameterization of the SMB-surface elevation feedbacks taken into account in PF and ignored in NF: All the SMB components are identical in both experiments and the vertical interpolation inferred from Franco et al's method is only applied to SMB. As SMB (in PF) changes as a function of GRISLI simulated topography changes, the ice thickness decreases. Therefore the NF-PF differences are positive.

*P15 L21 I am not sure the attribution to model resolution is as clear cut as it is written here, it seems like a speculation rather than a result for me. Using a different model as such could be an explanation as well I wouldn't exclude. Or is there any other information that convinces you about the resolution dependence, maybe your own experience running MAR at a lower resolu-tion? Please add if that is the case.*
Our aim was not to present the model resolution as a clear-cut statement to explain the differ-ences between the Vizcaino et al's results and ours since we used the expression "may be ex-plained". Moreover, Vizcaino et al. (2015) also mention the need to replicate their own study

with models having a finer resolution and a much complex representation of the model physics. As an example, SICOPOLIS3.0 is based on SIA only, suggesting that fast ice flow is not accurately represented. Moreover, the coarser resolution implies that ablation areas or processes such as katabatic winds, which are both strongly dependent on topography, and thus, on resolution are less well represented compared to models of finer resolution. Nevertheless, in the revised version, we tried to be less conclusive. The sentence has been reformulated in:

*"While the importance of the SMB-elevation feedback may be dependent on the model itself, the larger contribution found in Vizcaíno et al. (2015), compared to our own study, could be explained by the coarser resolutions of ECHAM5.2 (~ 3.75) and of SICOPOLIS3.0 (10 km) with respect to MAR and GRISLI resolutions, implying for example that ablation areas or processes such as katabatic winds are less well represented".*

*P15 L25 Albedo feedbacks have not been really discussed so far. Suggest to remove specific mention here.*
Yes, you are right. "Albedo" has been removed.

*P15 L8 You have to make clear somewhere (earlier) that *the ice sheet mask (in the ISM) is free to evolve in all three experiments*, but that MAR does not see mask changes in PF and NF, because it is calculated on a fixed observed ice sheet geometry. I.e. make a clear distinction between the (modelled) ice mask in the ISM and in MAR.*
As recommended we added this information earlier, in the description of NF and PF experiments (see Sections 3.1 and 3.2).

*P15 L31 It seems that at the very margin the albedo effect is over-compensated by the cooling due to wind anomalies. Make that clearer. Also note that the melt-albedo feedback (as distinguished above) should be active in the ablation at any case.*
We think that "over-compensated" is too strong and we prefer the term "counteracted" since there is a clear negative ice thickness anomaly at the margins of the ice sheet at the end of the 2W experiment. We clarified as follows:

*"As MAR sees the ice sheet retreating over time in 2W concomitantly with the increase in bare ground or tundra fractions (Fig. S5b), the albedo feedback takes place favouring further the ice melting, though counteracted by the katabatic wind anomalies (see Section 4.2)".*

*P16 L1 "A widely used method to estimate the GrIS contribution to global sea-level rise".*
Modified.

*P16 L3 I would say you do something altogether different here by coupling two models. Reformulate.*
Reformulated as follows:

*"In the present study, we use a more complex method since the ice mass variations related to SMB changes are computed by MAR over a changing ice-sheet mask and topography as simulated by GRISLI".*

*P16 L4 "by GRISLI, and MAR sees those changes in the topography", or similar.*
Your suggestion has been taken into account (see our previous response).

*P16 L9 A note on notation. These SMB symbols read like "MSK minus NF". Maybe use underscores instead.*

We changed the notations in $SMB_{MSK_{NF}}$ and $SMB_{MSK_{2W}}$ respectively.

*P16 L12 "this error has a similar magnitude compared to errors made when ...".*
Modified.

*P16 L14 I have mentioned this in my earlier comments and without insisting on it, I would like to reiterate it here: I think it would be interesting to see what mask changes would be produced by using the SMB changes in NF and PF to modify the ice sheet mask offline.*
In order to answer your question, we computed the sum of the SMB anomalies obtained at the end of each year with respect to the mean SMB (i.e. SMB$_{ref}$) obtained during the reference period (1979—2005). This allows to evaluate the magnitude of the SMB variation relative to the reference period and to compute the corresponding variation of topography (obtained by neglecting the ice dynamics). For example, for year i the new topo is given by:

$$(topo)_i = \sum \Delta SMB_i + (topo)_{i-1}$$

With:

$$\Delta SMB_i = SMB_i - SMB_{i-1}$$

[Figure]

We then plotted the difference in ice thickness between the topography obtained in the way described above and the topography simulated by GRISLI in the NF experiment (in 2100). This figure (see above) shows a strong thinning at the ice-sheet margins. This result was expected because there is no ice flow from the ice-sheet interior to the margins since ice dynamics is not accounted for. To go a step further we could have determined the new ice mask associated with this new topography and carried out new MAR experiments to deduce diagnostics similarly to $SMB_{MSK_{NF}}$. However, we do think that this kind of experiments does not bring any added value to the manuscript. Indeed, the effect of the mask has been already extensively discussed. More-over, this figure highlights the role of the ice dynamics which has been described in section

4.1.3. It seems therefore not necessary to add this new figure in the new version manuscript because, to some extent it would be to some extent redundant with what has previously been done.

*P16 L18-20 This is already a long sentence and may be better to split, but I would suggest to add "two uncoupled experiments" before PF.*
The sentence has been splitted in two and changes have been added as suggested.

*P16 L21 remove "first" before step.*
Removed.

*P16 L27 I agree that extending your experiments by using additional and different GCMs would be an important next step in its own right. I would mention that first, not at the end of the paragraph. How the difference between 2W and PF would play out in those runs is certainly interesting, too. I would speculate that the stronger the forcing, the more important the differences between coupled and uncoupled experiments are, similar to the finding of stronger relative error for longer simulations (i.e. stronger forcing).*
The sentence you refer to has been changed in:
*"Whatever the experimental design, the large spread in SLR projections highlights the great uncertainty associated with the choice of the global climate model used to force MAR at its lateral boundaries. It raises the question to what extent the differences between 2W and PF or NF experiments would be amplified (resp. mitigated) with a stronger (resp. weaker) climate forcing than that simulated by MIROC5".*

*P16 L30 "Another question ..."*
Modified.

*P16 L32 "Another consequence is ..."*
Modified.

*P17 L5 I think repeating the exercise with a different RCM would also be important, since they may have different sensitivities.*
The sentence has been changed in:
*"There is therefore a strong need for iterating the present study with different global climate simulations run under an extended RCP8.5 scenario, but also with different regional climate models, that may have different sensitivities, to assess more accurately the impact of the different GrIS-atmosphere feedbacks and to better evaluate the uncertainty associated with the projected sea-level rise contribution from the GrIS."*

*P17 L6 "Another limitation is ..."*
Modified.

*P17 L9 Replace "we choose to" by "we have chosen in this study to".*
Replaced.

*P17 L14 I am not convinced it would be a good compromise to accept unrealistic model drift in the ISM in exchange of a better absolute SMB. I would say there is simply no way around further improving the initialisation, which may also require higher resolution of the ice sheet model.*

There is apparently a misunderstanding. In this paragraph we discuss the limit of using an anomaly method to represent topography changes in the MAR model. We propose a solution to avoid the use of this anomaly method by using the GrIS topography simulated after the ISM initialisation as initial condition of the MAR model. Following this, MAR could then be initialised with the GRISLI simulated topography instead of the observed one. We do not propose a method having an unrealistic model drift (indeed, we propose to still use the Scrl topography that minimises the model drift. The key question here concerns the error made when using Sctrl as initial MAR topography rather than the present-day observed topography taken from Bamber et al. (2013). In this respect we have also to draw your attention on a the typographic error in the previous version (i.e. *GRISLI* was used instead of *MAR*. We have changed the sentence in: "*As an example, a reasonable compromise to avoid the use of anomaly method would be to use the topography obtained at the end of the spin-up iterative process (rather than $S_{ctrl}$) as initial  MAR topography to keep the mismatch with the observed topography as low as possible, and to initialise and perform MAR simulations with this spin-up topography*".

*P17 L18 This is not a full sentence. Reformulate.*
Reformulated:
*"In addition, difference of resolution between MAR (25 km) and GRISLI (5 km) can cause artefacts in the results, especially at the edges of the ice sheet."*

*P17 L33 I agree, but aside from improving the model in this regard, you would also need to force the model with appropriate ocean forcing, which is so far excluded in your setup.*
We fully agree with your comment and we do think that considering the oceanic forcing and or including an oceanic component in the modelling is a main step to improve the SLR projections. This is discussed two paragraphs later.

*P18 L4 ""it seems to be preferable" is a bit weak a statement. I would say "it is essential".*
Modified as suggested.

*P18 L17 Add "taken" before "constant in time".*
Added.

*P18 L27 Can you find a better description than "GRISLI-like models"? Maybe "Large-scale GrIS models".*
As we already used the term "Large-scale GrIS models" few lines over, we changed "GRISLI-like models" by the same used term.

*P19 L4 I don't think the models used in Edwards et al were more sophisticated. In fact, GRISLI was one of them.*
There was an error in the reference of Edwards et al in the previous version. In fact, instead of Edwards et al. (2014a), the appropriate reference is Edwards et al. (2014b). In this paper, besides using GRISLI, these authors also used Full Stokes models (Elmer/Ice) and High order approximation models (GISM, MPAS and CISM), which are more complex models than the hybrid GRISLI based on the SIA/SSA.

*P19 L10 Apparently the use of AD is discouraged. Use CE instead or just write "year 2000 to 2150".*
We thank you for the update. We remove *AD* in the entire manuscript.

*P19 L16 I hope "In turn, changes in the shape of Greenland modify the ice velocities." Is not a conclusion of this study. That's ice sheet modelling.*

We fully agree with you. However, what is obvious for the ice-sheet modelling community is not necessarily obvious for another community. Since this paper deals with RCM-ISM coupling, it may interest readers from climate modelling community who may be not familiar with the consequences of topography changes on ice dynamics. Moreover, our section 4.1.3 is devoted to changes in ice dynamics. It seems therefore justified to remind in the conclusion that changes in ice-sheet geometry influence the ice velocities. However, we slightly modified the sentence so as not to give the impression that this issue is totally novel:

*"Accounting for the GrIS-atmosphere feedbacks amplifies the ice mass loss and changes in ice-sheet geometry with increased surface slopes from the central regions to margin areas and consequences on the Greenland ice velocities".*

*P19 L18 "accounting for the feedbacks".*
Modified.

*P19 L19 "sea-level rise of 20.4 cm in 2150 against 7.9 cm only in 2100" only illustrates that the forcing increases, but not that the feedback does. You could use changing differences between 2W and NF for that.*

Thank you for this remark. We modified the sentence as follows:

*"The effect of accounting for the feedbacks between GrIS and the atmosphere increases with time and becomes significant at the end of the 21$^{st}$ century, as illustrated by the 2W – NF difference in GrIS contribution to sea-level rise in 2150, i.e. 1.9 cm, against 0.3 cm in 2100."*

*P19 L25 Problematic conclusion, because RCM-only experiments (like you) also ignore other factors like ocean forcing. So they overestimate the SMB component, but that may be compensated by missing outlet-glacier changes. These details do matter.*

Our sentence refers to the fact that RCM-based studies that do not account for changes in ice sheet extent (as in PF and NF) and/or ice-sheet topography (as in NF) overestimate the SLR contribution. This is somehow unrelated to the considerations of the oceanic feedbacks due to ice-sheet melting. We acknowledge that this source of uncertainty cannot be quantified with our setup that does not represent the oceanic feedback. The direction (over or underestimation) as well as the magnitude of these feedbacks have still to be quantified but are out of scope of our manuscript. The role of the oceanic feedbacks are nonetheless already extensively discussed in the discussion section. We also would like to draw your attention on the fact that 1/ a higher resolution of the ice-sheet model would allow to better resolve the outlet glaciers and to better represent the ice discharge to the ocean, but not the SMB which is primarily driven by the atmospheric forcing, 2/ capturing the changes in outlet glaciers (simulated by the ice-sheet model) has no effect on an experiment such NF since changes in ice-sheet geometry are not taken into account. In order not to present this last sentence as a universal truth, we have attenuated our statement by adding *"with our modelling setup"*. This obviously implies that the effect of the ocean component is not taken into account.

*P19 L30 You don't mention the code for the GRISLI model. How can the readers obtain the code of the ice sheet model?*
We apologized for the omission and have added this information.

*P20 L6 It is not really my role to comment on the Acknowledgements, but in appreciation of your own work and that of the other reviewers, I think it is more than "the writing" that has improved in the manuscript. Suggest to write "helped to improve the manuscript".*

We are of course very grateful for all the comments you provided

**Tables:**

*P26 Table 1 No need to average (see general comment). I suggest to calculate 2050 2100 and 2150 differences to the year 2000 instead.*

This has been corrected.

*P27 Table 2 Same comment as for Table 1. No need to average. Then remove "Mean" from the caption.*

Table 2 has been merged with table 1. However, the column referring to SMB (not SLR) is now included in the text Sect. 4.4. This has been clarified in the caption.

*For a better overview these results (SLR) could be joined with table 1 and the two SMB values reported in the text.*

*Change label in first row to SMB_(MSK_NF).*

*Same comment as above: these SMB symbols read like "MSK minus NF". Maybe use underscores instead.*

As recommended, tables are now joined, symbols have been modified and SMB values in Gt/yr are only specified in the text as follows:

*"Differences in SMB values exceed 23 % in 2150 (-842 Gt yr$^{-1}$ for $SMB_{MSK_{NF}}$ against – 647 Gt yr-1 for $SMB_{MSK_{2W}}$)".*

**Figures:**

*Figure 1: Caption: What ice mask definition is used for the integral over the "entire GrIS", Clarify. Over which periods are the regression lines defined? Specify in the caption.*

We used the ice-sheet mask taken from Bamber et al. (2013). The regression lines are computed from 2000 to 2039 and from 2041 to 2099. This information now appears in the caption. Note also that in the previous version, the different values reported in the figures were given in m/yr. For consistency with the main text, we have changed the figure and converted the SMB components in Gt yr$^{-1}$.

*What does the black dashed line stand for? If it is for 0, a solid thin line may be better and should extend all the way to the left y-axis*

We replaced the black dashed zero line by a light grey solid line and indicated in the caption what it corresponds to.

*Figure 2: Why are b and c given on the MAR grid? Shouldn't they be given on the GRISLY grid for consistency?*

Snowfall and rainfall amounts are not directly used by GRISLI and are therefore not downscaled at the ISM resolution.

*In this figure as well as many other, anomalies are denoted x -- y. Not sure if the symbol "--" stands for minus. If so, why not use the minus sign "-" instead?*

As recommended we replaced "--" by "-" for anomaly notation.

*You should add a note that the colour key is non-linear.*
The colour scale is linear but not symmetric between positive and negative values. This has been specified in the caption

*The dashed grey surface elevation contours are not visible on a A4 printout of the paper (same for most of the other 2D plots). Consider using a darker colour (black). On the contrary, the lat/lon lines are much less important and could be less pronounced.*
Your recommendations have been taken into account for all 2D plots.

*Figure 3: Also here, it seems strange to report surface elevation as a time average, when you could as well just plot the 2150 result. Even stranger for the ELA! Just plot the 2000, 2100 and 2150 results in line with your legend in the figure. It looks like the 2000 and 2100 lines of the ELA are identical between experiments. You should mention that or that the lines are not (hardly) distinguished to avoid confusion. Use colours for the ELA lines that are not part of the colour key.*
As recommended, we no longer use time averaging for the surface elevation and the colors for the ELA have also been changed. However, as ELA is inferred from climatic variables that are averaged on a 10-year period throughout the paper (see our response to your general comment), we used a 10-year time-averaging for the ELA representation. Following your recommendation, we also changed the colours of ELA.

*Figure 4: Why not use the same colour scheme as in Figure 5a/b? That would make comparison easier.*
We now use the same color scale as Figures 5a/b.

*Figure 5: It would help the interpretation a lot to compare integrated SMB \*anomalies\* (Fig 2a) and integrated ice flux divergence \*anomalies\* instead. Now the two plots are on first view identical and it is hard to visually extract the difference.*
See our response to your comment P12 L8-9

*Explain in the text what the integrated ice flux divergence is when introducing EQU 2.*
We added the definition of ice flux divergence in Section 2.2.1 when we present the different terms of Equation 2.

*Add to caption that the grey shade is the land-sea mask.*
Instead of the land-sea mask, we specified in the caption that the grey shade corresponds to non-ice-covered areas. .

*Figure 6: Since you introduce 2W as the standard experiment, I would plot NF-2W and PF-2W instead (opposite sign). Add to caption that the grey shade is the land-sea mask. Remove the log10 label from the plot (applies also for other plots).*
As recommended we plotted NF-2W and PF-2W instead of 2W-NF and 2W-PF. We also changed the caption. For grey shade, see response to your comment on Fig. 5.

*Figure 7: Plot labels a,b,c, ... and refer to those in the caption.*
Done

*The Joughin et al (2010) data set has since been updated to full coverage over the GrIS. Consider using this improved data (ref below). The inset panels for observations are way to small. Include as additional full size panels on the top. You can gain extra space by removing redundant colour keys (the middle two are repeated). Remove "surface" before "velocity".*

*JOUGHIN, I., SMITH, B., & HOWAT, I. (2018). A complete map of Greenland ice velocity derived from satellite data collected over 20 years. Journal of Glaciology, 64(243), 1-11. doi:10.1017/jog.2017.73*
We have modified the Figure 7 as recommended.

*Figure 8: Same here, suggest to plot NF-2W differences instead.*
Modified as suggested.

*Figure 9: does not appear to be in the manuscript. I see only the caption!*
This should be ok for any pdf viewer now.

*Figure 10: Same comments as for Fig 8*
Modified as suggested.

*Figure 11: Shouldn't red in a) and blue in b) be identical? For some points that does not seem to be the case.*
There was an error in the labels (not in the caption though). The red in a) is NF-2W while the blue in b) is PF-2W

*Again, I doubt the averaging is really needed. If maintained, the labels should be adjusted to the mid-point of the averaging period.*
The plot has been redone without time averaging.

*Except for a few points, PF-NF is only negative. Why?*
See our response to your comment P14-L29.

*Figure 12: Please use a different set of colours in b and a to avoid confusion. The averaging is really not needed here.*
The caption was wrong, the averaging is not used here. We have corrected the caption in the new version of the manuscript.

*Supplement*

*Figure S1:Match figure title with caption (anomaly vs error).*
Caption corrected as recommended

*Figure S2: Mention that the scale is non-linear.*
The colour scale is linear but not symmetric between positive and negative values. Caption modified accordingly..

*Figure S3: Suggest to repeat and update the caption from S2 for convenience of the reader. Mention that the scale is non-linear.*
Updated as recommended.

*Figure S4: Mention that the scale is non-linear.*
As previously mentioned, the colour scale is linear but not symmetric between positive and negative values. This had been added in the caption

*Figure S5: Not clear what "loosing 100 % of the ice cover" means. I think you mean the pixels that have 0% ice cover in the end, even if they had less than 100 % to start with. Maybe "pixels loosing all of the initial ice cover"?*
You are right, we clarified the caption.

*Figure S6: Remove first part of the caption which is scrambled. Shouldn't the annual snowfall anomaly (a) be equal to the sum of the 4 seasons? Or are you plotting the snowfall anomaly in m/yr instead of m/yr? Same problem may apply to S7 and S4, but I don't know if plotting ratios solves it for you in the latter.*
With the seasonal values expressed in m/yr, the annual anomaly is not the sum of the 4 seasons but the mean of the 4 seasons.

*Figure S7: Suggest to repeat and update the caption from S6 for convenience of the reader.*
Done.

*Figure S8 Mention that the scale is non-linear. The red solid line is not visible for me on this plot.*
Once again, the colour scale is linear but not symmetric between positive and negative values. There was an error in the previous Fig. S8 caption (there no red line). We no longer refer to this line in the new version of the manuscript.

*Figure S10: I would e.g. call this a difference not an anomaly. I think it should be a reference to Eq. 2 instead.*
Plot and caption have been changed as suggested.

*Generally throughout the manuscript. I would use "differences" between different experiments and "anomalies" within the same experiment or when a differences reveals an anomalous signal. "Errors" can be used when a clear target reference exists.*
Thank you for the comment. This has been changed throughout the revised paper.

*Figure S11: Consider plotting an updated velocity field with full coverage:*

[revised manuscript text omitted]